

# Four-derivative couplings and BPS dyons
# in heterotic CHL orbifolds

**Guillaume Bossard[1], Charles Cosnier-Horeau[1,2,3] and Boris Pioline[2,3,4⋆]**

**1** Centre de Physique Théorique, Ecole Polytechnique, Université Paris-Saclay,
91128 Palaiseau Cedex, France
**2** Sorbonne Universités, UPMC Université Paris 6, UMR 7589, F-75005 Paris, France
**3** Laboratoire de Physique Théorique et Hautes Energies, CNRS UMR 7589,
Université Pierre et Marie Curie, 4 place Jussieu, 75252 Paris cedex 05, France
**4** Theoretical Physics Department, CERN, Case C01600, CH-1211 Geneva 23, Switzerland

⋆ boris.pioline@cern.ch

## Abstract

**Three-dimensional string models with half-maximal supersymmetry are believed to be invariant under a large U-duality group which unifies the S and T dualities in four dimensions. We propose an exact, U-duality invariant formula for four-derivative scalar couplings of the form $F(\Phi)(\nabla\Phi)^4$ in a class of string vacua known as CHL $\mathbb{Z}_N$ heterotic orbifolds with $N$ prime, generalizing our previous work which dealt with the case of heterotic string on $T^6$. We derive the Ward identities that $F(\Phi)$ must satisfy, and check that our formula obeys them. We analyze the weak coupling expansion of $F(\Phi)$, and show that it reproduces the correct tree-level and one-loop contributions, plus an infinite series of non-perturbative contributions. Similarly, the large radius expansion reproduces the exact $F^4$ coupling in four dimensions, including both supersymmetric invariants, plus infinite series of instanton corrections from half-BPS dyons winding around the large circle, and from Taub-NUT instantons. The summation measure for dyonic instantons agrees with the helicity supertrace for half-BPS dyons in 4 dimensions in all charge sectors. In the process we clarify several subtleties about CHL models in $D = 4$ and $D = 3$, in particular we obtain the exact helicity supertraces for 1/2-BPS dyonic states in all duality orbits.**

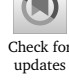

# 1 Introduction

In the absence of a first principle non-pertubative formulation of superstring theory, the study of string vacua with extended supersymmetry continues to be one of the few sources of insight into the strong coupling regime. By exploiting invariance under U-dualities, which the full quantum theory is believed to enjoy [1, 2, 3, 4], as well as supersymmetric Ward identities, it is often possible to determine certain couplings in the low energy effective action exactly, for all values of

the moduli (as demonstrated by [5] and numerous subsequent works). The expansion of these couplings near boundaries of the moduli space, corresponding to cusps of the U-duality group, then reveals, beyond power-like terms computable in perturbation theory, infinite series of exponentially suppressed corrections interpreted as semi-classical contributions in the putative string field theory. A particularly interesting class of examples is that of BPS saturated couplings in three-dimensional string vacua: in the limit where a circle in the internal space decompactifies, these couplings receive exponentially suppressed contributions from BPS states in four dimensions, along with further suppressed contributions from Taub-NUT instantons. These couplings can therefore be viewed as BPS black hole partitions, which encode the exact degeneracies (or more precisely, helicity supertraces) of BPS black hole micro-states [6, 7, 8].

In the recent letter [7], we investigated the $F(\Phi)(\nabla\Phi)^4$ and $G(\Phi)\nabla^2(\nabla\Phi)^4$ couplings in the low energy effective action of three-dimensional string vacua with 16 supercharges, focussing on the simplest example of such vacua, namely heterotic strings compactified on a torus $T^7$, or equivalently, type II strings compactified on $K3 \times T^3$. Based on the known perturbative contributions to these couplings, we conjectured exact formulae for the coefficients $F(\Phi)$ and $G(\Phi)$ for all values of the moduli $\Phi$, which satisfy the requisite supersymmetric Ward identities and are manifestly invariant under U-duality. In the limit where one circle inside $T^7$ decompactifies, we claimed that these formulae reproduce the correct helicity supertraces for 1/2-BPS and 1/4-BPS states with primitive charges, for all values of the moduli $\phi$ in four dimensions.

The goal of the present work is to demonstrate these claims in the case of the $(\nabla\Phi)^4$ coupling,[1] revisiting the analysis in [9], and extend our conjecture to a class of string vacua with 16 supercharges known as CHL orbifolds [10], restricting to $\mathbb{Z}_N$ orbifolds with $N$ prime for simplicity.

In Section 2, after reviewing relevant aspects of heterotic CHL vacua with 16 supercharges in four and three dimensions, we state the helicity supertraces of 1/2-BPS dyons with arbitrary charge in four dimensions (referring to Appendix A for the derivation of the perturbative BPS spectrum), and determine the precise form of the U-duality group $G_3(\mathbb{Z})$ in three dimensions, consistent with S-duality and T-duality in four dimensions. We then propose a manifestly U-duality invariant formula (2.27) for the coefficient $F_{abcd}(\Phi)$ of the $(\nabla\Phi)^4$ couplings, obtained by covariantizing the known one-loop contribution under $G_3(\mathbb{Z})$, extending the proposal in [7] for the maximal rank case ($N = 1$).

In Section 3, using superspace arguments we establish the supersymmetric Ward identities (2.23) which constrain the coupling $F_{abcd}(\Phi)$, and show that the proposal (2.27) satisfies these relations.

In Section 4, we analyze (2.27) in the limit where $g_3 \to 0$, and show that it reproduces the known tree-level and one-loop contributions in heterotic perturbation theory, plus an infinite series of NS5-brane, Kaluza–Klein monopole and H-monopole instanton corrections.

In Section 5, we similarly analyze (2.27) in the large radius limit $R \to \infty$, and show that it reproduces the known $F^4$ and $\mathcal{R}^2$ couplings in $D = 4$, along with an infinite series of exponentially suppressed corrections of order $e^{-R\mathcal{M}(Q,P)}$ with $Q$ and $P$ collinear, weighted by the helicity supertrace $\Omega_4(Q,P)$, and further exponentially suppressed corrections from Taub-NUT monopoles.

In most computations, we allow for lattices of arbitrary signature $(p,q)$, before specifying to the most relevant case $(p,q) = (2k,8)$ at the end. Details of some computations are relegated to Appendices. The one-loop vacuum amplitude for heterotic CHL models, from which the perturbative BPS spectrum, $F^4$ and $(\nabla\Phi)^4$ couplings are easily read off, is constructed in Appendix §A. In §B we decompose the Ward identity on all Fourier modes in the degeneration limit

---

[1] An analysis of the $\nabla^2(\nabla\Phi)^4$ couplings will appear in a separate publication.

$O(p, q) \rightarrow O(p - 1, q - 1)$, and show that all Fourier coefficients are uniquely determined up to a moduli-independent summation measure. In §C and §D we collect some notations which arise in the analysis of §4 and §5. In Appendix §E we obtain a Poincaré series representation of the relevant genus-one modular integrals, and use the same method to construct Eisenstein series for $O(p, q, \mathbb{Z})$.

## 2 Dualities, BPS spectrum and $(\nabla\Phi)^4$ couplings in CHL vacua

In this section, we recall relevant aspects of heterotic CHL vacua with 16 supercharges in four and three dimensions, restricting to the case of $\mathbb{Z}_N$ orbifolds with $N$ prime for simplicity. While most of the results are well known, we pay special attention to the quantization conditions for the electromagnetic charges of 4D dyons, and to the precise form of the U-duality groups in $D = 4$ and $D = 3$. Finally, we state our proposal for the non-perturbative $(\nabla\Phi)^4$ coupling, which is the focus of the remainder of this work.

### 2.1 Moduli space and 1/2-BPS dyons in $D = 4$

Recall that in four-dimensional string vacua with 16 supercharges, the moduli space is locally a product

$$\mathcal{M}_4 = \left[ \frac{SL(2, \mathbb{R})}{SO(2)} \times G_{r-6,6} \right] / G_4(\mathbb{Z}) , \tag{2.1}$$

where $G_{p,q} \equiv O(p, q)/[O(p) \times O(q)]$ denotes the orthogonal Grassmannian of positive $q$-planes in a fiducial vector space $\mathbb{R}^{p,q}$ of signature $(p, q)$ (a real symmetric space of dimension $pq$), $r$ is the rank of the Abelian gauge group, and $G_4(\mathbb{Z})$ is an arithmetic subgroup of $SL(2, \mathbb{R}) \times O(r - 6, 6, \mathbb{R})$. In heterotic string theory compactified on a torus $T^6$, the first factor is parametrized by the axiodilaton $S = b + 2\pi i/g_4^2$, where $b$ is the scalar dual to the Kalb-Ramond two-form, while the second factor, with $r = 28$, is the Narain moduli space [11]. The U-duality group $G_4(\mathbb{Z})$ is then the product of the S-duality group $SL(2, \mathbb{Z})$, acting on $S$ by fractional linear transformations $S \mapsto \frac{aS+b}{cS+d}$ [1, 2], and of the T-duality group $O(22, 6, \mathbb{Z})$, which is the automorphism group of the even self-dual Narain lattice $\Lambda_{22,6} = E_8 \oplus E_8 \oplus \mathit{II}_{6,6}$, where $E_8$ denotes the root lattice of $E_8$ and $\mathit{II}_{d,d}$ denotes $d$ copies of the standard hyperbolic lattice $\mathit{II}_{1,1}$. The effective action is singular on real codimension-6 loci where the projection $Q_R$ of a vector $Q \in \Lambda_{22,6}$ with norm $Q^2 = 2$ on the negative 6-plane parametrized by $G_{r-6,6}$ vanishes, corresponding to points of gauge symmetry enhancement. The same moduli space (2.1) arises in type IIA string compactified on $K3 \times T^2$, where the first factor parametrizes the Kähler modulus of $T^2$, while the second factor parametrizes the axiodilaton, the complex modulus of $T^2$, the $K3$ moduli and the holonomies of the RR gauge fields on $T^2 \times K3$. These two string vacua are in fact related by heterotic/type II duality [12], which in particular turns S-duality into a geometrical symmetry.

Vacua with lower values of $r$ can be constructed as freely acting orbifolds of the maximal rank model with $r = 28$ [10, 13, 14, 15]. On the heterotic side, one mods out by a $\mathbb{Z}_N$ rotation of the heterotic lattice $\Lambda_{22,6}$ at values of the Narain moduli where such a symmetry exists, combined with an order $N$ shift along one circle inside $T^6$. This projection removes $28 - r$ of the gauge fields in 4 dimensions, along with their scalar partners. On the type II side, one can similarly mod out by a symplectic automorphism of order $N$ on K3, combined with an order $N$ shift on $T^2$. It is convenient to label this action by the data $\{m(a), a|N\}$ and the associated cycle shape $\prod_{a|N} a^{m(a)}$ such that $\sum_{a|N} am(a) = 24$, corresponding to the cycle decomposition of the $\mathbb{Z}_N$ action on the

| $N$ | Cycle Shape | $k$ | $r$ | $\Lambda_{k,8-k}$ | $\Lambda_m \cong \Lambda_e^*$ | $|\Lambda_m^*/\Lambda_m|$ |
|---|---|---|---|---|---|---|
| 1 | $1^{24}$ | 12 | 28 | | $E_8 \oplus E_8 \oplus II_{6,6}$ | 1 |
| 2 | $1^8 2^8$ | 8 | 20 | $E_8[2]$ | $E_8[2] \oplus II_{1,1}[2] \oplus II_{5,5}$ | $2^{10}$ |
| 3 | $1^6 3^6$ | 6 | 16 | $D_6[3] \oplus D_2[-1]$ | $A_2 \oplus A_2 \oplus II_{3,3}[3] \oplus II_{3,3}$ | $3^8$ |
| 5 | $1^4 5^4$ | 4 | 12 | $D_4[5] \oplus D_4[-1]$ | $II_{3,3}[5] \oplus II_{3,3}$ | $5^6$ |
| 7 | $1^3 7^3$ | 3 | 10 | $D_3[7] \oplus D_5[-1]$ | $\left[\begin{smallmatrix} -4 & -1 \\ -1 & -2 \end{smallmatrix}\right] \oplus II_{2,2}[7] \oplus II_{2,2}$ | $7^5$ |

Table 1: A class of $\mathbb{Z}_N$ CHL orbifolds. Here $k = 24/(N+1)$ is the weight of the cusp form whose inverse counts 1/2 BPS states, $r = 2k+4$ is the rank of the gauge group and $\Lambda_m$ is the lattice of magnetic charges in four dimensions. The discriminant group $\Lambda_m^*/\Lambda_m$ is isomorphic to $\mathbb{Z}_N^{k+2}$. Agreement between the lattice $\Lambda_m$ listed here and $\Lambda_{r-6,6}$ defined in (2.2) follows from the lattice isomorphisms (A.33).

even homology lattice $H_{\text{even}}(K3) \sim \mathbb{Z}^{24}$. For simplicity we shall restrict ourselves to CHL orbifolds with $N$ prime and cycle shape $1^k N^k$ with $k = 24/(N+1)$. In this case, one can decompose $\Lambda_{22,6} = \Lambda_{Nk,8-k} \oplus II_{1,1} \oplus II_{k-3,k-3}$, such that the $\mathbb{Z}_N$ action acts on the first term by a $\mathbb{Z}_N$ rotation, on the second term by an order $N$ shift, leaving $II_{k-3,k-3}$ invariant (see §A.2 for details on this construction). We denote by $\Lambda_{k,8-k}$ the quotient of $\Lambda_{Nk,8-k}$ under the $\mathbb{Z}_N$ rotation (see Table 1). The U-duality group $G_4(\mathbb{Z})$ includes $\Gamma_1(N) \times \widetilde{O}(r-6,6,\mathbb{Z})$, where $\Gamma_1(N)$ is the congruence subgroup of $SL(2,\mathbb{Z})$ corresponding to matrices $\left(\begin{smallmatrix} a & b \\ c & d \end{smallmatrix}\right)$ with $c = 0 \bmod N, a = d = 1 \bmod N$, and $\widetilde{O}(r-6,6,\mathbb{Z})$ is the restricted automorphism group of the lattice

$$\Lambda_{r-6,6} = \Lambda_{k,8-k} \oplus II_{1,1}[N] \oplus II_{k-3,k-3} , \tag{2.2}$$

*i.e.* the subgroup of the automorphism group of $\Lambda_{r-6,6}$ which acts trivially on the discriminant group $\Lambda_{r-6,6}^*/\Lambda_{r-6,6}$. Here and below, for any lattice $\Lambda$, we denote by $\Lambda[\alpha]$ the same lattice with a quadratic form rescaled by a factor $\alpha$ (which is equivalent to rescaling the lattice vectors by $\sqrt{\alpha}$). Note that the lattice (2.2) is still even, *i.e.* $Q^2 \in 2\mathbb{Z}$ for $Q \in \Lambda_{r-6,6}$, but it is no longer unimodular, rather it is a lattice of level $N$, in the sense that $Q^2 \in 2\mathbb{Z}/N$ for any $Q \in \Lambda_{r-6,6}^*$. Singularities now occur on codimension-$q$ loci where $Q_R^2 = 0$ for a norm 2 vector $Q \in \Lambda_{r-6,6}$, or for a norm $2/N$ vector $Q \in \Lambda_{r-6,6}^*$.

While the U-duality group $G_4(\mathbb{Z})$ must certainly include $\Gamma_1(N) \times \widetilde{O}(r-6,6,\mathbb{Z})$, it may actually be larger. Moreover, special BPS observables may well be invariant under an even larger group. In particular the four-derivative couplings in $D = 4$ turn out to be invariant under the action of the larger duality group $\Gamma_0(N) \times O(r-6,6,\mathbb{Z})$, where $\Gamma_0(N)$ is the subgroup of matrices $\left(\begin{smallmatrix} a & b \\ c & d \end{smallmatrix}\right)$ with $c = 0 \bmod N$ and $O(r-6,6,\mathbb{Z})$ is the full automorphism group of the lattice $\Lambda_{r-6,6}$. For example, the exact $\mathcal{R}^2$ coupling in the low-energy effective action is given by [19, 20, 21]

$$-\frac{1}{(8\pi)^2} \int d^4x \sqrt{-g} \log(S_2^k |\Delta_k(S)|^2)(\mathcal{R}_{\mu\nu\rho\sigma}\mathcal{R}^{\mu\nu\rho\sigma} - 4\mathcal{R}_{\mu\nu}\mathcal{R}^{\mu\nu} + \mathcal{R}^2) , \tag{2.3}$$

where $\Delta_k$ is the unique cusp form of weight $k$ under $\Gamma_0(N)$, nowhere vanishing except at the cusps $i\infty$ and 0,

$$\Delta_k(\tau) = \eta^k(\tau)\eta^k(N\tau) . \tag{2.4}$$

In the weak coupling limit $S_2 \to \infty$, the expansion

$$-\log(S_2^k|\Delta_k(S)|^2) = 4\pi S_2 - k\log S_2 + k\sum_{m=1}^{\infty}\left(\sum_{d|m}d + \sum_{Nd|m}Nd\right)\frac{q_S^m + \bar{q}_S^m}{m}, \qquad (2.5)$$

with $q_S = e^{2\pi iS}$ reveals, beyond the expected tree-level contribution and logarithmic mixing with the non-local part of the effective action, an infinite series of exponentially suppressed corrections ascribed to NS5-branes wrapped on $T^6$ [19]. While not all $\Gamma_0(N) \times O(r-6,6,\mathbb{Z})$ transformations are expected to be U-dualities of the full theory but only of the BPS sector, for brevity we shall refer to them respectively as S- and T-dualities.

In [18] it was observed that the coupling (2.3) is in fact invariant under the larger group $\widehat{\Gamma}_0(N)$, obtained by adjoining to $\Gamma_0(N)$ the Fricke involution, which acts on modular forms of weight $k$ under $\Gamma_0(N)$ via $f_k(\tau) \mapsto \hat{f}_k(\tau) = (-i\tau\sqrt{N})^{-k}f_k(-1/(N\tau))$. Based on a detailed study of geometric dualities in the type II dual description, it was conjectured[2] that the full U-duality group in $D = 4$ also includes the so-called Fricke S-duality, which acts on the first factor in (2.1) by the Fricke involution $S \mapsto -1/(NS)$, accompanied by a suitable action of $O(r-6,6,\mathbb{R})$ on the second factor. Additional evidence for the existence of Fricke S-duality comes from the spectrum of BPS states, to which we now turn.

Point-like particles in $D = 4$ carry electric and magnetic charges $(Q, P) \in \Lambda_{em}$ under the $r$ Maxwell fields, where

$$\Lambda_{em} = \Lambda_e \oplus \Lambda_m, \quad \Lambda_m = \Lambda_{r-6,6} = \Lambda_e^*. \qquad (2.6)$$

The lattice $\Lambda_m$ is tabulated in the sixth column of Table 1, taken from [18]. It agrees with the result (2.2) upon making use of the lattice isomorphisms (A.33). In view of the remarks below (2.2), one has, for any $(Q, P) \in \Lambda_{em}$,

$$Q^2 \in \frac{2}{N}\mathbb{Z}, \quad P^2 \in 2\mathbb{Z}, \quad P \cdot Q \in \mathbb{Z}. \qquad (2.7)$$

The last property in particular ensures that the Dirac-Schwinger-Zwanziger pairing $Q \cdot P' - Q' \cdot P$ is integer. Moreover, it was observed in [18] that the lattice $\Lambda_m$ is in fact $N$-modular, *i.e.* it satisfies

$$\Lambda_m^* \simeq \Lambda_m[1/N]. \qquad (2.8)$$

In other words, there exists an $O(r-6,6,\mathbb{R})$ matrix $\sigma$ such that $\sqrt{N}\sigma$ maps the lattice $\Lambda_m$ into itself and such that

$$\Lambda_m^* = \frac{\sigma}{\sqrt{N}}\Lambda_m \quad (\supset \Lambda_m). \qquad (2.9)$$

A simple example of $N$-modular lattice is $\Lambda_{d,d}[N] \oplus \Lambda_{d,d}$, which is relevant for $N = 5$ above. In this case one can parametrize an element in the lattice in $(\mathbb{Z}^d, N\mathbb{Z}^d, \mathbb{Z}^d, \mathbb{Z}^d)$ and an element of the dual lattice in $(\mathbb{Z}^d/N, \mathbb{Z}^d, \mathbb{Z}^d, \mathbb{Z}^d)$ and define $\sigma \in O(2d, 2d, \mathbb{R})$ such that

$$\frac{\sigma}{\sqrt{N}} = \frac{1}{\sqrt{N}}\begin{pmatrix} 0 & 0 & \frac{1}{\sqrt{N}}\mathbb{1}_{d,d} & 0 \\ 0 & 0 & 0 & \sqrt{N}\mathbb{1}_{d,d} \\ \sqrt{N}\mathbb{1}_{d,d} & 0 & 0 & 0 \\ 0 & \frac{1}{\sqrt{N}}\mathbb{1}_{d,d} & 0 & 0 \end{pmatrix} = \begin{pmatrix} 0 & 0 & \frac{1}{N}\mathbb{1}_{d,d} & 0 \\ 0 & 0 & 0 & \mathbb{1}_{d,d} \\ \mathbb{1}_{d,d} & 0 & 0 & 0 \\ 0 & \frac{1}{N}\mathbb{1}_{d,d} & 0 & 0 \end{pmatrix}.$$
$$(2.10)$$

---

[2]More generally, Fricke S-duality is conjectured to hold whenever the cycle shape satisfies the balancing condition $m(a) = m(N/a)$ for all $a|N$. [18]

The map (2.9) defines the action $(Q, P) \mapsto (-\sigma \cdot P/\sqrt{N}, \sigma^{-1} \cdot Q\sqrt{N})$ of the Fricke S-duality on $\Lambda_{em}$, which maps $(Q^2, P^2, P \cdot Q) \mapsto (P^2/N, NQ^2, -P \cdot Q)$ and therefore preserves the quantization conditions (2.7). It also allows to identify $N\Lambda_m^*$ as a sublattice of $\Lambda_m$

$$N\Lambda_m^* = \sqrt{N}\sigma\Lambda_m \subset \Lambda_m \, . \tag{2.11}$$

Electric charge vectors $Q \in \Lambda_m \subset \Lambda_e$ are called untwisted, while vectors $Q \in \Lambda_e \smallsetminus \Lambda_m$ are called twisted. More generally, we shall call dyonic charge vectors $(Q, P)$ lying in $\Lambda_m \oplus N\Lambda_e \subset \Lambda_e \oplus \Lambda_m$ untwisted, and twisted otherwise.[3] Untwisted dyons are in particular such that

$$Q^2 \in 2\mathbb{Z} \, , \quad P^2 \in 2N\mathbb{Z} \, , \quad P \cdot Q \in N\mathbb{Z} \, . \tag{2.12}$$

Half-BPS states exist only when $Q, P$ are collinear. Their mass is then determined in terms of the charges via

$$\mathcal{M}^2(Q, P) = \frac{2}{S_2}(Q_R - SP_R) \cdot (Q_R - \bar{S}P_R) \, , \tag{2.13}$$

where, for a vector $Q^I \in \mathbb{R}^{p,q}$ ($I = 1 \ldots p+q$), we denote by $Q_L^a$ ($a = 1 \ldots p$) and $Q_R^{\hat{a}}$ ($\hat{a} = 1 \ldots q$) its projections on the positive $p$-plane and its orthogonal complement parametrized by the orthogonal Grassmannian $G_{p,q}$, such that $Q^2 = Q_L^2 - Q_R^2$.

For primitive purely electric states (such that $Q \in \Lambda_e$ but $Q/d \notin \Lambda_e$ for all $d > 1$), corresponding to left-moving excitations in the twisted sectors of the perturbative heterotic string, it is known that the helicity supertrace $\Omega_4(Q, 0)$ is given by [22, 17, 24, 25, 23]

$$\Omega_4(Q, 0) = c_k\left(-\frac{NQ^2}{2}\right) \, , \quad \frac{1}{\Delta_k(\tau)} = \sum_{\substack{m \in \mathbb{Z} \\ m \geq -1}} c_k(m) q^m = \frac{1}{q} + k + \ldots, \tag{2.14}$$

where $q = 2^{2\pi i \tau}$ and $\Delta_k(\tau)$ is the same cusp form (2.4) which enters in the exact $\mathcal{R}^2$ coupling. In Appendix A, we rederive this result by constructing the one-loop vacuum amplitude for the CHL models under consideration, and show that primitive purely electric states corresponding to left-moving excitations in the untwisted sector have an additional contribution (first observed for $N = 2$ in [26])

$$\Omega_4(Q, 0) = c_k\left(-\frac{Q^2}{2}\right) + c_k\left(-\frac{NQ^2}{2}\right) \, . \tag{2.15}$$

Invariance under both $\Gamma_0(N)$ and Fricke S-duality implies that the same formulae apply to generic primitive dyons with $Q^2$ being replaced by $\frac{1}{N}\gcd(NQ^2, P^2, Q \cdot P)$. It follows that the helicity supertrace for general 1/2 BPS primitive dyons is given by

$$\Omega_4(Q, P) = c_k\left(-\frac{\gcd(NQ^2, P^2, Q \cdot P)}{2}\right) \, . \tag{2.16}$$

for twisted electromagnetic charge $(Q, P) \in (\Lambda_e \oplus \Lambda_m) \smallsetminus (\Lambda_m \oplus N\Lambda_e)$, and by

$$\Omega_4(Q, P) = c_k\left(-\frac{\gcd(NQ^2, P^2, Q \cdot P)}{2}\right) + c_k\left(-\frac{\gcd(NQ^2, P^2, Q \cdot P)}{2N}\right) \, . \tag{2.17}$$

for untwisted charge $(Q, P) \in \Lambda_m \oplus N\Lambda_e$. In contrast, primitive 1/2-BPS states of the maximal rank theory have a single contribution

$$\Omega_4(Q, P) = c\left(-\frac{\gcd(Q^2, P^2, Q \cdot P)}{2}\right) \, , \quad \frac{1}{\Delta(\tau)} = \sum_{\substack{m \in \mathbb{Z} \\ m \geq -1}} c(m) q^m = \frac{1}{q} + 24 + \ldots \tag{2.18}$$

---

[3]Note that this terminology is defined to be consistent with Fricke and $\Gamma_0(N)$ S-duality, but twisted magnetic charges do not correspond to any twisted sector in the conventional sense.

## 2.2 Moduli space and 1/2-BPS couplings in $D = 3$

Upon further compactification on a circle, additional moduli arise from the radius $R$ of the circle, from the holonomies $a^{1I}$ of the $r$ gauge fields, and from the scalars $a^{2I}, \psi$ dual to the $r$ Maxwell fields and to the Kaluza–Klein gauge field in three dimensions, extending (2.1) to [27]

$$\mathcal{M}_3 = G_{r-4,8}/G_3(\mathbb{Z}) . \tag{2.19}$$

The U-duality group $G_3(\mathbb{Z})$ includes $G_4(\mathbb{Z})$, the Heisenberg group of large gauge transformations acting on $a^{I,i}, \psi$, and the automorphism group $O(r-5,7,\mathbb{Z})$ (or rather a subgroup containing $\widetilde{O}(r-5,7,\mathbb{Z})$) of the Narain lattice $\Lambda_{r-5,7} = \Lambda_{r-6,6} \oplus II_{1,1}$ corresponding to T-duality in heterotic string compactified on $T^7$. The action of these subgroups is most easily seen in the vicinity of the cusps $R \to \infty$ and $g_3 \to 0$, corresponding to the decompactification limit to $D = 4$ and the weak heterotic coupling limit in $D = 3$, where (2.19) reduces to

$$\mathcal{M}_3 \to \begin{cases} \mathbb{R}_R^+ \times \mathcal{M}_4 \times \tilde{T}^{2r+1} \\ \mathbb{R}_{1/g_3^2}^+ \times \left[ \frac{O(r-5,7)}{O(r-5) \times O(7)} / O(r-5,7,\mathbb{Z}) \right] \times T^{r+2} \end{cases} \tag{2.20}$$

Here, $\tilde{T}^{2r+1}$ is a circle bundle over the torus $T^{2r}$ parametrized by the holonomies $a^{i,I}$, with fiber parametrized by the NUT potential $\psi$, while $T^{r+2}$ corresponds to the scalars dual to the Maxwell gauge fields after compactifying the heterotic string on $T^7$. In heterotic perturbation theory, the effective action in $D = 3$ is singular on codimension-7 loci where $Q_R^2 = 0$ for a norm 2 vector $Q \in \Lambda_{r-5,7}$, or for a norm $2/N$ vector $Q \in \Lambda_{r-5,7}^*$.

For the case $r = 28$, it is well-known that these subgroups generate the automorphism group $O(24,8,\mathbb{Z})$ of the 'non-perturbative Narain lattice' $\Lambda_{24,8} = \Lambda_{22,6} \oplus II_{2,2}$ [28]. To the extent of our knowledge, the U-duality group for CHL models has not been discussed in the literature, but it is natural to expect that it includes the restricted automorphism group $\widetilde{O}(r-4,8,\mathbb{Z})$ of an extended Narain lattice of the form $\Lambda_{r-4,8} = \Lambda_m \oplus \Lambda_{2,2}$. We find that the following choice reproduces the correct S and T-dualities in $D = 4$:

$$\Lambda_{r-4,8} = \Lambda_m \oplus II_{1,1} \oplus II_{1,1}[N] , \tag{2.21}$$

where $II_{1,1}[N]$ is the standard hyperbolic lattice with quadratic form rescaled by a factor of $N$, such that $\Lambda_{r-4,8}^*/\Lambda_{r-4,8} \simeq \mathbb{Z}_N^{k+4}$. In terms of the usual construction of $II_{2,2}$ by windings $(n_1, n_2) \in \mathbb{Z}^2$, momenta $(m_1, m_2) \in \mathbb{Z}^2$ and quadratic form $2m_1 n_1 + 2m_2 n_2$, we define $II_{1,1} \oplus II_{1,1}[N]$ as the sublattice of $II_{2,2}$ where $n_2$ is restricted to be a multiple of $N$. The restricted automorphism group of $II_{1,1} \oplus II_{1,1}[N]$ was determined in [18, 29], and includes $\sigma_{T \leftrightarrow S} \ltimes [\Gamma_1(N) \times \Gamma_1(N)]$, acting by fractional linear transformations on the moduli $(T, S)$ parametrizing $G_{2,2}$, such that $|m_1 + Sm_2 + Tn_1 + STn_2|^2/(S_2 T_2)$ is invariant (see [20, §C], case V for $N = 2$, or [30, §3.1.3] for arbitrary $N$). In the present context, $T$ is interpreted as $\psi + iR^2$, while $S$ is the heterotic axi-odilaton. Thus, $\widetilde{O}(r-4,8,\mathbb{Z})$ contains the S-duality group $\Gamma_1(N)$ and T-duality group $\widetilde{O}(r-6,6,\mathbb{Z})$ in four dimensions. In addition, Fricke S-duality in four dimensions follows from the fact that the non-perturbative lattice (2.21) is itself $N$-modular,

$$\Lambda_{r-4,8}^* \simeq \Lambda_{r-4,8}[1/N] . \tag{2.22}$$

More evidence for the claim (2.21) will come from the analysis of BPS couplings in $D = 3$, to which we now turn.

In this work, we focus on the coupling of the form $F(\Phi)(\nabla\Phi)^4$ in the low energy effective action in $D = 3$, where $F(\Phi)$ is a symmetric rank four tensor $F_{abcd}(\Phi)$, and $(\nabla\Phi)^4$ is a shorthand notation for a particular contraction of the pull-back of the right-invariant one-forms $P_{a\hat{a}}$ on $G_{r-4,8}$ to $\mathbb{R}^{2,1}$ (see (3.15)). As stated in [7], and further explained below, supersymmetry requires that the coefficient $F_{abcd}(\Phi)$ satisfies the tensorial differential equations

$$\mathcal{D}_{(e}{}^{\hat{g}}\mathcal{D}_{f)\hat{g}}F_{abcd} = \frac{2-q}{4}\,\delta_{ef}\,F_{abcd} + (4-q)\,\delta_{e)(a}\,F_{bcd)(f} + 3\,\delta_{(ab}\,F_{cd)ef)} + \frac{15k}{(4\pi)^2}\delta_{(ab}\delta_{cd}\delta_{ef)}\delta_{q,6}\,, \tag{2.23a}$$

$$\mathcal{D}_{[e}{}^{[\hat{e}}\mathcal{D}_{f]}{}^{\hat{f}]}F_{abcd} = 0\,, \qquad\qquad \mathcal{D}_{[e}{}^{\hat{a}}F_{a]bcd} = 0\,, \tag{2.23b}$$

where the constant term in the first line occurs from the regularisation in $q = 6$ (see 3.57), and where $\mathcal{D}_{a\hat{b}}$ are the covariant derivatives in tangent frame on $G_{p,q}$. In fact, we shall show that all components of the tensor $F_{abcd}$ can be recovered from its trace $F_{\mathrm{tr}}(\Phi) \equiv F_{ab}^{\ ab}(\Phi)$ by acting with the differential operators $\mathcal{D}_{a\hat{b}}$ (see (3.26)). Supersymmetry requires that $F_{\mathrm{tr}}(\Phi)$ be an eigenmode of the Laplacian on $G_{r-4,8}$ with a specified eigenvalue, while U-duality requires that it should be invariant under $\widetilde{O}(r-4,8,\mathbb{Z})$. (Note however that the second order differential equations satisfied by $F_{\mathrm{tr}}(\Phi)$ does not imply (2.23), so it should not be thought of as a prepotential for $F_{abcd}$.)

In CHL $\mathbb{Z}_N$ orbifold of heterotic string on $T^7$, $F_{abcd}$ gets tree-level and one-loop contributions, both of which are solutions of (2.23), invariant under the full T-duality group $O(r-5,7,\mathbb{Z})$. As we show in Appendix A, the one-loop contribution is given by a modular integral[4]

$$F_{abcd}^{(\text{1-loop})} = \text{R.N.} \int_{\Gamma_0(N)\backslash\mathcal{H}} \frac{d\tau_1 d\tau_2}{\tau_2^2} \frac{\Gamma_{\Lambda_{r-5,7}}[P_{abcd}]}{\Delta_k(\tau)}\,, \tag{2.24}$$

where $\Delta_k(\tau)$ is the same cusp form (2.4) which appeared in the $\mathcal{R}^2$ couplings in $D = 4$, and $\Gamma_{\Lambda_{p,q}}[P_{abcd}]$ denotes the Siegel–Narain theta series for the lattice $\Lambda_{p,q}$,

$$\Gamma_{\Lambda_{p,q}}[P_{abcd}] = \tau_2^{q/2} \sum_{Q\in\Lambda_{p,q}} P_{abcd}(Q)\,e^{i\pi Q_L^2\tau - i\pi Q_R^2\bar{\tau}}\,, \tag{2.25}$$

with an insertion of the polynomial

$$P_{abcd}(Q) = Q_{L,a}Q_{L,b}Q_{L,c}Q_{L,d} - \frac{3}{2\pi\tau_2}\delta_{(ab}Q_{L,c}Q_{L,d)} + \frac{3}{16\pi^2\tau_2^2}\delta_{(ab}\delta_{cd)}, \tag{2.26}$$

$\Gamma_0(N)\backslash\mathcal{H}$ is any fundamental domain for the action of $\Gamma_0(N)$ on the Poincaré upper half-plane $\mathcal{H}$, and R.N. denotes a suitable regularization prescription (see (3.30)). In view of the form of the one-loop contribution, it is therefore natural to conjecture [9, 7] that the exact $(\nabla\Phi)^4$ coupling is the obvious generalization of (2.24), where the Narain lattice $\Gamma_{\Lambda_{r-5,7}}$ is replaced by its non-perturbative extension (2.21),

$$F_{abcd}(\Phi) = \text{R.N.} \int_{\Gamma_0(N)\backslash\mathcal{H}} \frac{d\tau_1 d\tau_2}{\tau_2^2} \frac{\Gamma_{\Lambda_{r-4,8}}[P_{abcd}]}{\Delta_k(\tau)}\,. \tag{2.27}$$

A similar formula holds for the trace part $F_{\mathrm{tr}}(\Phi) \equiv \delta^{ab}\delta^{cd}F_{abcd}(\Phi)$,

$$F_{\mathrm{tr}}(\Phi) = \text{R.N.} \int_{\Gamma_0(N)\backslash\mathcal{H}} \frac{d\tau_1 d\tau_2}{\tau_2^2} \Gamma_{\Lambda_{r-4,8}} \cdot D_{-k+2}D_{-k}\frac{1}{\Delta_k(\tau)}\,, \tag{2.28}$$

---

[4]A similar computation for four-graviton couplings in CHL models was performed in [31].

where $D_w = \frac{i}{\pi}(\partial_\tau - \frac{iw}{2\tau_2})$ is the Maass raising operator, mapping modular forms of weight $w$ to weight $w + 2$. The proposals (2.27) and (2.28) are manifestly invariant (or covariant) under the full automorphism group $O(r-4, 8, \mathbb{Z})$ of the non-perturbative lattice (2.21), which contains the true U-duality group in $D = 3$. Moreover, since the latter is $N$-modular, $\Gamma_{\Lambda_{r-4,8}}$ is invariant under the combined action of the Fricke involution on $\mathcal{H}$ and the rotation $\sigma \in O(r-4, 8, \mathbb{R})$ realizing the isomorphism (2.22),

$$\Gamma_{\Lambda_{r-4,8}}(\Phi, \tau)[P_{abcd}] = \left(-i\tau\sqrt{N}\right)^{-k} \Gamma_{\Lambda_{r-4,8}}[P_{abcd}]\left(\sigma \cdot \Phi, -\frac{1}{N\tau}\right) . \tag{2.29}$$

Since $\Delta_k$ is also an eigenmode of the Fricke involution on $\mathcal{H}$, and since the fundamental domain $\Gamma_0(N)\backslash\mathcal{H}$ can be chosen to be invariant under this involution, it follows that $F_{abcd}(\Phi)$ (and therefore $F_{\text{tr}}(\Phi)$) is covariant (invariant) under the action of $\sigma$ on $G_{r-4,8}$. As already anticipated, this action descends to Fricke S-duality in $D = 4$.

It is also important to note that the couplings (2.27) and (2.28) are singular on codimension-8 loci where $Q_R^2 = 0$ for some norm 2 vector $Q \in \Lambda_{r-4,8}$, or norm $2/N$ vector $Q \in \Lambda^*_{r-4,8}$. When the vector $Q$ is of the form $Q = (0, \widetilde{Q}, 0) \in \Lambda_{r-4,8}$ with $\widetilde{Q} \in \Lambda_{r-5,7}$, this singularity is visible at the level of the one-loop correction to the $(\nabla\Phi)^4$ coupling, and is due to additional states becoming massless. However, the one-loop correction is singular in real codimension 7, while the full non-perturbative coupling (assuming that (2.27) is correct) is singular in real codimension 8. Indeed, the invariant norm $Q_R^2 = \widetilde{Q}_R^2 + \frac{1}{2}g_3^2(\widetilde{Q} \cdot a)^2$ vanishes only when both $\widetilde{Q}_R^2 = 0$ and $\widetilde{Q} \cdot a = 0$. This partial resolution may be seen as an analogue of the resolution of the conifold singularity on the vector multiplet branch in type II strings compactified on a CY threefold times a circle, or equivalently on the hypermultiplet branch in the mirror description [32]. Singularities associated to generic vectors $Q \in \Lambda_{r-4,8}$ are not visible at any order in perturbation theory, and are associated to 'exotic' particles in $D = 3$ becoming massless [33, 34].

## 3 Establishing and solving supersymmetric Ward identities

In this section, we establish the supersymmetric Ward identities (2.23), from linearized superspace considerations, relate the components of the tensor $F_{abcd}$ to its trace $F_{\text{tr}} \equiv F_{ab}^{\ ab}$, and show that the genus-one modular integral (2.27) obeys this identity. For completeness, we solve the first equation of (2.23) in appendix B, and show that it is satisfied by each Fourier mode of $F_{abcd}$.

### 3.1 $(\nabla\Phi)^4$ type invariants in three dimensions

In three dimensional supergravity with half-maximal supersymmetry, the linearised superfield $W_{\hat{a}a}$ satisfies the constraints [27, 35, 36]

$$D_\alpha^i W_{\hat{a}a} = (\Gamma_{\hat{a}})^{ij}\chi_{\alpha\hat{j}a} , \qquad D_\alpha^i \chi_{\beta\hat{j}a} = -i(\sigma^\mu)_{\alpha\beta}(\Gamma^{\hat{a}})_{\hat{j}}^{\ i}\partial_\mu W_{\hat{a}a} , \tag{3.1}$$

with $\hat{a} = 1$ to 8 for the vector of $O(8)$, $i = 1$ to 8 for the positive chirality Weyl spinor of $Spin(8)$ and $\hat{i} = 1$ to 8 for the negative chirality Weyl spinor. The 1/2 BPS linearised invariants are defined using harmonics of $Spin(8)/U(4)$ parametrizing a $Spin(8)$ group element $u^r_{\ i}, u_{ri}$ in the Weyl spinor representation of positive chirality [37],

$$2u_{r(i}u^r_{\ j)} = \delta_{ij} , \qquad \delta^{ij}u_{ri}u^s_{\ j} = \delta_r^s , \qquad \delta^{ij}u_{ri}u_{sj} = 0 , \qquad \delta^{ij}u^r_{\ i}u^s_{\ j} = 0 , \tag{3.2}$$

$u_{r\hat{\imath}}, u^r{}_{\hat{\imath}}$ in the Weyl spinor representation of negative chirality,

$$2u_{r(\hat{\imath}}u^r{}_{\hat{\jmath})} = \delta_{\hat{\imath}\hat{\jmath}}\,, \qquad \delta^{\hat{\imath}\hat{\jmath}}u_{r\hat{\imath}}u^s{}_{\hat{\jmath}} = \delta^s_r\,, \qquad \delta^{\hat{\imath}\hat{\jmath}}u_{r\hat{\imath}}u_{s\hat{\jmath}} = 0\,, \qquad \delta^{\hat{\imath}\hat{\jmath}}u^r{}_{\hat{\imath}}u^s{}_{\hat{\jmath}} = 0\,, \tag{3.3}$$

and $u^+{}_{\hat{a}}, u^{rs}{}_{\hat{a}}, u^-{}_{\hat{a}}$ in the vector representation,

$$2u^+{}_{(\hat{a}}u^-{}_{\hat{b})} + \frac{1}{2}\varepsilon_{rstu}u^{rs}{}_{\hat{a}}u^{tu}{}_{\hat{b}} = \delta_{\hat{a}\hat{b}}\,, \quad \delta^{\hat{a}\hat{b}}u^+{}_{\hat{a}}u^-{}_{\hat{b}} = 1\,, \quad \delta^{\hat{a}\hat{b}}u^{rs}{}_{\hat{a}}u^{tu}{}_{\hat{b}} = \frac{1}{2}\varepsilon^{rstu}\,,$$

$$\delta^{\hat{a}\hat{b}}u^+{}_{\hat{a}}u^+{}_{\hat{b}} = 0\,, \quad \delta^{\hat{a}\hat{b}}u^+{}_{\hat{a}}u^{rs}{}_{\hat{b}} = 0\,, \quad \delta^{\hat{a}\hat{b}}u^-{}_{\hat{a}}u^{rs}{}_{\hat{b}} = 0\,, \quad \delta^{\hat{a}\hat{b}}u^-{}_{\hat{a}}u^-{}_{\hat{b}} = 0\,, \tag{3.4}$$

with $r = 1$ to $4$ of $U(4)$. They are related through the relations

$$u^+{}_{\hat{a}}u_{ri}(\Gamma^{\hat{a}})^{ij} = \sqrt{2}u_{r\hat{\imath}}\delta^{\hat{\imath}\hat{\jmath}}\,, \quad u_{ri}u_{s\hat{\jmath}}(\Gamma_{\hat{a}})^{ij} = \varepsilon_{rstu}u^{tu}{}_{\hat{a}}\,, \quad u^{rs}{}_{\hat{a}}u_{ti}(\Gamma^{\hat{a}})^{ij} = 2\delta^{[r}_t u^{s]}{}_{\hat{\imath}}\delta^{\hat{\imath}\hat{\jmath}}\,,$$

$$u_{ri}u^s{}_{\hat{\jmath}}(\Gamma_{\hat{a}})^{ij} = \sqrt{2}u^-{}_{\hat{a}}\delta^s_r\,, \quad u^r{}_i u_{s\hat{\jmath}}(\Gamma_{\hat{a}})^{ij} = \sqrt{2}u^+{}_{\hat{a}}\delta^r_s\,, \quad u^r{}_i u^s{}_{\hat{\jmath}}(\Gamma_{\hat{a}})^{ij} = 2u^{rs}{}_{\hat{a}}\,. \tag{3.5}$$

The superfield $W^+_a \equiv u^{+\hat{a}}W_{\hat{a}a}$ then satisfies the G-analyticity property

$$u^r{}_i D^i_\alpha u^{+\hat{a}}W_{\hat{a}a} \equiv D^r_\alpha W^+_a = 0\,. \tag{3.6}$$

One can obtain a linearised invariant from the action of the eight derivatives $D_{\alpha r} \equiv u_{ri}D^i_\alpha$'s on any homogeneous function of the $W^+_a$'s. After integrating over the harmonic variables with the normalisation $\int du = 1$ and using

$$\int du\, u^-{}_{\hat{a}_1}\ldots u^-{}_{\hat{a}_n} W^{+a_1}\cdots W^{+a_n} = \frac{6!\,n!}{(6+2n)(5+n)!}W^{a_1}{}_{(a_1}\cdots W^{a_n}{}_{a_n)'}\,, \tag{3.7}$$

with the projection $(\hat{a}_1\ldots\hat{a}_n)'$ on the traceless symmetric component (recall that $u^-{}_{\hat{a}}u^{-\hat{a}} = 0$), one gets [5]

$$\frac{(6+2n)(5+n)!}{6!\,n!}\int du\, u^-{}_{\hat{a}_1}\ldots u^-{}_{\hat{a}_n}[D^8]\frac{1}{(n+4)!}c_{a_1\ldots a_{n+4}}W^{+a_1}\ldots W^{+a_{n+4}}$$

$$= \frac{1}{n!}c_{a_1\ldots a_n abcd}W^{a_1}{}_{(\hat{a}_1}W^{a_2}{}_{\hat{a}_2}\ldots W^{a_n}{}_{\hat{a}_n)'}\mathcal{L}^{(0)abcd}$$

$$+ \frac{1}{(n-1)!}c_{a_1\ldots a_n abcd}W^{a_2}{}_{(\hat{a}_2}W^{a_3}{}_{\hat{a}_3}\ldots W^{a_n}{}_{\hat{a}_n}\mathcal{L}^{(0)a_1 abcd}_{\hat{a}_1)'} + \ldots$$

$$+ \frac{1}{(n-4)!}c_{a_1\ldots a_n abcd}W^{a_5}{}_{(\hat{a}_5}W^{a_6}{}_{\hat{a}_6}\ldots W^{a_n}{}_{\hat{a}_n}\mathcal{L}^{(0)a_1 a_2 a_3 a_4 abcd}_{\hat{a}_1\hat{a}_2\hat{a}_3\hat{a}_4)'} + \partial(\ldots)\,, \tag{3.9}$$

---

[5]In particular for a single vector multiplet

$$[D^8]\frac{1}{(n+4)!}(W^+)^{n+4} = \frac{1}{n!}(W^+)^n\big(2\partial_\mu W_{rs}\partial_\nu W^{rs}\partial^\mu W_{tu}\partial^\nu W^{tu} - \partial_\mu W_{rs}\partial^\mu W^{rs}\partial_\nu W_{tu}\partial^\nu W^{tu}\big)$$

$$- \frac{8}{(n+1)!}(W^+)^{n+1}\partial_\mu W_{rs}\partial_\nu W^{rs}\partial^\mu\partial^\nu W^- + \frac{8}{(n+2)!}(W^+)^{n+2}\partial_\mu\partial_\nu W^-\partial^\mu\partial^\nu W^- + \ldots\,. \tag{3.8}$$

where the $\mathcal{L}_n^{n+4}$ are symmetric tensors consisting of a homogeneous polynomial of order $4+n$ in $\partial_\mu W^{a\hat{a}}$, $\chi_{\alpha\hat{\imath}a}$ and $\partial_\mu\chi_{\alpha\hat{\imath}a}$, i.e.

$$\mathcal{L}^{(0)abcd} = 2\partial_\mu W^{(a}{}_{\hat{a}}\partial^\mu W^b{}_{\hat{b}}\partial_\nu W^{c|\hat{a}}\partial^\nu W^{d)\hat{b}} - \partial_\mu W^{(a}{}_{\hat{a}}\partial^\mu W^{b|\hat{a}}\partial_\nu W^c{}_{\hat{b}}\partial^\nu W^{d)\hat{b}} + \dots$$

$$\mathcal{L}^{(0)abcde}_{\hat{a}} \sim 4\times\chi^2(\partial W)^3 + 5\times\chi^3\partial\chi\partial W$$

$$\mathcal{L}^{(0)a_1 a_2 abcd}_{\hat{a}_1\hat{a}_2} \sim 6\times\chi^4(\partial W)^2 + 2\times\chi^5\partial\chi$$

$$\mathcal{L}^{(0)a_1 a_2 a_3 abcd}_{\hat{a}_1\hat{a}_2\hat{a}_3} \sim \chi^6\partial W$$

$$\mathcal{L}^{(0)a_1 a_2 a_3 a_4 abcd}_{\hat{a}_1\hat{a}_2\hat{a}_3\hat{a}_4} \sim \chi^8\,, \tag{3.10}$$

where we only wrote the bosonic part of the first polynomial, and only indicated the number of independent structures for the others, such that $\chi^8$ is for example the unique Lorentz singlet in the irreducible representation of $O(8)$ with four symmetrised indices without trace and eight symmetrised $O(r-8)$ indices. A total derivative has been extracted in (3.9) in order to remove all second derivative terms $\partial_\mu\partial_\nu W^{a\hat{a}}$.

At the non-linear level, derivatives of the scalar fields only appear through the pull-back of the right-invariant form $P_{a\hat{b}}$ defined from the Maurer–Cartan form

$$\mathrm{d}g\, g^{-1} = \begin{pmatrix} \mathrm{d}p_{La}{}^I \eta_{IJ} p_{Lb}{}^J & -\mathrm{d}p_{La}{}^I \eta_{IJ} p_{R\hat{b}}{}^J \\ \mathrm{d}p_{R\hat{a}}{}^I \eta_{IJ} p_{Lb}{}^J & -\mathrm{d}p_{R\hat{a}}{}^I \eta_{IJ} p_{R\hat{b}}{}^J \end{pmatrix} \equiv \begin{pmatrix} -\omega_{ab} & P_{a\hat{b}} \\ P_{b\hat{a}} & -\omega_{\hat{a}\hat{b}} \end{pmatrix}, \tag{3.11}$$

where $\eta_{IJ}$ is the $O(r-8,8)$ metric and $p_{L,a}{}^I$, $p_{R,\hat{b}}{}^I$ are the left and right projections parametrised by the Grassmaniann $G_{r-8,8}$. The right-invariant metric on $G_{r-8,8}$ is defined as $G_{\mu\nu} = 2P_{\mu a\hat{b}} P_\nu^{a\hat{b}}$ and the covariant derivative in tangent frame acts on a symmetric tensor as

$$\mathcal{D}_{a\hat{b}} A_{a_1\dots a_m,\hat{b}_1\dots\hat{b}_n} \equiv P_{\mu a\hat{b}} G^{\mu\nu}(\partial_\nu A_{a_1\dots a_m,\hat{b}_1\dots\hat{b}_n} + m\omega_{\nu(a_1}{}^c A_{a_2\dots a_m)c,\hat{b}_1\dots\hat{b}_n} + n\omega_{\nu(\hat{b}_1}{}^{\hat{c}} A_{a_1\dots a_m,|\hat{b}_2\dots\hat{b}_n)\hat{c}})\,. \tag{3.12}$$

The supersymmetry invariant associated to a tensor $F_{abcd}$ on the Grassmanian defines a Lagrange density $\mathcal{L}$ that decomposes naturally as

$$\mathcal{L} = F_{a_1 a_2 a_3 a_4}\mathcal{L}^{a_1 a_2 a_3 a_4} + \mathcal{D}_{(a_1}{}^{\hat{a}} F_{a_2 a_3 a_4 a_5)}\mathcal{L}^{a_1\dots a_5}{}_{\hat{a}} + \mathcal{D}_{(a_1}{}^{\hat{a}_1}\mathcal{D}_{a_2}{}^{\hat{a}_2} F_{a_3 a_4 a_5 a_6)}\mathcal{L}^{a_1\dots a_6}{}_{\hat{a}_1\hat{a}_2}$$
$$+ \mathcal{D}_{(a_1}{}^{\hat{a}_1}\mathcal{D}_{a_2}{}^{\hat{a}_2}\mathcal{D}_{a_3}{}^{\hat{a}_3} F_{a_4 a_5 a_6 a_7)}\mathcal{L}^{a_1\dots a_7}{}_{\hat{a}_1\hat{a}_2\hat{a}_3}$$
$$+ \mathcal{D}_{(a_1}{}^{\hat{a}_1}\mathcal{D}_{a_2}{}^{\hat{a}_2}\mathcal{D}_{a_3}{}^{\hat{a}_3}\mathcal{D}_{a_4}{}^{\hat{a}_4} F_{a_5\dots a_8)}\mathcal{L}^{a_1\dots a_8}{}_{\hat{a}_1\dots\hat{a}_4}\,, \tag{3.13}$$

where the $\mathcal{L}^{n+4}{}_n$ are $O(r-8,8)$ invariant polynomial functions of the following covariant fields:

$$P_{\mu a\hat{b}} = \partial_\mu\phi^\mu P_{\mu a\hat{b}}, \quad \chi_{\alpha\hat{\imath}a}, \quad \mathcal{D}_\mu\chi_{\alpha\hat{\imath}a} = \nabla_\mu\chi_{\alpha\hat{\imath}a} + \partial_\mu\phi^\mu\left(\omega_{\mu a}{}^b\chi_{\alpha\hat{\imath}a} + \frac{1}{4}\omega_{\mu\hat{a}\hat{b}}(\Gamma^{\hat{a}\hat{b}})_{\hat{\imath}}{}^{\hat{\jmath}}\chi_{\alpha\hat{\jmath}a}\right), \tag{3.14}$$

and the dreibeins and the gravitini fields. Because non-linear invariants define a linear invariant by truncation to lowest order in the fields (3.14), the covariant densities $\mathcal{L}^{4+n}_n$ reduce at lowest order to homogeneous polynomials of degree $n+4$ in the covariant fields (3.14) that coincide with the linearised polynomials $\mathcal{L}^{(0)n+4}_n$, in particular

$$\mathcal{L}^{abcd} = \sqrt{-g}\left(2P_\mu^{(a}{}_{\hat{a}}P^{\mu b}{}_{\hat{b}}P_\nu^{c|\hat{a}}P^{\nu d)\hat{b}} - P_\mu^{(a}{}_{\hat{a}}P^{\mu b|\hat{a}}P_\nu^c{}_{\hat{b}}P^{\nu d)\hat{b}} + \dots\right). \tag{3.15}$$

The important conclusion to draw from the linearised analysis is that the $O(r-8,8)$ right-invariants tensors $\mathcal{L}_n^{n+4}$ appearing in the ansatz (3.13) are symmetric in both sets of indices and traceless in

the $O(8)$ indices. Checking the supersymmetry invariance (modulo a total derivative) of $\mathcal{L}$ in this basis, one finds that there is no term to cancel the supersymmetry variation

$$\delta F_{abcd} = \left(\bar{\epsilon}_i(\Gamma^{\hat{f}})^{ij}\chi^e_j\right)\mathcal{D}^{e\hat{f}}F_{abcd} \tag{3.16}$$

of the tensor $F_{abcd}$ and of its derivative when open $O(r-8)$ indices are antisymmetrized, hence the tensor $F_{abcd}$ must satisfy the constraints

$$\mathcal{D}_{[a}{}^{[\hat{a}}\mathcal{D}_{b]}{}^{\hat{b}]}F_{cdef} = 0\,, \qquad \mathcal{D}_{[e}{}^{\hat{a}}F_{a]bcd} = 0\,. \tag{3.17}$$

Similarly, because the $\mathcal{L}^{n+4}_n$ are traceless in the $O(8)$ indices, the $O(8)$ singlet component of $\delta(\mathcal{D}F)\mathcal{L}^5_1$ can only be cancelled by terms coming from $F\delta\mathcal{L}^4$, i.e.

$$F_{abcd}\delta\mathcal{L}^{abcd} + \frac{1}{8}\mathcal{D}_e{}^{\hat{a}}\mathcal{D}_{f\hat{a}}F_{abcd}(\bar{\epsilon}\,\Gamma^{\hat{c}}\chi^e)\mathcal{L}^{abcdf}_{\hat{c}} \sim 0 \tag{3.18}$$

modulo terms arising from the supercovariantisation,[6] so that the covariant components must satisfy

$$\delta\mathcal{L}^{abcd} + \frac{5b_1}{8}(\bar{\epsilon}\,\Gamma^{\hat{c}}\chi_e)\mathcal{L}^{abcde}_{\hat{c}} + \frac{5b_2}{8}(\bar{\epsilon}\,\Gamma^{\hat{c}}\chi^{(a})\mathcal{L}^{bcd)e}_{\hat{c}}{}_e = \nabla_\mu(\dots) \tag{3.19}$$

and the tensor $F_{abcd}$ an equation of the form

$$\mathcal{D}_e{}^{\hat{a}}\mathcal{D}_{f\hat{a}}F_{abcd} = 5b_1\delta_{e(f}F_{abcd)} + 5b_2\,\delta_{(fa}F_{bcd)e}\,, \tag{3.20}$$

for some numerical constants $b_1, b_2$ which are fixed by consistency. In particular the integrability condition on the component antisymmetric in $e$ and $f$ implies $b_2 = 2b_1 + 4$.

Before determining the constants $b_i$, it is convenient to generalize $F_{abcd}$ to a completely symmetric tensor $F^{(p,q)}_{abcd}$ on a general Grassmanian $G_{p,q}$, which would arise by considering a superfield in $D = 10-q$ dimensions with $3 \leq q \leq 6$, with harmonics parametrizing similarly the Grassmannian $G_{q-2,2}$ [40]. The corresponding invariant takes the form $\mathcal{L} = F^{(p,q)}_{abcd}\mathcal{L}^{abcd} + \dots$ with

$$\begin{aligned}
\mathcal{L}^{abcd} = \sqrt{-g}\Big(&F^{(a}_{\mu\nu}F^{b|\nu\sigma}F^b_{\sigma\rho}F^{d)\rho\mu} - \frac{1}{4}F^{(a}_{\mu\nu}F^{b|\mu\nu}F^b_{\sigma\rho}F^{d)\sigma\rho}\\
&+ (4F^{(a}_{\mu\sigma}F^{b|\sigma}_\nu - \eta_{\mu\nu}F^{(a}_{\sigma\rho}F^{b|\sigma\rho})P^{\mu|c}{}_{\hat{a}}P^{\nu|d)\hat{a}}\\
&+ 2P^{(a}_\mu{}_{\hat{a}}P^{\mu\,b}{}_{\hat{b}}P^{c|\hat{a}}_\nu P^{\nu d)\hat{b}} - P^{(a}_\mu{}_{\hat{a}}P^{\mu\,b|\hat{a}}P^c_{\nu\hat{b}}P^{\nu d)\hat{b}} + \dots\Big)
\end{aligned} \tag{3.21}$$

where $F_{abcd}$ is subject to the constraints (3.17) and

$$\mathcal{D}_e{}^{\hat{a}}\mathcal{D}_{f\hat{a}}F^{(p,q)}_{abcd} = b_1\,\delta_{ef}F^{(p,q)}_{abcd} + 2b_2\delta_{f(a}F^{(p,q)}_{bcd)e} + (2b_2-q)\delta_{e(a}F^{(p,q)}_{bcd)f} + 3b_3\,\delta_{(ab}F^{(p,q)}_{cd)ef}\,. \tag{3.22}$$

with coefficients $b_1, b_2, b_3$ a priori depending on $p, q$.

A first integrability condition for (3.22) is obtained through

$$\begin{aligned}
0 = \mathcal{D}_e{}^{\hat{a}}(\mathcal{D}_{f\hat{a}}F^{(p,q)}_{abcd} - \mathcal{D}_{(a|\hat{a}}F^{(p,q)}_{bcd)f}) = &\left(b_1 - \frac{2b_2-q}{4}\right)(\delta_{ef}F^{(p,q)}_{abcd} - \delta_{e(a}F^{(p,q)}_{bcd)f})\\
&+ \frac{3}{2}(b_2-b_3)(\delta_{f(a}F^{(p,q)}_{bcd)e} - \delta_{(ab}F^{(p,q)}_{cd)ef})\,, \tag{3.23}
\end{aligned}$$

---

[6]The same construction in superspace implies that the lift of $\mathcal{L}$ in superspace is d-closed [38], such that $d_\omega\mathcal{L}^{abcd} = \frac{15}{16}P^{\hat{c}}{}_e \wedge \mathcal{L}^{abcde}_{\hat{c}} - \frac{5}{8}P^{\hat{c}(a} \wedge \mathcal{L}^{bcd)e}_{\hat{c}}{}_e$, in agreement with equation (3.19). Therefore, the terms associated to the variation of the gravitini that we disregard here do not spoil the argument [39].

which implies $b_1 = \frac{b_2 - q}{4}$ and $b_3 = b_2$, consistently with (3.20). Similarly, considering

$$
\begin{aligned}
\mathcal{D}_g{}^{\hat{a}}\big(\mathcal{D}_e{}^{\hat{b}}\mathcal{D}_{f\hat{b}}F_{abcd}^{(p,q)}\big) - \mathcal{D}_f{}^{\hat{a}}\big(\mathcal{D}_e{}^{\hat{b}}\mathcal{D}_{g\hat{b}}F_{abcd}^{(p,q)}\big) &= 2b_1\delta_{e[f}\mathcal{D}_{g]}{}^{\hat{a}}F_{abcd}^{(p,q)} + 2b_2\delta_{a)[f}\mathcal{D}_{g]}{}^{\hat{a}}F_{e(bcd}^{(p,q)} \\
&= [\mathcal{D}_g{}^{\hat{a}},\mathcal{D}_e{}^{\hat{b}}]\mathcal{D}_{f\hat{b}}F_{abcd}^{(p,q)} - [\mathcal{D}_f{}^{\hat{a}},\mathcal{D}_e{}^{\hat{b}}]\mathcal{D}_{g\hat{b}}F_{abcd}^{(p,q)} \\
&\quad + \mathcal{D}_e{}^{\hat{b}}[\mathcal{D}_{[g}{}^{\hat{a}},\mathcal{D}_{f]\hat{b}}]F_{abcd}^{(p,q)} \\
&= \frac{2-q}{2}\delta_{e[f}\mathcal{D}_{g]}{}^{\hat{a}}F_{abcd}^{(p,q)} + 2\delta_{a)[f}\mathcal{D}_{g]}{}^{\hat{a}}F_{e(bcd}^{(p,q)} , \quad (3.24)
\end{aligned}
$$

and therefore $b_1 = \frac{2-q}{4}$ and $b_2 = 1$ and so $b_3 = 1$ so that

$$
\mathcal{D}_e{}^{\hat{a}}\mathcal{D}_{f\hat{a}}F_{abcd}^{(p,q)} = 5\frac{2-q}{4}\delta_{e(f}F_{abcd)}^{(p,q)} + 5\delta_{(fa}F_{bcd)e}^{(p,q)} . \tag{3.25}
$$

Taking traces of this equation one can show that the entire tensor is determined by its trace component $F_{\text{tr}}^{(p,q)} \equiv F_{ab}^{(p,q)ab}$ through

$$
F_{abcd}^{(p,q)} = \frac{1}{(8+p-q)(6+p-q)}\Big(2\mathcal{D}_{(a}{}^{\hat{e}}\mathcal{D}_{b|\hat{e}}\mathcal{D}_c{}^{\hat{f}}\mathcal{D}_{d)\hat{f}} + (2q-7)\delta_{(ab}\mathcal{D}_c{}^{\hat{e}}\mathcal{D}_{d)\hat{e}} + \tfrac{3(q-2)(q-4)}{8}\delta_{(ab}\delta_{cd)}\Big)F_{\text{tr}}^{(p,q)} . \tag{3.26}
$$

The function $F_{\text{tr}}^{(p,q)}$ is an eigenmode of the Laplacian $\Delta_{G_{p,q}} \equiv 2\mathcal{D}_{a\hat{b}}\mathcal{D}^{a\hat{b}}$ on $G_{p,q}$, and satisfies

$$
\Delta_{G_{p,q}}F_{\text{tr}}^{(p,q)} = -\frac{1}{2}(p+4)(q-6)F_{\text{tr}}^{(p,q)} , \qquad \mathcal{D}_{[a}{}^{[\hat{a}}\mathcal{D}_{b]}{}^{\hat{b}]}F_{\text{tr}}^{(p,q)} = 0 . \tag{3.27}
$$

It is worth noting, however, that Eq. (3.25) for the tensor defined by (3.26) is an additional constraint on the function $F_{\text{tr}}$, which does not follow by integrability from the two equations (3.27).

Finally, let us note that the discussion so far only applies to the local Wilsonian effective action. As we shall see in the next subsection, the Ward identity satisfied by the renormalized coupling $\hat{F}_{abcd}$ is corrected in four dimensions (for $q = 6$) because of the 1-loop divergence of the supergravity amplitude [41], leading to the source term in (2.23).

## 3.2 The modular integral solves the Ward identities

In this subsection we shall prove that the modular integral (2.27) is a solution of the supersymmetric Ward identities (2.23). More generally, we shall show that the modular integral

$$
F_{abcd}^{(p,q)}(\Phi) = \text{R.N.} \int_{\Gamma_0(N)\backslash\mathcal{H}} \frac{d\tau_1 d\tau_2}{\tau_2^2} \frac{\Gamma_{\Lambda_{p,q}}[P_{abcd}]}{\Delta_k(\tau)} , \tag{3.28}
$$

where $\Delta_k(\tau)$ is the cusp form (2.4) of weight $k$ under $\Gamma_0(N)$, $\Lambda_{p,q}$ is a level $N$ even lattice of signature $(p,q)$ with $\frac{p-q}{2} + 4 = k$, and $P$ is the quartic polynomial (2.26), satisfies the constraints (3.17) and (3.22). Moreover, its trace $\delta^{ab}\delta^{cd}F_{abcd}^{(p,q)}(\Phi)$ is given by

$$
F_{\text{tr}}^{(p,q)}(\Phi) = \text{R.N.} \int_{\Gamma_0(N)\backslash\mathcal{H}} \frac{d\tau_1 d\tau_2}{\tau_2^2}\Gamma_{\Lambda_{p,q}} \cdot D_{-k+2}D_{-k}\frac{1}{\Delta_k(\tau)} . \tag{3.29}
$$

Before going into the proof however, it will be useful to spell out the regularization prescription which we use to define these otherwise divergent modular integrals. We follow the procedure

developed in [42, 43, 44], whereby the integral is first carried out on the truncated fundamental domain $\mathcal{F}_{N,\Lambda} = \mathcal{F}_N \cap \{\tau_2 < \Lambda\} \cap \{\frac{\tau_2}{N|\tau|^2} > \Lambda\}$, where $\mathcal{F}_N$ is the standard fundamental domain for $\Gamma_0(N)\backslash\mathcal{H}$, invariant under the Fricke involution $\tau \mapsto -1/(N\tau)$, and then the limit $\Lambda \to \infty$ is taken after subtracting any divergent term in $\Lambda$. In the case of the integral (3.28), the divergent term originates from the contribution of the vector $Q = 0$ in $\Gamma_{\Lambda_{p,q}}[P_{abcd}]$, so the regularized integral is defined for $q \neq 6$ by

$$F_{abcd}^{(p,q)}(\Phi) = \lim_{\Lambda \to \infty} \left[ \int_{\mathcal{F}_{N,\Lambda}} \frac{d\tau_1 d\tau_2}{\tau_2^2} \frac{\Gamma_{\Lambda_{p,q}}[P_{abcd}]}{\Delta_k(\tau)} - \frac{3\alpha_k}{16\pi^2} \frac{\Lambda^{\frac{q-6}{2}}}{\frac{q-6}{2}} \delta_{(ab}\delta_{cd)} \right], \qquad (3.30)$$

where $\alpha_k = (1+v)k = (1+v)c_k(0)$ for CHL orbifolds with $N > 1$, and $\alpha_{12} = (1+v)\frac{c(0)}{2}$ in the maximal rank case.[7] For $q < 6$, no subtraction is necessary, as long as the integral is carried out first along $\tau_1 \in [-\frac{1}{2}, \frac{1}{2}]$ in the region $\tau \to \infty$. For $q = 6$, the integral is logarithmically divergent, and the regularized integral is defined instead by

$$\widehat{F}_{abcd}^{(p,6)}(\Phi) = \lim_{\Lambda \to \infty} \left[ \int_{\mathcal{F}_{N,\Lambda}} \frac{d\tau_1 d\tau_2}{\tau_2^2} \frac{\Gamma_{\Lambda_{p,6}}[P_{abcd}]}{\Delta_k(\tau)} - \frac{3\alpha_k}{16\pi^2} \log\Lambda \, \delta_{(ab}\delta_{cd)} \right]. \qquad (3.31)$$

The logarithmic divergence at $q = 6$ is consistent with the expected divergence in the one-loop scattering amplitude of four gauge bosons in $D = 4$ supergravity [41]. Equivalently, following [45] one may consider the modular integral

$$F_{abcd}^{(p,q)}(\Phi, \epsilon) = \int_{SL(2,\mathbb{Z})\backslash\mathcal{H}} \frac{d\tau_1 d\tau_2}{\tau_2^{2-\epsilon}} \sum_{\gamma \in \Gamma_0(N)\backslash SL(2,\mathbb{Z})} \frac{\Gamma_{\Lambda_{p,q}}[P_{abcd}]}{\Delta_k(\tau)}\bigg|_\gamma , \qquad (3.32)$$

which converges for $\mathrm{Re}(\epsilon) < \frac{6-q}{2}$, and defines the renormalized integral as the constant term in the Laurent expansion at $\epsilon = 0$ of the analytical continuation of $F_{abcd}^{(p,q)}(\Phi, \epsilon)$. The result will then differ from (3.31) by an irrelevant additive constant. In what follows, we shall often abuse notation and omit the hat in $\widehat{F}_{abcd}^{(p,q)}$ when stating properties valid for arbitrary $q$. It is also important to note that while the regularized integral (3.30) or (3.31) is finite at generic points on $G_{p,q}$, it diverges on a real codimension-$q$ loci of $G_{p,q}$, where $Q_{R,\hat{a}} = 0$ for a vector $Q \in \Lambda_{p,q}$ with $Q^2 = 2$, or for a vector $Q \in \Lambda_{p,q}^*$ with $Q^2 = 2/N$ (see (E.12)).

In order to establish that $F_{abcd}^{(p,q)}$ satisfies the constraints (3.22), we shall first establish differential equations for a general class of lattice partition functions

$$\Gamma_{\Lambda_{p,q}}[P] = \tau_2^{\frac{q}{2}} \sum_{Q \in \Lambda_{p,q}} P(Q) e^{i\pi Q_L^2 \tau - i\pi Q_R^2 \bar{\tau}} , \qquad (3.33)$$

where the polynomial $P(Q)$ is obtained by acting with the operator $\tau_2^n e^{-\frac{\Delta}{8\pi\tau_2}}$, with

$$\Delta \equiv \sum_a \left(\frac{\partial}{\partial Q_L^a}\right)^2 + \sum_{\hat{a}} \left(\frac{\partial}{\partial Q_R^{\hat{a}}}\right)^2 , \qquad (3.34)$$

---

[7] $v$ is defined below (4.29) and depends on the volume of $\Lambda_{p,q}^*/\Lambda$. $v = 1$ for the perturbative Narain lattice, and $v = 1/N$ for the non-perturbative Narain lattice (2.21).

on a homogeneous polynomial of bidegree $(m,n)$ in $(Q_L, Q_R)$, respectively. As shown in [45], $\Gamma_{\Lambda_{p,q}}[P]$ satisfies

$$\Gamma_{\Lambda_{p,q}}[P](-1/\tau) = \frac{(-\mathrm{i})^{\frac{p-q}{2}} \tau^{\frac{p-q}{2}+m-n}}{\sqrt{|\Lambda_{p,q}^*/\Lambda_{p,q}|}} \Gamma_{\Lambda_{p,q}^*}[P](\tau) \,, \tag{3.35}$$

which implies that it transforms as a modular form of weight $\frac{p-q}{2} + m - n$ under $\Gamma_0(N)$. More specifically, we shall consider $\Gamma_{\Lambda_{p,q}}\big[P_{a_1\dots a_m, \hat b_1 \dots \hat b_n}\big]$ with

$$P_{a_1\dots a_m, \hat b_1 \dots \hat b_n} = \tau_2^n e^{-\frac{\Delta}{8\pi\tau_2}} \big(Q_{L,a_1} \dots Q_{L,a_m} Q_{R,\hat b_1} \dots Q_{R,\hat b_n}\big) \,. \tag{3.36}$$

The quartic polynomial $P_{abcd}$ defined in (2.26) arises in the case $(m,n) = (4,0)$, so that $\Gamma_{\Lambda_{p,q}}[P_{abcd}]$ is a modular form of weight $\frac{p-q}{2} + 4 = k$, ensuring the modular invariance of the integrands in (3.28) and (3.29). Upon contracting the indices, it is easy to check that $\delta^{ab}\delta^{cd}\Gamma_{\Lambda_{p,q}}[P_{abcd}] = D_{k-2}D_{k-4}\Gamma_{\Lambda_{p,q}}[1]$, so the claim that (3.29) gives the trace of (3.28) follows by integration by parts.

To obtain the differential equations satisfied by (3.28), we shall act with the covariant derivative $\mathcal{D}_{a\hat b}$, defined in (3.11) and (3.12). As mentioned below (2.13), $p_{L,a}{}^I$, $p_{R,\hat b}{}^I$ are the left and right orthogonal projectors on the Grassmaniann $G_{p,q} = O(p,q)/[O(p)\times O(q)]$. Using the derivative rules

$$\mathcal{D}_{a\hat b}\, p_{L,c}{}^I = \frac{1}{2}\delta_{ac}\, p_{R,\hat b}{}^I \,, \qquad \mathcal{D}_{a\hat b}\, p_{R,\hat c}{}^I = \frac{1}{2}\delta_{\hat b\hat c}\, p_{L,a}{}^I \,, \tag{3.37}$$

one can effectively define the action of the covariant derivative on a function that only depends on $Q_L$ and $Q_R$ as

$$\mathcal{D}_{a\hat b} = \frac{1}{2}\big(Q_{L,a}\partial_{\hat b} + Q_{R,\hat b}\partial_a\big) \,, \tag{3.38}$$

where $\partial_a = \frac{\partial}{\partial Q_L^a}$, $\partial_{\hat b} = \frac{\partial}{\partial Q_R^{\hat b}}$. Acting with $\mathcal{D}_{e\hat g}$ on (3.33) we get

$$\mathcal{D}_{e\hat g}\, \Gamma_{\Lambda_{p,q}}\big[P_{a_1\dots a_m, \hat b_1 \dots \hat b_n}\big] = \Gamma_{\Lambda_{p,q}}\Big[\big(\mathcal{D}_{e\hat g} - 2\pi\tau_2 Q_{L,e}Q_{R,\hat g}\big)P_{a_1\dots a_m \hat b_1 \dots \hat b_n}\Big] \,. \tag{3.39}$$

Using (3.38), one computes the commutation relations

$$[\Delta, \mathcal{D}_{e\hat g}] = 2\partial_e\partial_{\hat g} \,, \qquad\qquad [\Delta, Q_{L,e}Q_{R,\hat g}] = 4\mathcal{D}_{e\hat g} \,, \tag{3.40}$$

$$[\Delta, Q_{L,e}Q_{L,f}] = 2\delta_{ef} + 4Q_{L,(e}\partial_{f)} \,, \qquad [\Delta, Q_{L,(e}\partial_{f)}] = 2\partial_e\partial_f \,. \tag{3.41}$$

Using them along with the Baker-Campbell-Hausdorff formula

$$e^{\frac{\Delta}{8\pi\tau_2}}\, \mathcal{O}\, e^{-\frac{\Delta}{8\pi\tau_2}} = \mathcal{O} + \frac{1}{8\pi\tau_2}[\Delta, \mathcal{O}] + \frac{1}{2!}\frac{1}{(8\pi\tau_2)^2}[\Delta, [\Delta, \mathcal{O}]] + \dots \,, \tag{3.42}$$

one easily obtains

$$\mathcal{D}_{e\hat g}\, \Gamma_{\Lambda_{p,q}}\big[P_{a_1\dots a_m, \hat b_1 \dots \hat b_n}\big] = -2\pi\tau_2\, \Gamma_{\Lambda_{p,q}}\Big[e^{-\frac{\Delta}{8\pi\tau_2}}\Big(Q_{L,e}Q_{R,\hat g} - \frac{1}{(4\pi\tau_2)^2}\partial_e\partial_{\hat g}\Big)e^{\frac{\Delta}{8\pi\tau_2}}P_{a_1\dots a_m, \hat b_1 \dots \hat b_n}\Big] \,. \tag{3.43}$$

Note that the similarity transformation is such that the operator acts on the simple monomial in $Q_{a_1}\dots Q_{a_m}Q_{\hat b_1}\dots Q_{\hat b_n}$ according to (3.36), such that it directly follows from (3.43) that

$$\mathcal{D}_{e\hat g}\, \Gamma_{\Lambda_{p,q}}\big[P_{a_1\dots a_m, \hat b_1 \dots \hat b_n}\big] = \Gamma_{\Lambda_{p,q}}\Big[-2\pi\, P_{ea_1\dots a_m, \hat g\hat b_1 \dots \hat b_n} + \frac{mn}{8\pi}\delta_{e(a_1}P_{a_2\dots a_m),(\hat b_2\dots \hat b_n}\delta_{\hat b_1)\hat g}\Big] \,. \tag{3.44}$$

Upon antisymmetrizing in $(e, a_1)$, we get

$$\mathcal{D}_{[e}{}^{\hat{g}}\Gamma_{\Lambda_{p,q}}\big[P_{a_1]...a_m,\hat{b}_1...\hat{b}_n}\big] = \frac{1}{8\pi^2\tau_2^2}\,\Gamma_{\Lambda_{p,q}}\Big[e^{-\frac{\Delta}{8\pi\tau_2}}\,\partial_{[e}\partial^{\hat{g}}\,e^{\frac{\Delta}{8\pi\tau_2}}\,P_{a_1]...a_m,\hat{b}_1...\hat{b}_n}\Big]. \tag{3.45}$$

which vanishes when $n = 0$ since $e^{\frac{\Delta}{8\pi\tau_2}}P_{a_1...a_m}$ does not depend on $Q_R$. Acting a second time with $\mathcal{D}_{a\hat{b}}$ and antisymmetrizing, we get

$$\mathcal{D}_{[e}{}^{[\hat{e}}\mathcal{D}_{f]}{}^{\hat{f}]}\Gamma_{\Lambda_{p,q}}\big[P_{a_1...a_m,\hat{b}_1...\hat{b}_n}\big] = -2\Gamma_{\Lambda_{p,q}}\Big[e^{-\frac{\Delta}{8\pi\tau_2}}\,Q_{L,[e}Q_R{}^{[\hat{e}}\partial_{f]}\partial^{\hat{f}]}\,e^{\frac{\Delta}{8\pi\tau_2}}\,P_{a_1...a_m,\hat{b}_1...\hat{b}_n}\Big], \tag{3.46}$$

which similarly vanishes when $n = 0$. Setting $m = 4$, we conclude that the modular integral (3.28) satisfies

$$\mathcal{D}_{[e}{}^{\hat{a}}F_{a]bcd} = 0\,, \qquad \mathcal{D}_{[e}{}^{[\hat{e}}\mathcal{D}_{f]}{}^{\hat{f}]}F_{abcd} = 0\,, \tag{3.47}$$

which therefore establishes the last two equations in (2.23). Note that these two equations do not rely on any particular property of the function $1/\Delta_k$.

Now, the first equation of (2.23) arises from applying the quadratic operator $\mathcal{D}^2_{ef} \equiv \mathcal{D}_{(e}{}^{\hat{g}}\mathcal{D}_{f)\hat{g}}$ on the partition function with polynomial insertion,

$$4\mathcal{D}^2_{ef}\Gamma_{\Lambda_{p,q}}\big[P_{a_1...a_m,\hat{b}_1...\hat{b}_n}\big] = \Gamma_{\Lambda_{p,q}}\Big[\Big(4\mathcal{D}^2_{ef} - 8\pi\tau_2 Q_{L,(e}Q_R{}^{\hat{g}}\mathcal{D}_{f)\hat{g}} \\ + 16\pi^2\tau_2^2\Big(Q_{L,e}Q_{L,f} - \frac{\delta_{ef}}{4\pi\tau_2}\Big)\Big(Q_R^2 - \frac{q}{4\pi\tau_2}\Big) - q\delta_{ef}\Big)P_{a_1...a_m\hat{b}_1...\hat{b}_n}\Big], \tag{3.48}$$

which gives, using (3.40) and (3.42)

$$4\mathcal{D}^2_{ef}\Gamma_{\Lambda_{p,q}}\big[P_{a_1...a_m,\hat{b}_1...\hat{b}_n}\big] = \Gamma_{\Lambda_{p,q}}\Big[e^{-\frac{\Delta}{8\pi\tau_2}}\Big(16\pi^2\tau_2^2 Q_R^2 Q_{L,e}Q_{L,f} + \frac{\partial_e\partial_f\partial_R^2}{16\pi^2\tau_2^2} \\ - Q_{L,(e}\partial_{f)}(2Q_R{}^{\hat{g}}\partial_{\hat{g}} + q) - \delta_{ef}(Q_R{}^{\hat{g}}\partial_{\hat{g}} + q)\Big)e^{\frac{\Delta}{8\pi\tau_2}}\,P_{a_1...a_m\hat{b}_1...\hat{b}_n}\Big] \tag{3.49}$$

The first term on the r.h.s. can be rewritten as the action of the Maass lowering operator $\bar{D}_w = -i\pi\tau_2^2\partial_{\bar{\tau}}$, mapping modular forms of weight $w$ to weight $w-2$. Indeed,

$$\bar{D}_w\Gamma_{\Lambda_{p,q}}\big[P_{efa_1...a_m,\hat{b}_1...\hat{b}_n}\big] = -\pi^2\tau_2^2\,\Gamma_{\Lambda_{p,q}}\Big[\Big(Q_R^2 - \frac{q+2n}{4\pi\tau_2}\Big)P_{efa_1...a_m,\hat{b}_1...\hat{b}_n}\Big] \\ + \frac{1}{16}\Gamma_{\Lambda_{p,q}}\Big[\Delta\,P_{efa_1...a_m,\hat{b}_1...\hat{b}_n}\Big] \tag{3.50} \\ = \Gamma_{\Lambda_{p,q}}\Big[e^{-\frac{\Delta}{8\pi\tau_2}}\Big(\tfrac{1}{16}\partial_L^2 - (\pi\tau_2 Q_R)^2\Big)e^{\frac{\Delta}{8\pi\tau_2}}\,P_{efa_1...a_m,\hat{b}_1...\hat{b}_n}\Big].$$

where in the second line, we used the fact that $\Delta$ commutes with $e^{-\frac{\Delta}{8\pi\tau_2}}$. The r.h.s. of (3.49) can thus be written as

$$4\mathcal{D}^2_{ef}\Gamma_{\Lambda_{p,q}}\big[P_{a_1...a_m,\hat{b}_1...\hat{b}_n}\big] = (2 - (q+n))\delta_{ef}\Gamma_{\Lambda_{p,q}}\big[P_{a_1...a_m,\hat{b}_1...\hat{b}_n}\big] \\ + m(4 - (q+2n))\delta_{|e)(a_1}\Gamma_{\Lambda_{p,q}}\big[P_{a_2...a_m)(f|,\hat{b}_1...\hat{b}_n}\big] + m(m-1)\delta_{(a_1a_2}\Gamma_{\Lambda_{p,q}}\big[P_{a_3...a_m)ef,\hat{b}_1...\hat{b}_n}\big] \tag{3.51} \\ + \tfrac{m(m-1)n(n-1)}{16\pi^2}\delta_{e(a_1}\delta_{|f|a_2}\Gamma_{\Lambda_{p,q}}\big[P_{a_3...a_m),(\hat{b}_1...\hat{b}_{n-2}}\big]\delta_{\hat{b}_{n-1}\hat{b}_n)} - 16\bar{D}_w\Gamma_{\Lambda_{p,q}}\big[P_{efa_1...a_m,\hat{b}_1...\hat{b}_n}\big],$$

where only the last term remains to be computed explicitly. Specializing to the case of main interest, we obtain

$$\Box_{ef}\cdot\Gamma_{\Lambda_{p,q}}\big[P_{abcd}\big] = -4\bar{D}_w\,\Gamma_{\Lambda_{p,q}}\big[P_{abcdef}\big] \tag{3.52}$$

where, for any tensor $F_{abcd}$, we denote

$$\Box_{ef} \cdot F_{abcd} \equiv \mathcal{D}^2_{ef} F_{abcd} + \frac{(q-2)}{4} \delta_{ef} F_{abcd} + (q-4)\delta_{(e|(a} F_{bcd)|f)} - 3\delta_{(ab} F_{cd)ef} . \tag{3.53}$$

We can now integrate both sides of (3.52) times $1/\Delta_k$ on the truncated fundamental domain $\mathcal{F}_{N,\Lambda}$, leading to

$$\Box_{ef} \int_{\mathcal{F}_{N,\Lambda}} \frac{\mathrm{d}\tau_1 \mathrm{d}\tau_2}{\tau_2^2} \frac{\Gamma_{\Lambda_{p,q}}[P_{abcd}]}{\Delta_k} = -4 \int_{\mathcal{F}_{N,\Lambda}} \frac{\mathrm{d}\tau_1 \mathrm{d}\tau_2}{\tau_2^2} \frac{1}{\Delta_k} \bar{D}_{k+2} \Gamma_{\Lambda_{p,q}}\left[P_{abcdef}\right] . \tag{3.54}$$

The r.h.s. is a boundary term, because $\bar{D}_{-k}(1/\Delta_k) = 0$ by holomorphicity. To compute the boundary term we use Stokes' theorem in the form

$$\int_{\partial \mathcal{F}_{N,\Lambda}} f \, g \, \mathrm{d}\tau = \int_{\mathcal{F}_\Lambda} \mathrm{d}(f \, g \, \mathrm{d}\tau) = \frac{2}{\pi} \int_{\mathcal{F}_{N,\Lambda}} \frac{\mathrm{d}\tau_1 \mathrm{d}\tau_2}{\tau_2^2} (\bar{D}_w f \, g + f \, \bar{D}_{w'} \, g), \tag{3.55}$$

where $f$ and $g$ are any modular forms of weight $w$ and $w' = -w + 2$ and $2\mathrm{d}\tau_1 \mathrm{d}\tau_2 = \mathrm{i}\mathrm{d}\tau \wedge \mathrm{d}\bar{\tau}$. By modular invariance, the boundary term reduces to an integral along the segment $\{1/2 \le \tau_1 < 1/2, \tau_2 = \Lambda\}$ , and its image under the Fricke involution (for $N > 1$). The latter can be mapped to the former upon using (3.35). At generic points on the Grassmannian $G_{p,q}$, the contributions of non-zero vectors in $\Lambda_{p,q}$ and $\Lambda^*_{p,q}$ are exponentially suppressed, leaving only the contribution of $Q = 0$:

$$\Box_{ef} \int_{\mathcal{F}_{N,\Lambda}} \frac{\mathrm{d}\tau_1 \mathrm{d}\tau_2}{\tau_2^2} \frac{\Gamma_{\Lambda_{p,q}}[P_{abcd}]}{\Delta_k} = \Lambda^{\frac{q-6}{2}} \frac{15 \, \alpha_k}{2(4\pi)^2} \delta_{(ab} \delta_{cd} \delta_{ef)}, \tag{3.56}$$

where $\alpha_k = (1 + v)k$ was introduced below (3.30). Acting with the same operator $\mathcal{D}^2_{ef}$ on the subtraction in (3.30), we see that the term proportional to $\Lambda^{(q-6)/2}$ cancels, except for $q = 6$ where the substraction in (3.31) leaves a finite remainder. Thus, we find, as claimed earlier, that the modular integral (3.28) is annihilated by the second-order differential operator $\Box_{ef}$ defined in (3.53), up to a constant source term present when $q = 6$,

$$\Box_{ef} F^{(p,q)}_{abcd} = \frac{15 \, \alpha_k}{2(4\pi)^2} \delta_{(ab} \delta_{cd} \delta_{ef)} \delta_{q,6} . \tag{3.57}$$

In B, as a consistency check we show that this equation is verified by each Fourier mode in the degeneration limit $O(p,q) \to O(p-1,q-1)$.

# 4 Weak coupling expansion of exact $(\nabla\Phi)^4$ couplings

In this section, we study the expansion of the proposal (2.27) in the limit where the heterotic string coupling $g_3$ goes to zero, and show that it reproduces the known tree-level and one-loop amplitudes, along with an infinite series of NS5-brane, Kaluza–Klein monopole and H-monopole instanton corrections. We start by analyzing the expansion of the tensorial modular integral defin-

ing the coupling and its trace

$$F_{abcd}^{(p,q)}(\Phi) = \text{R.N.} \int_{\Gamma_0(N)\backslash\mathcal{H}} \frac{d\tau_1 d\tau_2}{\tau_2^2} \frac{\Gamma_{\Lambda_{p,q}}[P_{abcd}]}{\Delta_k(\tau)} \,, \tag{4.1a}$$

$$F_{\text{tr}}^{(p,q)}(\Phi) = \text{R.N.} \int_{\Gamma_0(N)\backslash\mathcal{H}} \frac{d\tau_1 d\tau_2}{\tau_2^2} \Gamma_{\Lambda_{p,q}} D_{-k+2} D_{-k} \frac{1}{\Delta_k(\tau)} \,, \tag{4.1b}$$

for a level $N$ even lattice $\Lambda_{p,q}$ of arbitrary signature $(p,q)$, in the limit near the cusp where $O(p,q)$ is broken to $O(1,1) \times O(p-1,q-1)$, so that the moduli space decomposes into

$$G_{p,q} \to \mathbb{R}^+ \times G_{p-1,q-1} \ltimes \mathbb{R}^{p+q-2} \,. \tag{4.2}$$

For simplicity, we first discuss the maximal rank case $N = 1$, $p - q = 16$, where the integrand is invariant under the full modular group, before dealing with the case of $N$ prime, where the integrand is invariant under the Hecke congruence subgroup $\Gamma_0(N)$. The reader uninterested by the details of the derivation may skip to §4.3, where we specialize to the values $(p,q) = (r-4,8)$ relevant for the $(\nabla\Phi)^4$ couplings in $D = 3$ and interpret the various contributions as perturbative and non-perturbative effects in heterotic string theory compactified on $T^7$. In §4.5 we discuss the case $(p,q) = (r-7,5)$ relevant for $H^4$ couplings in type IIB string theory compactified on $K3$.

## 4.1 $O(p,q) \to O(p-1,q-1)$ for even self-dual lattices

We first consider the case where the lattice $\Lambda_{p,q}$ is even self-dual and factorizes in the limit (4.2) as

$$\Lambda_{p,q} \to \Lambda_{p-1,q-1} \oplus I\!I_{1,1} \,. \tag{4.3}$$

We shall denote by $R$ the coordinate on $\mathbb{R}^+$ and by $a^I$, $I = 2 \dots p+q-1$ the coordinates on $\mathbb{R}^{p+q-2}$. $R$ parametrizes a one-parameter subgroup $e^{RH_0}$ in $O(p,q)$, such that the action of the non-compact Cartan generator $H_0$ on the Lie algebra $\mathfrak{so}_{p,q}$ decomposes into

$$\mathfrak{so}_{p,q} \simeq (\mathbf{p+q-2})^{(-2)} \oplus (\mathfrak{gl}_1 \oplus \mathfrak{so}_{p-1,q-1})^{(0)} \oplus (\mathbf{p+q-2})^{(2)} \,. \tag{4.4}$$

while the coordinates $a^I$ parametrize the unipotent subgroup obtained by exponentiating the grade 2 component in this decomposition. A generic charge vector $Q_{\mathcal{I}} \in \Lambda_{p,q} \simeq \mathbf{1}^{(-2)} \oplus (\mathbf{p+q-2})^{(0)} \oplus \mathbf{1}^{(2)}$ decomposes into $Q_{\mathcal{I}} = (m, \widetilde{Q}_I, n)$ where $(m,n) \in I\!I_{1,1} = \mathbb{Z}^2$ and $\widetilde{Q}_I \in \Lambda_{p-1,q-1}$, such that $Q^2 = -2mn + \widetilde{Q}^2$. The orthogonal projectors defined by $Q_L \equiv p_L^{\mathcal{I}} Q_{\mathcal{I}}$ and $Q_R \equiv p_R^{\mathcal{I}} Q_{\mathcal{I}}$ decompose according to

$$\begin{aligned}
p_{L,1}^{\mathcal{I}} Q_{\mathcal{I}} &= \frac{1}{R\sqrt{2}} \left( m + a \cdot \widetilde{Q} + \frac{1}{2} a \cdot a \, n \right) - \frac{R}{\sqrt{2}} n, \\
p_{L,a}^{\mathcal{I}} Q_{\mathcal{I}} &= \widetilde{p}_{L,a}^I (\widetilde{Q}_I + n a_I), \\
p_{R,1}^{\mathcal{I}} Q_{\mathcal{I}} &= \frac{1}{R\sqrt{2}} \left( m + a \cdot \widetilde{Q} + \frac{1}{2} a \cdot a \, n \right) + \frac{R}{\sqrt{2}} n, \\
p_{R,\hat{a}}^{\mathcal{I}} Q_{\mathcal{I}} &= \widetilde{p}_{R,\hat{a}}^I (\widetilde{Q}_I + n a_I),
\end{aligned} \tag{4.5}$$

where $\widetilde{p}_{L,a}^I, \widetilde{p}_{R,\hat{a}}^I$ ($\alpha = 2 \dots d+16$, $\hat{\alpha} = 2 \dots d$) are orthogonal projectors in $G_{p-1,q-1}$ satisfying $\widetilde{Q}^2 = \widetilde{Q}_L^2 - \widetilde{Q}_R^2$. In the following we shall denote $|Q_R| \equiv \sqrt{\widetilde{Q}_R^2}$.

To study the behavior of (4.1) in the limit $R \gg 1$,[8] it is useful to perform a Poisson resummation on $m$. For a lattice partition function $\Gamma_{\Lambda_{p,q}}$ with no insertion, as in the scalar integral (4.1b), this gives

$$\Gamma_{\Lambda_{p,q}} = R\,\tau_2^{\frac{q-1}{2}} \sum_{(m,n)\in\mathbb{Z}^2} \sum_{\widetilde{Q}\in\Lambda_{p-1,q-1}} e^{-\frac{\pi R^2|n\tau+m|^2}{\tau_2}} e^{2\pi i m(a\cdot\widetilde{Q}+\frac{1}{2}a\cdot a\,n)} q^{\frac{1}{2}\widetilde{Q}_L^2} q^{\frac{1}{2}\widetilde{Q}_R^2}. \qquad (4.6)$$

In the case of a lattice sum with momentum insertion, as in the tensor integral $F^{(p,q)}_{abcd}$ (4.1a), we must distinguish whether the indices $abcd$ lie along the direction 1 or along the directions $\alpha$. Denoting by $h$ the number of indices along direction 1, the previous result generalizes to

$$\Gamma_{\Lambda_{p,q}}\Big[e^{-\frac{\Delta}{8\pi\tau_2}}\big[(Q_{L,1})^h Q_{L,\alpha_1}\dots Q_{L,\alpha_{4-h}}\big]\Big] = R \sum_{(m,n)\in\mathbb{Z}^2} \left(\frac{R(n\bar\tau+m)}{i\tau_2\sqrt{2}}\right)^h e^{-\frac{\pi R^2|n\tau+m|^2}{\tau_2}}$$

$$\times\ \Gamma_{\Lambda_{p-1,p-1}+na}\Big[e^{-\frac{\Delta}{8\pi\tau_2}}\big[\widetilde{Q}_{L,\alpha_1}\dots\widetilde{Q}_{L,\alpha_{4-h}}\big]e^{2\pi i m(\widetilde{Q}-\frac{1}{2}a\,n)\cdot a}\Big]. \quad (4.7)$$

In this representation, modular invariance is manifest, since a transformation $\tau \mapsto \frac{a\tau+b}{c\tau+d}$ can be compensated by a linear transformation $(n,m) \mapsto (n,m)\left(\begin{smallmatrix} a & b \\ c & d \end{smallmatrix}\right)$, under which the second line of (4.7) transforms with weight $12-h$. As a relevant example for what follows, consider the case $(n,m) = k(c,d)$, $k = \gcd(m,n)$, then using an transformation $\left(\begin{smallmatrix} a & b \\ c & d \end{smallmatrix}\right) \in SL(2,\mathbb{Z})$

$$\sum_{\widetilde{Q}\in\Lambda_{p-1,q-1}+kc\,a} e^{-\frac{\Delta}{8\pi\tau_2}}\big[\widetilde{Q}_{L,\alpha_1}\dots\widetilde{Q}_{L,\alpha_{4-h}}\big]e^{2\pi i kd(\widetilde{Q}-\frac{1}{2}a\,kc)\cdot a} q^{\frac{1}{2}\widetilde{Q}_L^2} \bar q^{\frac{1}{2}\widetilde{Q}_R^2} =$$

$$(c\tau+d)^{12-h} \sum_{\widetilde{Q}\in\Lambda_{p-1,q-1}} e^{-\frac{\Delta}{8\pi\tau_2}}\big[\widetilde{Q}_{L,\alpha_1}\dots\widetilde{Q}_{L,\alpha_{4-h}}\big]e^{2\pi i k\widetilde{Q}\cdot a} q^{\frac{1}{2}\widetilde{Q}_L^2} \bar q^{\frac{1}{2}\widetilde{Q}_R^2}. \quad (4.8)$$

We can therefore compute the integral using the orbit method [46, 47, 48], namely decompose the sum over $(m,n)$ into various orbits under $SL(2,\mathbb{Z})$, and for each orbit $\mathcal{O}$, retain the contribution of a particular element $\varsigma \in \mathcal{O}$ at the expense of extending the integration domain $\mathcal{F}_1 = SL(2,\mathbb{Z})\backslash\mathcal{H}$ to $\Gamma_\varsigma\backslash\mathcal{H}$, where $\Gamma_\varsigma$ is the stabilizer of $\varsigma$ in $SL(2,\mathbb{Z})$,[9] by using the identity

$$\bigcup_{\gamma\in\Gamma_\varsigma\backslash SL(2,\mathbb{Z})} \gamma\cdot\mathcal{F}_1 = \Gamma_\varsigma\backslash\mathcal{H}. \qquad (4.9)$$

The coset representative $\varsigma \in \mathcal{O}$, albeit arbitrary, is usually chosen so as to make the unfolded domain $\Gamma_\varsigma\backslash\mathcal{H}$ as simple as possible. In the present case, there are two types of orbits:

**The trivial orbit** $(n,m) = (0,0)$ produces, up to a factor of $R$, the integrals (4.1) for the lattice $\Lambda_{p-1,q-1}$, provided none of the indices $abcd$ lie along the direction 1,

$$F^{(p,q),0}_{\alpha\beta\gamma\delta} = R\,F^{(p-1,q-1)}_{\alpha\beta\gamma\delta}, \qquad F^{(p,q),0}_{\mathrm{tr}} = R\,F^{(p-1,q-1)}_{\mathrm{tr}}, \qquad (4.10)$$

while it vanishes otherwise (*i.e.* when $h > 0$).

---

[8]Since $1/\Delta$ grows as $e^{\frac{2\pi}{\tau_2}}$ at $\tau_2 \to 0$, the following treatment which relies on exchanging the sum and the integral for unfolding is justified for $R^2 > 2$.

[9]This unfolding procedure requires particular care since the integrand is not of rapid decay near the cusp. We suppress these details here, and refer to [42, 45, 49, 43, 50, 44] for rigorous treatments.

**The rank-one orbit** corresponds to terms with $(n, m) \neq (0, 0)$. Setting $(n, m) = k(c, d)$, with $\gcd(c, d) = 1$ and $k \neq 0$, the doublet $(c, d)$ can always be rotated by an element of $SL(2, \mathbb{Z})$ into $(0, 1)$, whose stabilizer inside $SL(2, \mathbb{Z})$ is $\Gamma_\infty = \{\left(\begin{smallmatrix} 1 & n \\ 0 & 1 \end{smallmatrix}\right), n \in \mathbb{Z}\}$. Thus, doublets $(c, d)$ with $\gcd(c, d) = 1$ are in one-to-one correspondence with elements of $\Gamma_\infty \backslash SL(2, \mathbb{Z})$. For each $k$, one can therefore unfold the integration domain $SL(2, \mathbb{Z}) \backslash \mathcal{H}$ to $\mathcal{S} = \Gamma_\infty \backslash \mathcal{H} = \mathbb{R}^+_{\tau_2} \times (\mathbb{R}/\mathbb{Z})_{\tau_1}$, the unit width strip, provided one keeps only the term $(c, d) = (0, 1)$ in the sum. The resulting contribution to the tensor integral (4.1a) are

$$F^{(p,q),1}_{\alpha\beta\gamma\delta} = R \int_{\mathbb{R}^+} \frac{d\tau_2}{\tau_2^2} \int_{\mathbb{R}/\mathbb{Z}} d\tau_1 \sum_{k \neq 0} e^{-\pi R^2 k^2/\tau_2} \frac{\Gamma_{\Lambda_{p-1, q-1}} \left[ \tilde{P}_{\alpha\beta\gamma\delta} e^{2\pi i k a^I \tilde{Q}_I} \right]}{\Delta},$$

$$F^{(p,q),1}_{11\gamma\delta} = R \int_{\mathbb{R}^+} \frac{d\tau_2}{\tau_2^2} \int_{\mathbb{R}/\mathbb{Z}} d\tau_1 \sum_{k \neq 0} \left( \frac{Rk}{i\tau_2\sqrt{2}} \right)^2 e^{-\pi R^2 k^2/\tau_2} \frac{\Gamma_{\Lambda_{p-1, q-1}} \left[ \tilde{P}_{\alpha\beta} e^{2\pi i k a^I \tilde{Q}_I} \right]}{\Delta}, \qquad (4.11)$$

$$F^{(p,q),1}_{1111} = R \int_{\mathbb{R}^+} \frac{d\tau_2}{\tau_2^2} \int_{\mathbb{R}/\mathbb{Z}} d\tau_1 \sum_{k \neq 0} \left( \frac{Rk}{i\tau_2\sqrt{2}} \right)^4 e^{-\pi R^2 k^2/\tau_2} \frac{\Gamma_{\Lambda_{p-1, q-1}} \left[ e^{2\pi i k a^I \tilde{Q}_I} \right]}{\Delta},$$

where

$$\tilde{P}_{\alpha_1 \ldots \alpha_{4-h}} = e^{-\frac{\Delta}{8\pi\tau_2}} \left[ \tilde{Q}_{L, \alpha_1} \ldots \tilde{Q}_{L, \alpha_{4-h}} \right], \qquad (4.12)$$

while the contribution to its trace is

$$F^{(p,q),1}_{\text{tr}} = R \int_{\mathbb{R}^+} \frac{d\tau_2}{\tau_2^2} \int_{\mathbb{R}/\mathbb{Z}} d\tau_1 \sum_{k \neq 0} e^{-\pi R^2 k^2/\tau_2} \Gamma_{\Lambda_{p-1, q-1}} \left[ e^{2\pi i k a^I \tilde{Q}_I} \right] D^2 \left( \frac{1}{\Delta} \right). \qquad (4.13)$$

The integral over $\mathcal{S}$ can be computed by inserting the Fourier expansion

$$\frac{1}{\Delta} = \sum_{\substack{m \in \mathbb{Z} \\ m \geq -1}} c(m) q^m, \qquad D^2 \frac{1}{\Delta} = a_2 c(0) + \sum_{\substack{m \in \mathbb{Z} - \{0\} \\ m \geq -1}} \sum_{\ell=0}^{2} a_\ell m^{2-\ell} c(m) q^m \tau_2^{-\ell} \qquad (4.14)$$

where

$$a_0 = 4, \qquad a_1 = \frac{p - q + 6}{\pi}, \qquad a_2 = \frac{(p - q + 6)(p - q + 8)}{16\pi^2}. \qquad (4.15)$$

The integral over $\tau_1$ picks up the Fourier coefficient $c(m)$ with $m = -\frac{1}{2}\tilde{Q}^2$. The remaining integral over $\tau_2$ can be computed after expanding $\tilde{P}_{\alpha_1 \ldots \alpha_{4-h}} = \sum_{\ell=0}^{\lfloor \frac{4-h}{2} \rfloor} \tilde{P}^{(\ell)}_{\alpha_1 \ldots \alpha_{4-h}} \tau_2^{-\ell}$, where $\tilde{P}^{(\ell)}_{\alpha_1 \ldots \alpha_{4-h}}$ is a polynomial in $\tilde{Q}$ of degree $4 - h - 2\ell$, or zero when $2\ell > 4 - h$. Contributions with $\tilde{Q} = 0$ lead to power-like terms,

$$F^{(p,q),1,0}_{\alpha\beta\gamma\delta} = R^{q-6} \xi(q-6) \frac{3c(0)}{8\pi^2} \delta_{(\alpha\beta} \delta_{\gamma\delta)},$$

$$F^{(p,q),1,0}_{11\alpha\beta} = R^{q-6} \xi(q-6)(7-q) \frac{c(0)}{8\pi^2} \delta_{\alpha\beta}, \qquad (4.16)$$

$$F^{(p,q),1,0}_{1111} = R^{q-6} \xi(q-6)(7-q)(9-q) \frac{c(0)}{8\pi^2},$$

while the result vanishes for an odd number of indices along the direction 1, and for its trace

$$F^{(p,q),1,0}_{\text{tr}} = R^{q-6} \xi(q-6)(p-q+6)(p-q+8) \frac{c(0)}{8\pi^2}. \qquad (4.17)$$

Here we used $\tilde{P}^{(2)}_{abcd}(0) = \frac{3}{16\pi^2}\delta_{(ab}\delta_{cd)}$, $\tilde{P}^{(1)}_{ab}(0) = -\frac{1}{4\pi}\delta_{ab}$, and $\tilde{P}^{(0)} = 1$. Note that (4.17) and (4.16) have a simple pole at $q = 6$, which is subtracted by the regularization prescription mentioned below (3.32). For $q = 7$, the pole in (4.17), (4.16) cancels against the pole from the zero orbit contribution (4.10).

In contrast, non-zero vectors $\widetilde{Q}$ lead to exponentially suppressed contributions, which depend on the axions through a phase factor $e^{2\pi i k a \cdot \widetilde{Q}}$. After rescaling $\widetilde{Q} \mapsto Q/k$, we find that the Fourier coefficient with charge $Q \in \Lambda_{p-1,q-1} \setminus \{0\}$ is given by

$$F^{(p,q),1,Q}_{\alpha\beta\gamma\delta} = 4\,\bar{c}(Q)R^{\frac{q-1}{2}}\sum_{\ell=0}^{2}\frac{\tilde{P}^{(\ell)}_{\alpha\beta\gamma\delta}(Q)}{R^\ell}\frac{K_{\frac{q-3}{2}-\ell}\left(2\pi R\sqrt{2|Q_R|^2}\right)}{\sqrt{2|Q_R|^2}^{\frac{q-3}{2}-\ell}}$$

$$F^{(p,q),1,Q}_{1\alpha\beta\gamma} = 4\,\bar{c}(Q)R^{\frac{q-1}{2}}\sum_{\ell=0}^{1}\frac{\tilde{P}^{(\ell)}_{\alpha\beta\gamma}(Q)}{i\sqrt{2}R^\ell}\frac{K_{\frac{q-5}{2}-\ell}\left(2\pi R\sqrt{2|Q_R|^2}\right)}{\sqrt{2|Q_R|^2}^{\frac{q-5}{2}-\ell}} \qquad (4.18)$$

$$\vdots$$

$$F^{(p,q),1,Q}_{1111} = 4\,\bar{c}(Q)R^{\frac{q-1}{2}}\frac{\tilde{P}^{(0)}}{4}\frac{K_{\frac{q-11}{2}}\left(2\pi R\sqrt{2|Q_R|^2}\right)}{\sqrt{2|Q_R|^2}^{\frac{q-11}{2}}}$$

for the tensor integral, and

$$F^{(p,q),1,Q}_{\text{tr}} = 4\,\bar{c}(Q)R^{\frac{q-1}{2}}\sum_{\ell=0}^{2}\frac{a_\ell}{R^\ell}\left(-\frac{Q^2}{2}\right)^{2-\ell}\frac{K_{\frac{q-3}{2}-\ell}\left(2\pi R\sqrt{2|Q_R|^2}\right)}{\sqrt{2|Q_R|^2}^{\frac{q-3}{2}-\ell}} \qquad (4.19)$$

for its trace. In either case,

$$\bar{c}(Q) = \sum_{d|Q}c\left(-\frac{Q^2}{2d^2}\right)d^{q-7} . \qquad (4.20)$$

The physical interpretation of these results will be discussed in §4.3, after generalizing them to $\mathbb{Z}_N$ orbifolds.

## 4.2 Extension to $\mathbb{Z}_N$ CHL orbifolds

The degeneration limit (4.2) of the modular integrals (4.1) for $\mathbb{Z}_N$ CHL models with $N = 2,3,5,7$ can be treated similarly by adapting the orbit method to the case where the integrand is invariant under the Hecke congruence subgroup $\Gamma_0(N)$ [51, 52, 44]. In (4.1), $\Delta_k$ is the cusp form of weight $k = \frac{24}{N+1}$ defined in (2.4), and $\Gamma_{\Lambda_{p,q}}$ is the partition function for a lattice

$$\Lambda_{p,q} = \tilde{\Lambda}_{p-1,q-1} \oplus I\!I_{1,1}[N] , \qquad (4.21)$$

where $\tilde{\Lambda}_{p-1,q-1}$ is a level $N$ even lattice of signature $(p-1,q-1)$. The lattice $I\!I_{1,1}[N]$ is obtained from the usual unimodular lattice $I\!I_{1,1}$ by restricting the winding and momentum to be $(n,m) \in N\mathbb{Z} \oplus \mathbb{Z}$. After Poisson resummation on $m$, Eq. (4.6) and (4.7) continue to hold, except for the fact that $n$ is restricted to run over $N\mathbb{Z}$. The sum over $(n,m)$ can then be decomposed into orbits of $\Gamma_0(N)$:[10]

---

[10] Since $1/\Delta_k$ grows as $e^{\frac{2\pi}{N\tau_2}}$ at $\tau_2 \to 0$, the following treatment which relies on exchanging the sum and the integral for unfolding is justified for $NR^2 > 2$.

**Trivial orbit**    The term $(n,m) = (0,0)$ produces the same modular integral, up to a factor of $R$,

$$F^{(p,q),0}_{\alpha\beta\gamma\delta} = R \tilde{F}^{(p-1,q-1)}_{\alpha\beta\gamma\delta} \, , \quad F^{(p,q),0}_{\text{tr}} = R \tilde{F}^{(p-1,q-1)}_{\text{tr}} \, , \tag{4.22}$$

where $\tilde{F}^{(p-1,q-1)}_{\alpha\beta\gamma\delta}$, $\tilde{F}^{(p-1,q-1)}_{\text{tr}}$ are the integrals (4.1) for the lattice $\tilde{\Lambda}_{p-1,q-1}$ defined by (4.21).

**Rank-one orbits**    Terms with $(n,m) = k(c,d)$ with $k \neq 0$ and $\gcd(c,d) = 1$ fall into two different classes of orbits under $\Gamma_0(N)$:

- Doublets $k(c,d)$ such that $c = 0 \bmod N$ and $k \in \mathbb{Z}$ can be rotated by an element of $\Gamma_0(N)$ into $(0,1)$, whose stabilizer in $\Gamma_0(N)$ is $\Gamma_\infty = \{\begin{pmatrix} 1 & n \\ 0 & 1 \end{pmatrix}, n \in \mathbb{Z}\}$. For these elements, one can unfold the integration domain $\Gamma_0(N)\backslash\mathcal{H}$ into the unit width strip $\mathcal{S} = \Gamma_\infty\backslash\mathcal{H} = \mathbb{R}^+_{\tau_2} \times (\mathbb{R}/\mathbb{Z})_{\tau_1}$;

- Doublets $k(c,d)$ such that $c \neq 0 \bmod N$ and $k = 0 \bmod N$ can be rotated by an element of $\Gamma_0(N)$ into $(1,0)$, whose stabilizer in $\Gamma_0(N)$ is $S\,\Gamma_{\infty,N}\,S^{-1}$, where $\Gamma_{\infty,N} = \{\begin{pmatrix} 1 & n \\ 0 & 1 \end{pmatrix}, n \in N\mathbb{Z}\}$ and $S = \begin{pmatrix} 0 & -1 \\ 1 & 0 \end{pmatrix}$. One can unfold the integration domain $\Gamma_0(N)\backslash\mathcal{H}$ into $S\,\Gamma_{\infty,N}\,S^{-1}\backslash\mathcal{H}$, and change variable $\tau \to -1/\tau$ so as to reach $\mathcal{S}_N = \Gamma_{\infty,N}\backslash\mathcal{H} = \mathbb{R}^+_{\tau_2} \times (\mathbb{R}/N\mathbb{Z})_{\tau_1}$, the width-$N$ strip. Under this change of variable, the level-$N$ weight-$k$ cusp form transforms as $\Delta_k(-1/\tau) = N^{-\frac{k}{2}}(-i\tau)^k\Delta_k(\tau/N)$, while the partition function for the sublattice $\tilde{\Lambda}_{p-1,q-1}$ transforms as

$$\Gamma_{\tilde{\Lambda}_{p-1,q-1}}[P_{\alpha\beta\gamma\delta}](-1/\tau) = \tilde{v} N^{-\frac{k}{2}-1}(-i)^{\frac{p-q}{2}}\tau^k \Gamma_{\tilde{\Lambda}^*_{p-1,q-1}}[P_{\alpha\beta\gamma\delta}](\tau) \, , \tag{4.23}$$

where $\Gamma_{\tilde{\Lambda}^*_{p-1,q-1}}(\tau)$ denotes the sum over the dual lattice $\tilde{\Lambda}^*_{p-1,q-1}$, and $\tilde{v} N^{-\frac{k}{2}-1} = \left| \tilde{\Lambda}^*_{p-1,q-1}/\tilde{\Lambda}_{p-1,q-1} \right|^{-1/2}$. Note that $\tilde{v} = N^{1-\delta_{q,8}}$ for $q \leq 8$ in the cases of interest.

For the simplest component $F^{(p,q),1}_{\alpha\beta\gamma\delta}$, the sum of the two classes of orbits then reads

$$F^{(p,q),1}_{\alpha\beta\gamma\delta} = R \int_{\mathbb{R}^+} \frac{d\tau_2}{\tau_2^2} \int_{\mathbb{R}/\mathbb{Z}} d\tau_1 \frac{1}{\Delta_k(\tau)} \sum_{k \neq 0} e^{-\pi R^2 k^2/\tau_2} \Gamma_{\tilde{\Lambda}_{p-1,q-1}}\left[e^{2\pi i k a^I \tilde{Q}_I} P_{\alpha\beta\gamma\delta}\right]$$

$$+ R \int_{\mathbb{R}^+} \frac{d\tau_2}{\tau_2^2} \int_{\mathbb{R}/(N\mathbb{Z})} d\tau_1 \frac{\tilde{v}}{N} \frac{1}{\Delta_k(\tau/N)} \sum_{\substack{k \neq 0 \\ k = 0 \bmod N}} e^{-\pi R^2 k^2/\tau_2} \Gamma_{\tilde{\Lambda}^*_{p-1,q-1}}\left[e^{2\pi i k a^I \tilde{Q}_I} P_{\alpha\beta\gamma\delta}\right] \, . \tag{4.24}$$

The contributions from $\tilde{Q} = 0$ lead to power-like terms,

$$F^{(p,q)(1,0)}_{\alpha\beta\gamma\delta} = R^{q-6} \xi(q-6)\left(1 + \tilde{v} N^{q-7}\right)\frac{3c_k(0)}{8\pi^2}\delta_{(\alpha\beta}\delta_{\gamma\delta)} \, ,$$

$$F^{(1,0)}_{11\alpha\beta} = R^{q-6}\, \xi(q-6)(7-q)\left(1 + \tilde{v} N^{q-7}\right)\frac{c_k(0)}{8\pi^2}\delta_{\alpha\beta} \, , \tag{4.25}$$

$$F^{(1,0)}_{1111} = R^{q-6}\, \xi(q-6)(7-q)(9-q)\left(1 + \tilde{v} N^{q-7}\right)\frac{c_k(0)}{8\pi^2} \, ,$$

for the tensor integral and

$$F^{(p,q)(1,0)}_{\text{tr}} = R^{q-6}\xi(q-6)(p-q+6)(p-q+8)\left(1 + \tilde{v} N^{q-7}\right)\frac{c_k(0)}{8\pi^2} \tag{4.26}$$

for its trace, where $c_k(0) = k$ is the constant term in $1/\Delta_k$. As in (4.17) and (4.16), the pole at $q = 6$ is subtracted by the regularization prescription (3.30), while the pole at $q = 7$ cancels against the pole from the zero orbit contribution (4.22).

The terms with non-zero vector $\widetilde{Q}$ produce exponentially suppressed corrections of the same form as in the maximal rank case (4.18), but with a different summation measure, namely

$$\bar{c}_k(Q) = \sum_{\substack{d \geq 1, \\ Q/d \in \tilde{\Lambda}_{p-1,q-1}}} c_k\left(-\frac{Q^2}{2d^2}\right) d^{q-7} + \tilde{v} \sum_{\substack{d \geq 1, \\ Q/d \in N\tilde{\Lambda}^*_{p-1,q-1}}} c_k\left(-\frac{Q^2}{2Nd^2}\right)(Nd)^{q-7}, \qquad (4.27)$$

where the first term, arising from the first class of orbits, has support on $\tilde{\Lambda}_{p-1,q-1}$, and the second term, arising from the second class of orbits, has support on the sublattice $N\tilde{\Lambda}^*_{p-1,q-1} \subset \tilde{\Lambda}_{p-1,q-1}$. In the latter contribution, notice that one factor of $N$ in the numerator of the Fourier coefficient comes from the matching condition with $1/\Delta_k(\tau/N)$, and two factors of $N$ in its denominator come from all the divisors being originally multiples of $N$.

It will also be useful to consider a different degeneration limit of the type (4.2) where the lattice decomposes as

$$\Lambda_{p,q} = \Lambda_{p-1,q-1} \oplus I\!I_{1,1} , \qquad (4.28)$$

where $I\!I_{1,1}$ is the usual unimodular even lattice, with no restriction on the windings and momenta $(n, m)$, and $\Lambda_{p-1,q-1}$ is a level $N$ even lattice of signature $(p-1, q-1)$, not to be confused with the lattice $\tilde{\Lambda}_{p-1,q-1}$ above. The sum over $(n, m) \in \mathbb{Z} \oplus \mathbb{Z}$ can then be decomposed into orbits of $\Gamma_0(N)$. The trivial orbit is similar to (4.22), but now $F^{(p-1,q-1)}_{\alpha\beta\gamma\delta}$ and $F^{(p-1,q-1)}_{\text{tr}}$ are the modular integrals for the lattice $\Lambda_{p-1,q-1}$. For the rank-one orbit, the discussion goes as before, except that the second class of orbits $(m, n) = k(c, d)$ with $k = \gcd(m, n)$ and $c \neq 0 \bmod N$ has no restriction on $k$. For the simplest component $F^{(p,q),1}_{\alpha\beta\gamma\delta}$, the sum of the two classes of orbits then reads

$$F^{(p,q),1}_{\alpha\beta\gamma\delta} = R \int_{\mathbb{R}^+} \frac{d\tau_2}{\tau_2^2} \int_{\mathbb{R}/\mathbb{Z}} d\tau_1 \frac{1}{\Delta_k(\tau)} \sum_{k \neq 0} e^{-\pi R^2 k^2/\tau_2} \Gamma_{\Lambda_{p-1,q-1}}\left[e^{2\pi i k a^I \widetilde{Q}_I} P_{\alpha\beta\gamma\delta}\right]$$

$$+ R \int_{\mathbb{R}^+} \frac{d\tau_2}{\tau_2^2} \int_{\mathbb{R}/(N\mathbb{Z})} d\tau_1 \frac{1}{\Delta_k(\tau/N)} \frac{v}{N} \sum_{k \neq 0} e^{-\pi R^2 k^2/\tau_2} \Gamma_{\Lambda^*_{p-1,q-1}}\left[e^{2\pi i k a^I \widetilde{Q}_I} P_{\alpha\beta\gamma\delta}\right], \quad (4.29)$$

where $vN^{-\frac{k}{2}-1} = \left|\Lambda^*_{p-1,q-1}/\Lambda_{p-1,q-1}\right|^{-1/2}$ (which now simplifies to $v = N^{-\delta_{q,8}}$ for $q \leq 8$ in the cases of interest). The contributions from $\widetilde{Q} = 0$ lead to power-like terms,

$$F^{(p,q)(1,0)}_{\alpha\beta\gamma\delta} = R^{q-6}\xi(q-6)(1+v)\frac{3c_k(0)}{8\pi^2}\delta_{(\alpha\beta}\delta_{\gamma\delta)},$$

$$F^{(1,0)}_{11\alpha\beta} = R^{q-6}\xi(q-6)(7-q)(1+v)\frac{c_k(0)}{8\pi^2}\delta_{\alpha\beta}, \qquad (4.30)$$

$$F^{(1,0)}_{1111} = R^{q-6}\xi(q-6)(7-q)(9-q)(1+v)\frac{c_k(0)}{8\pi^2}$$

for the tensor integral and

$$F^{(p,q)(1,0)}_{\text{tr}} = R^{q-6}\xi(q-6)(p-q+6)(p-q+8)(1+v)\frac{c_k(0)}{8\pi^2} \qquad (4.31)$$

for its trace, where $c_k(0) = k$ is the constant term in $1/\Delta_k$.

The terms with non-zero vector $\widetilde{Q}$ produce exponentially suppressed corrections of the same form as in the maximal rank case (4.18), but with a different summation measure, namely

$$\bar{c}_k(Q) = \sum_{\substack{d \geq 1, \\ Q/d \in \Lambda_{p-1,q-1}}} c_k\left(-\frac{Q^2}{2d^2}\right) d^{q-7} + \upsilon \sum_{\substack{d \geq 1, \\ Q/d \in \Lambda^*_{p-1,q-1}}} c_k\left(-\frac{NQ^2}{2d^2}\right) d^{q-7}, \tag{4.32}$$

where the first term, arising from the first class of orbits, has support on $\Lambda_{p-1,q-1}$, and the second term, arising from the second class of orbits, has support on the dual lattice $\Lambda^*_{p-1,q-1}$. In the latter contribution, notice that one factor of $N$ in the numerator of the Fourier coefficient comes from the matching condition with $1/\Delta_k(\tau/N)$.

## 4.3 Perturbative limit of exact $(\nabla\Phi)^4$ couplings in $D = 3$

Specializing to $(p,q) = (2k, 8) = (r - 4, 8)$, and decomposing as $\Lambda_{2k,8} = \Lambda_{2k-1,7} \oplus I\!I_{1,1}[N]$, the limit (4.2) studied in this section corresponds to the expansion of the exact $(\nabla\Phi)^4$ couplings in $D = 3$ in the limit where the heterotic string coupling $g_3 = 1/\sqrt{R}$ becomes weak. To interpret the resulting contributions in the language of heterotic perturbation theory, one should remember that the U-duality function $F^{(2k,8)}_{abcd}(\Phi)$ is the coefficient of the $(\nabla\Phi)^4$ coupling in the low-energy action written in Einstein frame, such that the metric $\gamma_E$ is inert under U-duality,

$$S_3 = \int d^3x \sqrt{-\gamma_E} \left[ \mathcal{R}[\gamma_E] - (2\delta_{\hat{a}\hat{b}}\delta_{\hat{c}\hat{d}} - \delta_{\hat{a}\hat{c}}\delta_{\hat{b}\hat{d}}) F^{(2k,8)}_{abcd}(\Phi) \gamma_E^{\mu\rho} \gamma_E^{\nu\sigma} P^{a\hat{a}}_\mu P^{b\hat{b}}_\nu P^{c\hat{c}}_\rho P^{d\hat{d}}_\sigma \right] + \dots . \tag{4.33}$$

In terms of the string frame metric $\gamma = \gamma_E g_3^4$, one finds

$$S_3 = \int d^3x \sqrt{-\gamma} \left[ \frac{1}{g_3^2}\mathcal{R}[\gamma] - g_3^2 (2\delta_{\hat{a}\hat{b}}\delta_{\hat{c}\hat{d}} - \delta_{\hat{a}\hat{c}}\delta_{\hat{b}\hat{d}}) F^{(2k,8)}_{abcd}(\Phi) \gamma^{\mu\rho} \gamma^{\nu\sigma} P^{a\hat{a}}_\mu P^{b\hat{b}}_\nu P^{c\hat{c}}_\rho P^{d\hat{d}}_\sigma \right] + \dots . \tag{4.34}$$

Using $c_k(0) = k$ for CHL orbifolds with $N > 1$ or $c(0) = 2k$ in the maximal rank case, and $\xi(2) = \frac{\pi}{6}$, the results from §4.1 and §4.2 read

$$g_3^2 F^{(2k,8)}_{abcd} = \frac{3}{2\pi g_3^2}\delta_{(ab}\delta_{cd)} + F^{(2k-1,7)}_{abcd} + \sum_{Q\in\Lambda_{2k-1,7}}' \bar{c}_k(Q) e^{-\frac{2\pi\sqrt{2}|Q_R|}{g_3^2} + 2\pi i a \cdot Q} P^{(*)}_{abcd}, \tag{4.35}$$

where we omit the detailed form of exponentially suppressed corrections, and the summation measure is read off from (4.27)

$$\bar{c}_k(Q) = \sum_{\substack{d \geq 1, \\ Q/d \in \Lambda_{2k-1,7}}} d\, c_k\left(-\frac{Q^2}{2d^2}\right) + \sum_{\substack{d \geq 1, \\ Q/d \in N\Lambda^*_{2k-1,7}}} N\, d\, c_k\left(-\frac{Q^2}{2Nd^2}\right), \tag{4.36}$$

The first two terms in (4.35), originating from the zero orbit and rank-one orbit, respectively, should match the tree-level and one-loop contributions, respectively. Indeed, the dimensional reduction of the tree-level $\mathcal{R}^2 + (\text{Tr} F^2)^2$ coupling in ten-dimensional heterotic string theory [53, 54] leads to a tree-level $(\nabla\Phi)^4$ coupling in $D = 3$, with a coefficient which is by construction independent of $N$. A more detailed analysis of the ten-dimensional origin of this term will be given in §5.3.1. The second term in (4.35) of course matches the one-loop contribution (2.24) by

construction. The remaining non-perturbative terms can be interpreted as heterotic NS5-brane, KK5-brane and H-monopoles wrapped on any possible $T^6$ inside $T^7$ [9]. More precisely, NS5-brane and KK5-brane charges correspond to momentum and winding charges in the hyperbolic part $II_{1,1}[N] \oplus II_{k-2,k-2}$ of $\Lambda_m \oplus II_{1,1}$, while H-monopoles correspond to charges in the gauge lattice $\Lambda_{k,8-k}$ (for the heterotic string compactification on $T^7$, these sublattices must be replaced by $II_{7,7}$ and $E_8 \oplus E_8$ or $D_{16}$, respectively). Note that [9] studied these corrections on a special locus in moduli space, corresponding to $T^4/\mathbb{Z}_2$ realization of K3 surfaces on the type II side, and did not keep track of all gauge charges, which resulted in a different summation measure.

## 4.4 Decompactification limit of one-loop $F^4$ couplings

For general $(p,q) = (d + 2k - 8, d) = (d + r - 12, d)$ with $q \leq 7$, the modular integral (4.1a) is interpreted as the one-loop $F^4$ amplitude in a heterotic CHL orbifold compactified down to dimension $D = 10 - d$. The decomposition (4.21) corresponds to the case (a) where the radius $R$ of a circle in $T^d$ orthogonal to the $\mathbb{Z}_N$ orbifold action becomes large, while the limit (4.28) corresponds to the case (b) where the radius $R$ of the circle in $T^d$ singled out by the $\mathbb{Z}_N$ orbifold action becomes large in string units.

The power-like terms contributions in $R$ come in part from the trivial orbit, and from the zero-charge contribution to the rank-one orbit:

$$
\begin{aligned}
a): \quad & F^{(p,q)}_{\alpha\beta\gamma\delta} = R F^{(p-1,q-1)}_{\alpha\beta\gamma\delta} + R^{q-6}\xi(q-6)\frac{3(2k)}{8\pi^2}\delta_{(\alpha\beta}\delta_{\gamma\delta)} + \dots \\
b): \quad & F^{(p,q)}_{\alpha\beta\gamma\delta} = R \tilde{F}^{(p-1,q-1)}_{\alpha\beta\gamma\delta} + R^{q-6}\xi(q-6)\frac{3k(1+N^{q-6})}{8\pi^2}\delta_{(\alpha\beta}\delta_{\gamma\delta)} + \dots
\end{aligned}
\tag{4.37}
$$

The first term reproduces, up to a volume factor of $R$, the one-loop $F^4$ amplitude in $D + 1$ dimensions (4.10), either in the same CHL model (case a), or in the full heterotic string compactification (case b). Indeed, in the latter case, the partition function $\Gamma_{\Lambda_{p-1,q-1}}$ factorizes into $\Gamma_{II_{d+k-9,d+k-9}} \times \Gamma_{\Lambda_{k,8-k}}$. The fundamental domain $\Gamma_0(N)\backslash\mathcal{H}$ can be extended to $SL(2,\mathbb{Z})\backslash\mathcal{H}$, at the expense of replacing $\Gamma_{\Lambda_{k,8-k}}/\Delta_k$ by the sum over its images under $\Gamma_0(N)\backslash SL(2,\mathbb{Z}) = \{1, S, TS, \dots, T^{N-1}S\}$. As explained in §A, this sum reproduces $\Gamma_{\Lambda_{d+15,d-1}}/\Delta$, the partition function for the maximal rank theory in dimension $D + 1$.

The second term, originating from the zero-charge contribution to the rank-one orbit, can instead be understood as the limit $s \to 0$ of an infinite tower of terms of the schematic form $\sum_{m\neq 0}(\frac{m^2}{R^2} - s)^{3-\frac{d}{2}}F^4$ in the low-energy effective action, where $s$ is a Mandelstam variable, arising from threshold contributions of Kaluza–Klein excitations of the massless supergravity states in dimension $D + 1$. In the limit $R \to \infty$, this infinite series along with the term $m = 0$ from the non-local part of the action in dimension $D$ sums up to the contribution of massless supergravity states to the non-local part of the action in dimension $D + 1$. The pole at $q = 6$ in the second term of (4.37) originates from the logarithmic infrared divergence in the local part of the string effective action in dimension $D = 4$, and matches the expected ultraviolet divergence in 4-dimensional supergravity. The apparent pole at $q = 7$ cancels against a pole in the first term, due to the same logarithmic divergence. Indeed, the $1/\epsilon$ pole of the full amplitude $F^{(p,6)}_{abcd}(\Phi, \epsilon)$ can be extracted from its Laurent expansion at $\epsilon = 0$, namely

$$
F^{(p,6)}_{abcd}(\Phi, \epsilon) = -\frac{3(2k)}{16\pi^2\epsilon}\delta_{(ab}\delta_{cd)} + \mathcal{O}(1)
\tag{4.38}
$$

In addition, massive perturbative BPS states with non-vanishing charge $Q \in \Lambda_{d+2k-9,d-1}$ in dimension $D + 1$ and mass $\mathcal{M}(Q)$ lead to exponentially suppressed terms of order $e^{-2\pi R\mathcal{M}(Q)}$, weighted by the helicity supertrace $\Omega_4(Q)$, as expected on general grounds.

### 4.5 Perturbative limit of exact $H^4$ couplings in type IIB on $K3$

Here we briefly consider the case $q = 5$, $N = 1$, corresponding to type IIB string theory compactified on $K3$. In Einstein frame, the low energy effective action takes the form

$$S_6 = \int \mathrm{d}^6x \sqrt{-\gamma_E} \left[ \mathcal{R}[\gamma_E] - F^{21,5}_{abcd}(\Phi) H^a_{\mu\nu\kappa} H^{b\ \kappa}_{\rho\sigma} H^{c\mu\nu\lambda} H^{d\rho\sigma}_{\ \ \ \lambda} \right] + \dots ,  \tag{4.39}$$

where the three-form $H^\alpha$ with $\alpha \neq 1$ are the self-dual field-strengths of the reduction of the RR two-form, four-form and six-form on the self-dual part of the homology lattice $H^{\mathrm{even}}(K3) = E_8 \oplus E_8 \oplus II_{4,4}$ , while $H^1$ is the self-dual component of the NS-NS two-form field-strength. We shall restrict for simplicity to the components $\alpha, \beta, \gamma, \delta \neq 1$. In terms of the string frame metric $\gamma = g_s\gamma_E$ and setting $\mathcal{H}^a = g_s H^a$ (since Ramond-Ramond field are normalized as $H \sim 1/g_s$ in type II perturbation theory), we get

$$S_6 = \int \mathrm{d}^6x \sqrt{-\gamma} \left[ \frac{1}{g_s^2} \mathcal{R}[\gamma] - \frac{1}{g_s} F^{21,5}_{\alpha\beta\gamma\delta}(\Phi) H^\alpha_{\mu\nu\kappa} H^{\beta\ \kappa}_{\rho\sigma} H^{\gamma\mu\nu\lambda} H^{\delta\rho\sigma}_{\ \ \ \lambda} \right] + \dots  \tag{4.40}$$

Identifying $R = 1/g_s$, the large radius expansion of $F^{21,5}_{\alpha\beta\gamma\delta}$ becomes, schematically,

$$\frac{1}{g_s} F^{21,5}_{\alpha\beta\gamma\delta} = \frac{1}{g_s^2} F^{20,4}_{\alpha\beta\gamma\delta}(\Phi) + \frac{3}{2\pi} \delta_{(\alpha\beta} \delta_{\gamma\delta)} + \sideset{}{'}\sum_{Q \in \Lambda_{20,4}} \bar{c}(Q) e^{-\frac{2\pi\sqrt{2}|Q_R|^2}{g_s} - 2\pi \mathrm{i} a\cdot Q} P^*_{\alpha\beta\gamma\delta} .  \tag{4.41}$$

The first term proportional to $F^{20,4}_{abcd}$ is now recognized as a tree-level correction in type II on $K3$, the second term is a one-loop correction which to our knowledge has not been computed independently yet, and the remaining terms originate from D3, D1, D(-1) branes wrapped on $K3$ [55]. It is worth noting that decompactification limits of the form $O(2k, 8) \to O(2k-3, 5)$ exist in principle for all CHL models listed in Table 1, however, they cannot be interpreted in terms of six-dimensional chiral string vacua, due to anomaly cancellation constraints.

## 5 Large radius expansion of exact $(\nabla\Phi)^4$ couplings

In this section, we study the expansion of the proposal (2.27) in the limit where the radius $R$ of one circle in the internal space goes to infinity. We show that it reproduces the known $F^4$ and $\mathcal{R}^2$ couplings in $D = 4$, along with an infinite series of $\mathcal{O}(e^{-R})$ corrections from 1/2-BPS dyons whose wordline winds around the circle, as well as an infinite series of $\mathcal{O}(e^{-R^2})$ corrections from Taub-NUT instantons. We start by analyzing the expansion of genus-one modular integrals (4.1b) and (4.1a) for arbitrary values of $(p,q)$, in the limit near the cusp where $O(p,q)$ is broken to $O(2,1) \times O(p-2,q-2)$, so that the moduli space decomposes into

$$G_{p,q} \to \mathbb{R}^+ \times \left[ \frac{SL(2)}{SO(2)} \times G_{p-2,q-2} \right] \ltimes \mathbb{R}^{2(p+q-4)} \times \mathbb{R} .  \tag{5.1}$$

As in the previous section, we first discuss the maximal rank case $N = 1$, $p - q = 16$, where the integrand is invariant under the full modular group, before dealing with the case of $N$ prime. The reader uninterested by the details of the derivation may skip to §5.3, where we specialize to the values $(p, q) = (r - 4, 8)$ relevant for the $(\nabla \Phi)^4$ couplings in $D = 3$, and interpret the various contributions arising in the decompactification limit to $D = 4$.

## 5.1 $O(p, q) \to O(p - 2, q - 2)$ for even self-dual lattices

We first consider the case where the lattice $\Lambda_{p,q}$ is even self-dual and factorizes in the limit (5.1) as

$$\Lambda_{p,q} \to \Lambda_{p-2,q-2} \oplus II_{2,2} \,. \tag{5.2}$$

In order to study the behavior of the modular integral (4.1a) in the limit (5.1), we denote by $R, S, \phi, a^{I,i}, \psi$ the coordinates for each factors in (5.1), where $i = 1, 2$ and $I = 3, \ldots, p + q - 2$. The coordinate $R$ (not to be confused with the one used in §4) parametrizes a one-parameter subgroup $e^{RH_1}$ in $O(p, q)$, such that the action of the non-compact Cartan generator $H_1$ on the Lie algebra $\mathfrak{so}_{p,q}$ decomposes into

$$\mathfrak{so}_{p,q} \simeq \ldots \oplus (\mathfrak{gl}_1 \oplus \mathfrak{so}_{p-2,q-2})^{(0)} \oplus (\mathbf{2} \otimes (\mathbf{p} + \mathbf{q} - \mathbf{4}))^{(1)} \oplus \mathbf{1}^{(2)}, \tag{5.3}$$

while $(a^{iI}, \psi)$ parametrize the unipotent subgroup obtained by exponentiating the grade 1 and 2 components in this decomposition. We parametrize the $SO(2) \backslash SL(2, \mathbb{R})$ coset representative $v_\mu{}^i$ and the symmetric $SL(2, \mathbb{R})$ element $M \equiv v^T v$ by the complex upper half-plane coordinate $S = S_1 + iS_2$

$$v_\mu{}^i = \frac{1}{\sqrt{S_2}} \begin{pmatrix} 1 & S_1 \\ 0 & S_2 \end{pmatrix}, \quad M^{ij} = \delta^{\mu\nu} v_\mu{}^i v_\nu{}^j = \frac{1}{S_2} \begin{pmatrix} 1 & S_1 \\ S_1 & |S|^2 \end{pmatrix}. \tag{5.4}$$

A generic charge vector $Q_{\mathcal{I}} \in \Lambda_{p,q} \simeq \mathbf{p} + \mathbf{q} \simeq \mathbf{2}^{(-1)} \oplus (\mathbf{p} + \mathbf{q} - \mathbf{4})^{(0)} \oplus \mathbf{2}^{(1)}$ decomposes into $Q = (m^i, \widetilde{Q}_I, n_j)$, where $(m^i, n_i) \in II_{2,2}$ and $\widetilde{Q}_I \in \Lambda_{p-2,q-2}$ such that $Q^2 = -2m^i n_i + \widetilde{Q}^2$. The projectors defined by $Q_L \equiv p_L^{\mathcal{T}} Q_{\mathcal{I}}$ and $Q_R \equiv p_R^{\mathcal{T}} Q_{\mathcal{I}}$ decompose according to

$$
\begin{aligned}
p_{L,\mu}^{\mathcal{I}} Q_{\mathcal{I}} &= \frac{v_{i\mu}^{-1}}{R\sqrt{2}} \left( m^i + a^i \cdot \widetilde{Q} + (\psi \epsilon^{ij} + \tfrac{1}{2} a^i \cdot a^j) n_j \right) - \frac{R}{\sqrt{2}} v_\mu{}^i n_i \\
p_{L,\alpha}^{\mathcal{I}} Q_{\mathcal{I}} &= \tilde{p}_{L,\alpha}^I (\widetilde{Q}_I + n_i a_I^i) \\
p_{R,\mu}^{\mathcal{I}} Q_{\mathcal{I}} &= \frac{v_{i\mu}^{-1}}{R\sqrt{2}} \left( m^i + a^i \cdot \widetilde{Q} + (\psi \epsilon^{ij} + \tfrac{1}{2} a^i \cdot a^j) n_j \right) + \frac{R}{\sqrt{2}} v_\mu{}^i n_i \\
p_{R,\hat{\alpha}}^{\mathcal{I}} Q_{\mathcal{I}} &= \tilde{p}_{R,\hat{\alpha}}^I (\widetilde{Q}_I + n_i a_I^i) \,,
\end{aligned}
\tag{5.5}
$$

where $\tilde{p}_{L,\alpha}^I, \tilde{p}_{R,\hat{\alpha}}^I$ ($\alpha = 3 \ldots p$, $\hat{\alpha} = 3 \ldots q$) are orthogonal projectors in $G_{p-2,q-2}$ which satisfy $\widetilde{Q}^2 = \widetilde{Q}_L^2 - \widetilde{Q}_R^2$.

In order to study the region $R \gg 1$ it is useful to perform a Poisson resummation on the momenta $m^i$ along $II_{2,2}$. Note that this analysis is in principle valid for a region containing $R > \sqrt{2}$. In the case of the scalar integral (4.1b), one obtains

$$\Gamma_{\Lambda_{p,q}} = R^2 \tau_2^{\frac{q-1}{2}} \sum_{A \in \mathbb{Z}^{2 \times 2}} \Gamma_{\Lambda_{p-2,q-2}} \left[ e^{-\frac{\pi}{\tau_2} \frac{R^2}{S_2} \left| (1, S) A \binom{\tau}{1} \right|^2 - 2\pi i (\psi + iR^2) \det A + 2\pi i\, m_i (\widetilde{Q} \cdot a^i + \frac{a^i \cdot a^j}{2} n_j)} \right], \tag{5.6}$$

where $A = \begin{pmatrix} n_1 & m_1 \\ n_2 & m_2 \end{pmatrix}$. In the case of (4.1a), we must distinguish whether the indices $abcd$ lie along the direction $1,2$ or along the directions $\alpha$. Denoting by $h$ the number of indices of the first kind, we get

$$
\Gamma_{\Lambda_{p,q}} \left[ e^{-\frac{\Delta}{8\pi\tau_2}} \left( \prod_{i=1}^{h} (Q_{L,\mu_i}) Q_{L,\alpha_1} \dots Q_{L,\alpha_{4-h}} \right) \right] = R^2 \sum_{A \in \mathbb{Z}^{2\times2}} \left( \frac{R}{i\sqrt{2}} \right)^h
$$
$$
\times \exp\left( -\frac{\pi}{\tau_2} \frac{R^2}{S_2} \left| (1,S) A \begin{pmatrix} \tau \\ 1 \end{pmatrix} \right|^2 - 2\pi i\, T \det A \right) \prod_{k=1}^{h} \left[ \frac{1}{\sqrt{S_2}} \begin{pmatrix} 1 & S_1 \\ 0 & S_2 \end{pmatrix} A \begin{pmatrix} \bar{\tau} \\ 1 \end{pmatrix} \right]_{\mu_k}
$$
$$
\times \Gamma_{\Lambda_{p-2,q-2}+n_i a^i} \left[ e^{-\frac{\Delta}{8\pi\tau_2}} \left[ \widetilde{Q}_{L,\alpha_1} \dots \widetilde{Q}_{L,\alpha_{4-h}} \right] e^{2\pi i m_i (\widetilde{Q}\cdot a^i - \frac{a^i \cdot a^j}{2} n_j)} \right] \quad (5.7)
$$

In this representation, modular invariance is manifest, since a transformation $\tau \to \frac{a\tau+b}{c\tau+d}$ can be compensated by a linear action $A \to A\begin{pmatrix} d & -b \\ -c & a \end{pmatrix}$, under which the last line of (5.7) transforms with weight $12-h$. We can therefore decompose the sum over $A$ into various orbits under $SL(2,\mathbb{Z})$ and apply the unfolding trick to each orbit:

**The trivial orbit**   $A = 0$ produces, up to a factor of $R^2$, the integrals (4.1) or for the lattice $\Lambda_{p-2,q-2}$, provided none of the indices $abcd$ lie along the direction 1 or 2,

$$
F_{\alpha\beta\gamma\delta}^{(p,q),0} = R^2 F_{\alpha\beta\gamma\delta}^{(p-2,q-2)} , \quad F_{\text{tr}}^{(p,q),0} = R^2 F_{\text{tr}}^{(p-2,q-2)} , \quad (5.8)
$$

while it vanishes otherwise (*i.e.* when $h > 0$).

**Rank-one orbit:**   Matrices with $\det A = 0$ but $A \neq 0$ can be decomposed into $A = \begin{pmatrix} 0 & j \\ 0 & p \end{pmatrix} \begin{pmatrix} a & b \\ c & d \end{pmatrix}$, where $(j,p) \in \mathbb{Z}^2 \smallsetminus (0,0)$ and $\begin{pmatrix} a & b \\ c & d \end{pmatrix} \in \Gamma_\infty \backslash SL(2,\mathbb{Z})$. As before we can unfold the fundamental domain $SL(2,\mathbb{Z})\backslash\mathcal{H}$ to the strip $\mathcal{S} = \Gamma_\infty \backslash \mathcal{H} = \mathbb{R}^+_{\tau_2} \times (\mathbb{R}/\mathbb{Z})_{\tau_1}$ using (4.9), leading to

$$
F_{\mu_1\dots\mu_h\alpha_1\dots\alpha_{4-h}}^{(p,q),1} = R^2 \sum_{(j,p)}^{'} \prod_{i=1}^{h} \left( \frac{R}{i\sqrt{2}} \right)^h \left[ \frac{1}{\sqrt{S_2}} \begin{pmatrix} 1 & S_1 \\ 0 & S_2 \end{pmatrix} \begin{pmatrix} j \\ p \end{pmatrix} \right]_{\mu_i}
$$
$$
\times \int_{\mathbb{R}^+} \frac{d\tau_2}{\tau_2^{2+h}} \int_{\mathbb{R}/\mathbb{Z}} d\tau_1 \frac{e^{-\frac{\pi}{\tau_2}\frac{R^2}{S_2}|j+pS|^2}}{\Delta} \Gamma_{\Lambda_{p-2,q-2}} \left[ \widetilde{P}_{\alpha_1\dots\alpha_{4-h}} e^{2\pi i(j\widetilde{Q}\cdot a^1 + p\widetilde{Q}\cdot a^2)} \right],
$$
$$
F_{\text{tr}}^{(p,q),1} = R^2 \sum_{(j,p)}^{'} \int_{\mathbb{R}^+} \frac{d\tau_2}{\tau_2^{2+h}} \int_{\mathbb{R}/\mathbb{Z}} d\tau_1 e^{-\frac{\pi}{\tau_2}\frac{R^2}{S_2}|j+pS|^2} \Gamma_{\Lambda_{p-2,q-2}} \left[ e^{2\pi i(j\widetilde{Q}\cdot a^1 + p\widetilde{Q}\cdot a^2)} \right] D^2\left( \frac{1}{\Delta} \right) ,
$$
$$
(5.9)
$$

for the tensor integral with $0 \le h \le 4$ indices along the large torus and its trace respectively. Inserting the Fourier expansion (4.14), the integral over $\tau_1$ picks up the Fourier coefficient $c(m)$ with $m = -\frac{1}{2}\widetilde{Q}^2$. The remaining integral over $\tau_2$ can be computed after expanding $\widetilde{P}_{\alpha_1\dots\alpha_{4-h}} = \sum_{\ell=0}^{\lfloor\frac{4-h}{2}\rfloor} \widetilde{P}_{\alpha_1\dots\alpha_{4-h}}^{(\ell)} \tau_2^{-\ell}$, where $\widetilde{P}_{\alpha_1\dots\alpha_{4-h}}^{(\ell)}$ is a polynomial in $\widetilde{Q}$ of degree $4-h-2\ell \ge 0$, or

vanishing otherwise. The contribution of $\widetilde{Q} = 0$ produces power-like terms in $R^2$,

$$F_{\alpha\beta\gamma\delta}^{(p,q),1,0} = R^{q-6}\frac{3c(0)}{8\pi^2}\,\mathcal{E}^\star(\tfrac{8-q}{2},S)\,\delta_{(\alpha\beta}\delta_{\gamma\delta)},$$

$$F_{\mu\nu\gamma\delta}^{(p,q),1,0} = R^{q-6}\frac{c(0)}{4\pi^2}\Big[\tfrac{8-q}{4}\delta_{\alpha\beta}\delta_{\mu\nu} - \delta_{\alpha\beta}\mathcal{D}_{\mu\nu}\Big]\mathcal{E}^\star(\tfrac{8-q}{2},S),\qquad(5.10)$$

$$F_{\mu\nu\rho\sigma}^{(p,q),1,0} = R^{q-6}\frac{c(0)}{2\pi^2}\Big[\mathcal{D}_{\mu\nu\rho\sigma}^2 - \tfrac{10-q}{2}\delta_{(\mu\nu}\mathcal{D}_{\rho\sigma)} + \Big(\tfrac{8-q}{2}\Big)\Big(\tfrac{10-q}{2}\Big)\tfrac{3}{8}\delta_{(\mu\nu}\delta_{\rho\sigma)}\Big]\mathcal{E}^\star(\tfrac{8-q}{2},S)$$

for the tensor integral, and

$$F_{\mathrm{tr}}^{(p,q),1,0} = R^{q-6}\frac{c(0)}{8\pi^2}(p-q+6)(p-q+8)\mathcal{E}^\star(\tfrac{8-q}{2},S)\qquad(5.11)$$

for its trace. Here, $\mathcal{E}^\star(s,S)$ is the completed weight 0 non-holomorphic Eisenstein series,

$$\mathcal{E}^\star(s,S) = \frac{1}{2}\pi^{-s}\,\Gamma(s)\sideset{}{'}\sum_{(m,n)\in\mathbb{Z}^2}\frac{S_2^s}{|nS+m|^{2s}} \equiv \xi(2s)\mathcal{E}(s,S)\,,\qquad(5.12)$$

$\mathcal{D}_{\mu\nu}$ is the traceless differential operator on $\frac{SL(2,\mathbb{R})}{SO(2)}$ defined in appendix D, and $\mathcal{D}_{\mu\nu\rho\sigma}^2 = \mathcal{D}_{(\mu\nu}\mathcal{D}_{\rho\sigma)} - \tfrac{1}{4}\delta_{(\mu\nu}\delta_{\rho\sigma)}\mathcal{D}_{\tau\kappa}\mathcal{D}^{\tau\kappa}$ is the traceless operator of degree 2 in the symmetric representation. The equalities used to write (5.10) are detailed in (D.9), and similar expressions using non-holomorphic series of non-zero weight are given in (D.7). Recall that $\mathcal{E}^\star(s,S)$ is invariant under $s \mapsto 1-s$, and has simple poles at $s=0$ and $s=1$. As in the previous section, the pole at $q=6$ is subtracted by the regularization prescription mentioned below (3.32), while the pole at $q=8$ cancels against the pole from the zero orbit contribution (5.8).

Contributions of non-zero vectors $\widetilde{Q} \in \Lambda_{p-2,q-2}$, on the other hand, lead to exponentially suppressed contributions, *e.g.* for the trace of the tensor integral

$$2R^{\frac{q}{2}}\sideset{}{'}\sum_{\widetilde{Q}\in\Lambda_{p-2,q-2}}\sideset{}{'}\sum_{(j,p)}e^{2\pi i(j\widetilde{Q}\cdot a_1 + p\widetilde{Q}\cdot a_2)}\sum_{\ell=0}^{2}\frac{a_\ell}{R^{2\ell}}\Big(-\tfrac{\widetilde{Q}^2}{2}\Big)^{2-\ell}c\Big(-\tfrac{\widetilde{Q}^2}{2}\Big)\Big(\frac{2\widetilde{Q}_R^2 S_2}{|j+pS|^2}\Big)^{\frac{q-4-2\ell}{4}}$$

$$\times K_{\frac{q-4}{2}-\ell}\Big(2\pi\sqrt{\tfrac{2R^2}{U_2}}|j+pU||\widetilde{Q}_R|\Big)\quad(5.13)$$

Defining $(Q,P) = (j,p)\widetilde{Q}$, we see that the Fourier expansion with respect to $(a_1,a_2)$ has support on collinear vectors $(Q,P)$ with $Q,P \in \Lambda_{p-2,q-2}$. Extracting the greatest common divisor of $(j,p)$, we find that the Fourier coefficients with charge $Q'^i = (Q,P)$ and mass $\mathcal{M}(Q,P) = \sqrt{2Q_R'^i Q_R'^j M_{ij}}$ defined in (2.13) are given by

$$F_{\alpha\beta\gamma\delta}^{(p,q),1,Q'} = 4R^{\frac{q}{2}}\,\bar{c}(Q'^i)\sum_{\ell=0}^{2}\frac{\mathcal{P}_{\alpha\beta\delta\gamma}^{(\ell)}(Q'^i,S)}{R^{2\ell}}\frac{K_{\frac{q-4}{2}-\ell}(2\pi R\mathcal{M}(Q,P))}{\mathcal{M}(Q,P)^{\frac{q-4}{2}-\ell}}$$

$$F_{\mu\alpha\beta\gamma}^{(p,q),1,Q'} = 4R^{\frac{q}{2}}\,\bar{c}(Q'^i)\sum_{\ell=0}^{1}\frac{\mathcal{P}_{\mu\alpha\beta\delta}^{(\ell)}(Q'^i,S)}{i\sqrt{2}R^{2\ell}}\frac{K_{\frac{q-6}{2}-\ell}(2\pi R\mathcal{M}(Q,P))}{\mathcal{M}(Q,P)^{\frac{q-6}{2}-\ell}}$$

$$\vdots$$

$$F_{\mu\nu\rho\sigma}^{(p,q),1,Q'} = 4R^{\frac{q}{2}}\,\bar{c}(Q'^i)\frac{\mathcal{P}_{\mu\nu\sigma\rho}^{(0)}(Q'^i,S)}{4}\frac{K_{\frac{q-12}{2}}(2\pi R\mathcal{M}(Q,P))}{\mathcal{M}(Q,P)^{\frac{q-12}{2}}}\qquad(5.14)$$

for the tensor integral, and

$$F_{\text{tr}}^{(p,q),1,Q'} = 4R^{\frac{q}{2}}\,\bar{c}(Q'^i)\sum_{\ell=0}^{2}\frac{a_\ell}{R^{2\ell}}\left[-\frac{\gcd(Q'^i\cdot Q'^j)}{2}\right]^{2-\ell}\frac{K_{\frac{q-4}{2}-\ell}(2\pi R\mathcal{M}(Q,P))}{\mathcal{M}(Q,P)^{\frac{q-4}{2}-\ell}} \tag{5.15}$$

for its trace. The covariantized versions of $P_{abcd}(Q)$ with respect to the torus' metric, $\mathcal{P}^{(\ell)}_{\alpha\beta\gamma\delta},\ldots,\mathcal{P}^{(\ell)}_{\mu\nu\sigma\rho}$ are given in appendix C. Finally the degeneracy is given by

$$\bar{c}(Q,P) = \sum_{(Q,P)/d\in\Lambda^{\oplus 2}_{p-2,q-2}}\left(\frac{d^2}{\gcd(Q^2,Q\cdot P,P^2)}\right)^{\frac{q-8}{2}}c\left(-\frac{\gcd(Q^2,Q\cdot P,P^2)}{2d^2}\right), \tag{5.16}$$

with support $(Q,P)\in\Lambda_{p-2,q-2}\oplus\Lambda_{p-2,q-2}$.

**Rank-two orbit** Finally, rank-two matrices can be uniquely decomposed as $A = \begin{pmatrix} k & j \\ 0 & p \end{pmatrix}\begin{pmatrix} a & b \\ c & d \end{pmatrix}$ where $k > j \geq 0$ and $p \neq 0$ and $\begin{pmatrix} a & b \\ c & d \end{pmatrix}\in SL(2,\mathbb{Z})$. The matrices $A$ can therefore be restricted to $A = \begin{pmatrix} k & j \\ 0 & p \end{pmatrix}$, provided the integral is extended to the double cover of the upper half-plane $\mathcal{H}$. This leads to

$$F_{\mu_1\ldots\mu_h\alpha_1\ldots\alpha_{4-h}}^{(p,q),1} = 2R^2\sum_{\substack{k>j\geq 0 \\ p\neq 0}}\left(\frac{R}{i\sqrt{2}}\right)^h e^{-2\pi ikp(\psi+iR^2)}\int_{\mathbb{R}^+}\frac{d\tau_2}{\tau_2^{2+h}}\int_{\mathbb{R}}d\tau_1\frac{e^{-\frac{\pi}{\tau_2}\frac{R^2}{S_2}|k\tau+j+pS|^2}}{\Delta}$$

$$\times\prod_{l=1}^{h}\left[\frac{1}{\sqrt{S_2}}\begin{pmatrix} 1 & S_1 \\ 0 & S_2 \end{pmatrix}\begin{pmatrix} k\bar{\tau}+j \\ p \end{pmatrix}\right]_{\mu_l}\Gamma_{\Lambda_{p-2,q-2}+n_ia^i}\left[P_{\alpha_1\ldots\alpha_{4-h}}\,e^{2\pi i\left(j(\widetilde{Q}-\frac{1}{2}ka_1)\cdot a_1+p(\widetilde{Q}-\frac{1}{2}ka_1)\cdot a_2\right)}\right] \tag{5.17}$$

for the tensor integral, and to

$$F_{\text{tr}}^{(p,q),1} = 2R^2\sum_{\substack{k>j\geq 0 \\ p\neq 0}}e^{-2\pi ikp(\psi+iR^2)}\int_{\mathbb{R}^+}\frac{d\tau_2}{\tau_2^2}\int_{\mathbb{R}}d\tau_1\,e^{-\frac{\pi}{\tau_2}\frac{R^2}{S_2}|k\tau+j+pS|^2}$$

$$\times\Gamma_{\Lambda_{p-2,q-2}+n_ia^i}\left[e^{2\pi i\left(j(\widetilde{Q}-\frac{1}{2}ka_1)\cdot a_1+p(\widetilde{Q}-\frac{1}{2}ka_1)\cdot a_2\right)}\right]D^2\left(\frac{1}{\Delta}\right) \tag{5.18}$$

for its trace.

Inserting the Fourier expansion (4.14), the integral over $\tau_1$ is Gaussian while the integral over $\tau_2$ is of Bessel type. The sum over $0 \leq j < k$ enforces a Kronecker delta function modulo $k$,

$$\sum_{j=0}^{k-1}\exp\left[2\pi i\frac{j}{k}\left(\frac{\widetilde{Q}^2}{2}+m\right)\right] = \begin{cases} k & \text{if } \frac{\widetilde{Q}^2}{2}+m = lk, \quad l\in\mathbb{Z}, \\ 0 & \text{otherwise} \end{cases} \tag{5.19}$$

Relabelling the charges as $p\widetilde{Q}\to P$, $kp\to -M_1$ and $lp\to -M_2$, and defining $D = -\frac{P^2}{2}+M_1M_2$ one obtains, for the trace of the tensor integral,

$$F_{\text{tr}}^{(p,q),2} = \sum_{\substack{M_1\neq 0,M_2 \\ P\in\Lambda_{p-2,q-2}}}F_{\text{tr}}^{(p,q),2,M_1}\left(P-M_1a_1,M_2-a_1\cdot P+\tfrac{1}{2}(a_1\cdot a_1)M_1\right)$$

$$\times e^{2\pi i(P\cdot a_2+M_1(\psi-\frac{1}{2}a_1\cdot a_2)+(M_2-a_1\cdot P+\frac{1}{2}(a_1\cdot a_1)M_1)S_1)} \tag{5.20}$$

where $F_{\mathrm{tr}}^{(p,q),2,M_1}$ is the non-Abelian Fourier coefficient,

$$F_{\mathrm{tr}}^{(p,q),2,M_1}(P,M_2) = 4(R^2 S_2)^{\frac{q-2}{2}}\,\bar{c}(M_1,M_2,P)\sum_{\ell=0}^{2}\frac{a_\ell\,D^{2-\ell}}{(R^2 S_2)^\ell}\left(\frac{2\pi}{S_{\mathrm{cl}}}\right)^{\frac{q-5}{2}-\ell}K_{\frac{q-5}{2}-\ell}(S_{\mathrm{cl}})\,, \qquad (5.21)$$

$S_{\mathrm{cl}}$ is the classical action

$$S_{\mathrm{cl}}(M_1,M_2,P) = 2\pi\sqrt{(R^2 M_1 + S_2 M_2)^2 + 2R^2 S_2 P_R^2}\quad, \qquad (5.22)$$

and $\bar{c}(M_1,M_2,P)$ the summation measure

$$\bar{c}(M_1,M_2,P) = \sum_{\substack{d\mid(M_1,M_2)\\ P/d\in\Lambda_{p-2,q-2}}} c\left(\tfrac{D}{d^2}\right)d^{q-7}\,. \qquad (5.23)$$

It is worth noting that (5.20) is the general expansion of a function of $(S_1,a_1,a_2,\psi)$ invariant under discrete shifts $T_{b,\epsilon_1,\epsilon_2,\kappa}$ acting as

$$(S_1,a_1,a_2,\psi)\mapsto\left(S_1+b,a_1+\epsilon_1,a_2+\epsilon_2+ba_1,\psi+\kappa+\tfrac{1}{2}[\epsilon_2(a_1+\epsilon_1)-\epsilon_1(a_2+ba_1)]\right) \quad (5.24)$$

with $b,\kappa\in\mathbb{Z}$ and $\epsilon_1,\epsilon_2\in\mathbb{Z}^{p-2,q-2}$. Invariance under $T_{b,0,\epsilon_2,\kappa}$ is manifest, while invariance under $T_{0,\epsilon_1,0,0}$ is realized by shifting $P\mapsto P+M_1\epsilon_1, M_2\mapsto M_2+\epsilon_1 P+\tfrac{1}{2}M_1\epsilon_1^2$, which leaves $D$ and $\tilde{M}_2 = M_2 - a_1\cdot P+\tfrac{1}{2}(a_1\cdot a_1)M_1$ invariant. It is worth noting that in the special case $p=2$, $P_R^2$ vanishes identically so (5.22) simplifies to $S_{\mathrm{cl}} = 2\pi|R^2 M_1 + S_2 M_2|$.

Similarly, for the tensor integral, we get

$$F_{\alpha\beta\gamma\delta}^{(p,q),2,M_1}(P,M_2) = 4(R^2 S_2)^{\frac{q-2}{2}}\,\bar{c}(M_1,M_2,P)\sum_{\ell=0}^{2}\frac{\tilde{P}_{\alpha\beta\gamma\delta}^{(\ell)}(P)}{(R^2 S_2)^\ell}\left(\frac{2\pi}{S_{\mathrm{cl}}}\right)^{\frac{q-5}{2}-\ell}K_{\frac{q-5}{2}-\ell}(S_{\mathrm{cl}})$$

$$F_{2\alpha\beta\gamma}^{(p,q),2,M_1}(P,M_2) = 4(R^2 S_2)^{\frac{q-2}{2}}\,\bar{c}(M_1,M_2,P)\sum_{\ell=0}^{1}\frac{\tilde{P}_{\alpha\beta\gamma}^{(\ell)}(P)}{\mathrm{i}\sqrt{2}(R^2 S_2)^{\ell-\frac{1}{2}}}\left(\frac{2\pi}{S_{\mathrm{cl}}}\right)^{\frac{q-7}{2}-\ell}K_{\frac{q-7}{2}-\ell}(S_{\mathrm{cl}}) \qquad (5.25)$$

$$\vdots$$

$$F_{2222}^{(p,q),2,M_1}(P,M_2) = 4(R^2 S_2)^{\frac{q-2}{2}}\,\bar{c}(M_1,M_2,P)\frac{\tilde{P}^{(0)}}{4(R^2 S_2)^{-2}}\left(\frac{2\pi}{S_{\mathrm{cl}}}\right)^{\frac{q-13}{2}}K_{\frac{q-13}{2}}(S_{\mathrm{cl}}),$$

where we restricted to the cases $\mu,\nu,\ldots=2$ for simplicity.

## 5.2 Extension to $\mathbb{Z}_N$ CHL orbifolds

The degeneration limit (5.1) of the modular integrals (4.1) for $\mathbb{Z}_N$ CHL models with $N=2,3,5,7$ can be treated similarly by applying the orbit method. In (4.1), $\Delta_k$ is the cusp form of weight $k=\frac{24}{N+1}$ defined in (2.4), and $\Gamma_{\Lambda_{p,q}}[P_{abcd}]$ is the partition function with insertion of $P_{abcd}$ for a lattice

$$\Lambda_{p,q} = \Lambda_{p-2,q-2}\oplus I\!I_{1,1}\oplus I\!I_{1,1}[N]\,, \qquad (5.26)$$

where $\Lambda_{p-2,q-2}$ is a lattice of level $N$. The lattice $I\!I_{1,1}\oplus I\!I_{1,1}[N]$ is obtained from the usual unimodular lattice $I\!I_{2,2}$ by restricting the windings and momenta to $(n_1,n_2,m_1,m_2)\in\mathbb{Z}\oplus N\mathbb{Z}\oplus\mathbb{Z}\oplus\mathbb{Z}$, hence breaking the automorphism group $O(2,2,\mathbb{Z})$ to $\sigma_{S\leftrightarrow T}\ltimes[\Gamma_0(N)\times\Gamma_0(N)]$. After Poisson resummation on $m_2$, Eq. (5.6) and (5.7) continue to hold, except for the fact that $n_2$ is restricted to run over $N\mathbb{Z}$. The sum over $A=\begin{pmatrix}n_1 & m_1\\ n_2 & m_2\end{pmatrix}$ can then be decomposed into orbits of $\Gamma_0(N)$:[11]

---

[11]Note that the subsequent analysis is valid in the region of the moduli space where $NR^2 > 2S_2$.

**Trivial orbit**  The contribution of $A = 0$ reduces, up to a factor of $R^2$, to the integrals (4.1) for the lattice $\Lambda_{p-2,q-2}$,

$$F_{\alpha\beta\gamma\delta}^{(p,q),0} = R^2 \, F_{\alpha\beta\gamma\delta}^{(p-2,q-2)} \,, \qquad F_{\text{tr}}^{(p,q),0} = R^2 \, F_{\text{tr}}^{(p-2,q-2)} \,, \tag{5.27}$$

**Rank-one orbits**  Matrices $A$ of rank-one fall into two different classes of orbits under $\Gamma_0(N)$. For simplicity, let us first consider the case where $(n_2, m_2) \neq (0,0)$, and denote $(m_2, n_2) = p(n_2', m_2')$, with $p = \gcd(n_2, m_2)$:

- Matrices with $n_2' = 0 \bmod N$, as they are required to be rank-one, can be decomposed as $\begin{pmatrix} n_1 & m_1 \\ n_2 & m_2 \end{pmatrix} = \begin{pmatrix} 0 & j \\ 0 & p \end{pmatrix}\begin{pmatrix} a & b \\ c & d \end{pmatrix}$ with $(j,p) \in \mathbb{Z}^2 \smallsetminus \{(0,0)\}$, $p \neq 0$ and $\begin{pmatrix} a & b \\ c & d \end{pmatrix} \in \Gamma_\infty \backslash \Gamma_0(N)$. For this class of orbit, one can thus unfold directly the domain $\Gamma_0(N)\backslash\mathcal{H}$ into the unit strip $\mathcal{S} = \Gamma_\infty \backslash \mathcal{H} = \mathbb{R}_{\tau_2}^+ \times (\mathbb{R}/\mathbb{Z})_{\tau_1}$.

- Matrices with $n_2' \neq 0 \bmod N$ can be decomposed as $\begin{pmatrix} n_1 & m_1 \\ n_2 & m_2 \end{pmatrix} = \begin{pmatrix} j & 0 \\ p & 0 \end{pmatrix}\begin{pmatrix} a & b \\ c & d \end{pmatrix}$ with $(j,p) \in \mathbb{Z} \oplus N\mathbb{Z} \smallsetminus \{(0,0)\}$, $p \neq 0$ and $\begin{pmatrix} a & b \\ c & d \end{pmatrix} \in S\,\Gamma_{\infty,N}\,S^{-1}\backslash\Gamma_0(N)$, where $\Gamma_{\infty,N} = \{\begin{pmatrix} 1 & n \\ 0 & 1 \end{pmatrix}, n \in N\mathbb{Z}\}$. One can then unfold the fundamental domain $\Gamma_0(N)\backslash\mathcal{H}$ into $S\,\Gamma_{\infty,N}\,S^{-1}\backslash\mathcal{H}$, and change variable $\tau \to -1/\tau$ as in the weak coupling case (4.24) to recover the integration domain $\mathcal{S}_N = \Gamma_{\infty,N}\backslash\mathcal{H} = \mathbb{R}_{\tau_2}^+ \times (\mathbb{R}/N\mathbb{Z})_{\tau_1}$, the width-$N$ strip.

The remaining contributions $A$ with $(n_2, m_2) = (0,0)$ belong to the two classes of orbits above. Let $(n_1, m_1) = j(n_1', m_1')$, where $j = \gcd(n_1, m_1)$ and $j \in \mathbb{Z}$, then contributions with $n_1' = 0 \bmod N$ correspond to the cases $(j,p) = (j,0)$ in the first class above; contributions with $n_1' \neq 0 \bmod N$ correspond to $(j,p) = (j,0)$ in the second class above.

After unfolding and changing variable, the result for the simplest component $F_{\alpha\beta\gamma\delta}^{(p,q),1}$ reads (similarly to (4.24))

$$\begin{aligned} F_{\alpha\beta\gamma\delta}^{(p,q),1} = {}&R^2 \int_{\mathbb{R}^+} \frac{d\tau_2}{\tau_2^2} \int_{\mathbb{R}/\mathbb{Z}} d\tau_1 \, \frac{1}{\Delta_k(\tau)} \sideset{}{'}\sum_{(j,p)\in\mathbb{Z}^2} e^{-\frac{\pi R^2}{\tau_2 S_2}|j+pS|^2} \Gamma_{\Lambda_{p-2,q-2}} \left[ e^{2\pi i (j\widetilde{Q}\cdot a_1 + p\widetilde{Q})\cdot a_2} P_{\alpha\beta\gamma\delta} \right] \\ &+ R^2 \int_{\mathbb{R}^+} \frac{d\tau_2}{\tau_2^2} \int_{\mathbb{R}/\mathbb{Z}} d\tau_1 \, \frac{1}{\Delta_k(\tau/N)} \frac{\upsilon}{N} \sideset{}{'}\sum_{\substack{(j,p)\in\mathbb{Z}^2 \\ p=0\bmod N}} e^{-\frac{\pi R^2}{\tau_2 S_2}|j+pS|^2} \Gamma_{\Lambda_{p-2,q-2}^*} \left[ e^{2\pi i (j\widetilde{Q}\cdot a_1 + p\widetilde{Q})\cdot a_2} P_{\alpha\beta\gamma\delta} \right] \,, \end{aligned} \tag{5.28}$$

where $\Gamma_{\Lambda_{p-2,q-2}^*}$ is the partition function of the dual lattice $\Lambda_{p-2,q-2}^*$ and where $\upsilon = N^{k/2+1}|\Lambda_{p-2,q-2}^*/\Lambda_{p-2,q-2}|^{-1/2}$ (which reduces to $\upsilon = N^{1-\delta_{q,8}}$ for $q \leq 8$ in the cases of interest). The contributions from $\widetilde{Q} = 0$ thus give

$$\begin{aligned} F_{\alpha\beta\gamma\delta}^{(p,q),1,0} &= R^{q-6} \frac{3(2c_k(0))}{8\pi^2} \frac{1}{2} \left( \mathcal{E}_{\frac{8-q}{2}}^\star(S) + \upsilon N^{\frac{q-8}{2}} \mathcal{E}_{\frac{8-q}{2}}^\star(NS) \right) \delta_{(\alpha\beta}\delta_{\gamma\delta)}, \\ F_{\mu\nu\gamma\delta}^{(p,q),1,0} &= R^{q-6} \frac{2c_k(0)}{4\pi^2} \left[ \frac{8-q}{4}\delta_{\alpha\beta}\delta_{\mu\nu} - \delta_{\alpha\beta}\mathcal{D}_{\mu\nu} \right] \frac{1}{2} \left( \mathcal{E}_{\frac{8-q}{2}}^\star(S) + \upsilon N^{\frac{q-8}{2}} \mathcal{E}_{\frac{8-q}{2}}^\star(NS) \right), \\ F_{\mu\nu\rho\sigma}^{(p,q),1,0} &= R^{q-6} \frac{2c_k(0)}{2\pi^2} \\ &\quad \times \left[ \mathcal{D}_{\mu\nu\rho\sigma}^2 - \frac{10-q}{2}\delta_{(\mu\nu}\mathcal{D}_{\rho\sigma)} + \left(\frac{8-q}{2}\right)\left(\frac{10-q}{2}\right)\frac{3}{8}\delta_{(\mu\nu}\delta_{\rho\sigma)} \right] \frac{1}{2} \left( \mathcal{E}_{\frac{8-q}{2}}^\star(S) + \upsilon N^{\frac{q-8}{2}} \mathcal{E}_{\frac{8-q}{2}}^\star(NS) \right), \end{aligned} \tag{5.29}$$

for the tensor integral, and

$$F_{\mathrm{tr}}^{(p,q),1,0} = R^{q-6}(p-q+6)(p-q+8)\frac{2c_k(0)}{8\pi^2}\frac{1}{2}\left(\mathcal{E}^\star_{\frac{8-q}{2}}(S) + vN^{\frac{q-8}{2}}\mathcal{E}^\star_{\frac{8-q}{2}}(NS)\right),\qquad(5.30)$$

for its trace. Recall $c_k(0) = \frac{24}{N+1} = k$ is the zero mode of $1/\Delta_k = \sum_m c_k(m)q^m$. As in (5.11) and (5.10), the pole at $q = 6$ is minimally subtracted by the regularization prescription mentioned below (3.32), while the pole at $q = 8$ cancels against the pole from the zero orbit contribution (5.27).

The contributions with $\widetilde{Q} \neq 0$ are exponentially suppressed at large $R$, and have similar Fourier coefficients as in the full rank case (5.14), except for a different summation measure. Let us label the electromagnetic charges by $(Q,P) = (j,p)\widetilde{Q} = (j',p')\hat{Q}$ where $(j',p')$ are coprime integers. It will be useful to classify all possible rank-one charges $(Q,P)$ in orbits of the S-duality group $\Gamma_0(N)$ acting as $\binom{Q}{P} \to \binom{a\ b}{c\ d}\binom{Q}{P}$, where $\binom{a\ b}{c\ d} \in \Gamma_0(N)$.

- Charges $(Q,P)$ such that $p' = 0 \bmod N$ are in the same orbit as purely electric charges $(\hat{Q},0)$. Their Fourier coefficient gets contributions from both terms in (5.28) with $d = \gcd(j,p)$ and $\frac{\hat{Q}}{d} = \tilde{Q} \in \Lambda_{p-2,q-2}$ in the first case and $\frac{\hat{Q}}{d} = \tilde{Q} \in \Lambda^*_{p-2,q-2}$ in the second, such that they are weighted by the measure

$$\bar{c}_k(Q,P) = \sum_{\substack{d\geq 1 \\ \hat{Q}/d\in\Lambda_{p-2,q-2}}} c_k\left(-\frac{\hat{Q}^2}{2d^2}\right)\left(\frac{d^2}{\hat{Q}^2}\right)^{\frac{q-8}{2}} + v\sum_{\substack{d\geq 1 \\ \hat{Q}/d\in\Lambda^*_{p-2,q-2}}} c_k\left(-\frac{N\hat{Q}^2}{2d^2}\right)\left(\frac{d^2}{N\hat{Q}^2}\right)^{\frac{q-8}{2}},\qquad(5.31)$$

  where the first contribution has support $Q \in \Lambda_{p-2,q-2} \subset \Lambda^*_{p-2,q-2}$, while the second has support on $Q \in \Lambda^*_{p-2,q-2}$. Notice that the latter is matched against $1/\Delta_k(\tau/N)$, which explains the $N$ factor in the argument of $c_k$.

- Charges $(Q,P)$ such that $p' \neq 0 \bmod N$ are in the same orbit as purely magnetic charges $(0,\hat{P})$, where we relabelled $\hat{Q}$ as $\hat{P}$ for convenience. Their Fourier coefficient gets contributions from both terms in (5.28) with $d = \gcd(j,p)$ and $\frac{\hat{P}}{d} = \tilde{Q} \in \Lambda_{p-2,q-2}$ in the first case and $Nd = \gcd(j,p)$ (because $j = 0 \bmod N$) and $\frac{\hat{P}}{Nd} = \tilde{Q} \in \Lambda^*_{p-2,q-2}$ in the second, such that they are weighted by the measure

$$\bar{c}_k(Q,P) = \sum_{\substack{d\geq 1 \\ \hat{P}/d\in\Lambda_{p-2,q-2}}} c_k\left(-\frac{\hat{P}^2}{2d^2}\right)\left(\frac{d^2}{\hat{P}^2}\right)^{\frac{q-8}{2}} + v\sum_{\substack{d\geq 1 \\ \hat{P}/d\in N\Lambda^*_{p-2,q-2}}} c_k\left(-\frac{\hat{P}^2}{2Nd^2}\right)\left(\frac{Nd^2}{\hat{P}^2}\right)^{\frac{q-8}{2}},\qquad(5.32)$$

  where the first contribution has support $P \in \Lambda_{p-2,q-2}$, while the second has support $P \in N\Lambda^*_{p-2,q-2} \subset \Lambda_{p-2,q-2}$. In the latter contribution, one $N$ factor in the argument of $c_k$ comes from the matching condition, and two $N$ factors in its denominator come from all divisors $d$ being originally multiples of $N$.

**Rank-two orbit**  For the rank-two matrices $A$, the two classes of orbits are similarly given by studying $(n_2,m_2) = p(n'_2,m'_2)$, where $p = \gcd(n_2,m_2)$.

- Contributions for which $(n'_2, m'_2) = (0,1) \bmod N$ can be decomposed as $A = \begin{pmatrix} k & j \\ 0 & p \end{pmatrix}\begin{pmatrix} a & b \\ c & d \end{pmatrix}$, $0 \leq j < k$, $p \in \mathbb{Z} \smallsetminus \{0\}$ and $\begin{pmatrix} a & b \\ c & d \end{pmatrix} \in SL(2,\mathbb{Z})$, where its representative has trivial stabilizer. For this first class of orbits, the fundamental domain can be unfolded to the full upper half-plane $\mathcal{H} = \mathbb{R}^+_{\tau_2} \times \mathbb{R}_{\tau_1}$.

- Contributions for which $(n'_2, m'_2) = (1,0) \bmod N$ can have $A = \begin{pmatrix} j & k \\ p & 0 \end{pmatrix}\begin{pmatrix} a & b \\ c & d \end{pmatrix}$, $0 \leq j < Nk$, $p \in N\mathbb{Z} \smallsetminus \{0\}$ and $\begin{pmatrix} a & b \\ c & d \end{pmatrix} \in SL(2,\mathbb{Z})$, where representative has trivial stabilizer. For this second class of orbits, the fundamental domain can be unfolded to $\mathcal{H} = \mathbb{R}^+_{\tau_2} \times \mathbb{R}_{\tau_1}$ as well and the integrand can be brought back to the standard lattice sum representation using a transformation $\tau \to -1/\tau$, in the spirit of (5.28).

Both classes of contributions lead to the same type of non-Abelian Fourier coefficient as in the unorbifolded case (5.21) and (5.25), except for a different summation measure $\bar{c}(M_1, M_2, P)$. The first class have support $(M_1, M_2, P) \in \mathbb{Z} \oplus \mathbb{Z} \oplus \Lambda_{p-2,q-2}$, whereas the second class have support $(M_1, M_2, P) \in N\mathbb{Z} \oplus N\mathbb{Z} \oplus N\Lambda^*_{p-2,q-2}$. In fine the summation measure reads

$$\bar{c}_k(M_1, M_2, P) = \sum_{\substack{d|(M_1,M_2) \\ P/d \in \Lambda_{p-2,q-2}}} c_k\left(\frac{D}{d^2}\right) d^{q-7} + \upsilon \sum_{\substack{Nd|(M_1,M_2) \\ P/d \in N\Lambda^*_{p-2,q-2}}} c_k\left(\frac{D}{Nd^2}\right)(Nd)^{q-7}, \tag{5.33}$$

where we recall that $D = -\frac{1}{2}P^2 + M_1 M_2$. For the second class of orbits, one factor of $N$ in the argument of $c_k$ comes from the matching condition, and two factors of $1/N$ come from the fact that all divisors were originally multiples of $N$.

## 5.3 Large radius limit and BPS dyons

Specializing to $(p,q) = (2k,8) = (r-4,8)$, and choosing $\Lambda_{p-2,q-2} = \Lambda_m$, the degeneration studied in this section corresponds to the limit of the exact $(\nabla \Phi)^4$ amplitude in heterotic string on $T^7$ in the limit where a circle inside $T^7$, orthogonal to the $\mathbb{Z}_N$ action, decompactifies. The coordinate $R$ is identified as the radius of the large circle in units of the four-dimensional Planck length $l_P = g_4 l_H$. The contributions from the various orbits discussed in §5.1 and §5.2 are then interpreted as follows:

### 5.3.1 Effective action in $D = 4$

In the large $R$ limit, $F^{(2k,8)}_{\alpha\beta\gamma\delta}$ should reproduce the exact four-dimensional $F^4$ coupling, up to exponentially suppressed corrections. As already mentioned below (5.10) and (5.29), the contribution of the vector $\widetilde{Q} = 0$ to the rank-one orbit has a pole at $q = 8$. Using the regularisation (3.32), that formally sets $q = 8 + 2\epsilon$, one obtains

$$F^{(2k,8),1,0}_{\alpha\beta\gamma\delta}(\epsilon) = R^{2+2\epsilon} \frac{3(2k)}{(4\pi)^2} \left(\mathcal{E}^\star_{-\epsilon}(S) + N^\epsilon \mathcal{E}^\star_{-\epsilon}(NS)\right) \delta_{(\alpha\beta}\delta_{\gamma\delta)} \tag{5.34}$$

$$= R^2 \frac{3}{2(2\pi)^2}\left(\frac{k}{\epsilon} - \log(S_2^k |\Delta_k(S)|^2) + k\left(\log\left(\frac{R^2}{4\pi}\right) - \gamma\right)\right) \delta_{(\alpha\beta}\delta_{\gamma\delta)} + \mathcal{O}(\epsilon),$$

However, this pole cancels against the pole (4.38) in the trivial zero-orbit contribution (5.8), (5.27), leaving the finite result

$$F^{(2k,8)}_{\alpha\beta\gamma\delta} = R^2 \left(-\frac{3}{2(2\pi)^2}\left(\log(S_2^k |\Delta_k(S)|^4) - 2k \log R\right)\delta_{(\alpha\beta}\delta_{\gamma\delta)} + \hat{F}^{(2k-2,6)}_{\alpha\beta\gamma\delta}(\Phi)\right) + \dots \tag{5.35}$$

where $\hat{F}^{(2k-2,6)}_{\alpha\beta\gamma\delta}$ is the renormalized 1-loop coupling, up to an irrelevant additive constant, and the dots denote exponentially suppressed terms.

Thus, the conjectural formula (2.27) for the exact $(\nabla\Phi)^4$ coupling in $D=4$ predicts that the exact $F^4$ coupling in four dimensions should be given by

$$-\frac{3}{8\pi^2}\log(S_2^k|\Delta_k(S)|^2)\delta_{(ab}\delta_{cd)} + F^{(2k-2,6)}_{abcd}(\Phi) \,, \tag{5.36}$$

where for convenience we renamed the indices $\alpha, \beta, \ldots$ into $a, b, \ldots$ running from 1 to $2k-2$. Indeed, it is known that half-maximal supersymmetry in $D=4$ allows for two types of supersymmetry invariants with four derivatives: the first one is determined in terms of a holomorphic function of $S$, the second depends on the $G_{2k-2,6}$ moduli only, as described in (3.21), and both contribute to $F^4$ couplings [56]. The first term in (5.36) corresponds the first invariant, which also includes the $\mathcal{R}^2$ coupling (2.3), while the second was considered in [55], it is by construction exact at 1-loop and includes a four-derivative scalar couplings studied in [57].

The relative coefficient of the two invariants in (5.36) is in fact fixed by unitarity. Indeed, the logarithmic dependence of the one-loop amplitude with respect to the Mandelstam variables ($s_1 = s, s_2 = t, s_3 = u$) is determined by the 1-loop divergence of the four-photon supergravity amplitude [41]. Because the genus-one string theory amplitude $F^{(2k-2,6)}_{abcd}(\Phi, s_i)$ is finite in the ultraviolet, the corresponding supergravity amplitude pole in dimensional regularisation $D = 4 - 2\epsilon$ cancels by construction the pole of the coupling $F^{(2k-2,6)}_{abcd}(\Phi, \epsilon)$ regularised according to (3.32) (corresponding formally to $q = 6 + 2\epsilon$). Thus, in the low energy limit $-\ell_s^2 s_i \ll 1$ [12]

$$F^{(2k-2,6)}_{abcd}(\Phi,s_i) \sim F^{(2k-2,6)}_{abcd}(\Phi,\epsilon) + \frac{3(2k)}{(4\pi)^2}\left(\frac{1}{\epsilon} - \frac{1}{3}\sum_{i=1}^{3}\log(-\ell_s^2 s_i)\right)\delta_{(ab}\delta_{cd)} \tag{5.37}$$

$$\sim \hat{F}^{(2k-2,6)}_{abcd}(\Phi) - \frac{3}{8\pi^2}\log(S_2^k)\delta_{(ab}\delta_{cd)} - \frac{2k}{(4\pi)^2}\sum_{i=1}^{3}\log(-\ell_P^2 s_i)\delta_{(ab}\delta_{cd)} \,,$$

up to a fixed constant, where we used the relation $\frac{S_2}{2\pi}\ell_P^2 = \ell_s^2$ between Planck length and string length. Therefore, the relative coefficient of the two invariants in (5.36) is indeed such that the logarithm of $S_2$ in the coupling disappears in string frame, consistently with the fact that string amplitudes depend analytically on the string coupling constant when formulated in string frame [58].

The overal normalisation of the 4-photon amplitude can be determined from the 1-loop divergence as [41, 59] (with $t_8 f^4 = f_{\mu\nu}f^{\nu\sigma}f_{\sigma\rho}f^{\rho\mu} - \frac{1}{4}(f_{\mu\nu}f^{\mu\nu})^2$)

$$A_4(S,\Phi,s_i) = \frac{\kappa^4}{8}\left(\frac{3}{8\pi^2}\log(S_2^k|\Delta_k(S)|^2)\delta_{(ab}\delta_{cd)} - F^{(2k-2,6)}_{abcd}(\Phi,s_i)\right)t_8 F^a F^b F^c F^d \,. \tag{5.38}$$

---

[12]Recall that $2k-2$ is the number of vector multiplets in $D=4$.

More precisely, the 1PI effective action includes the local terms

$$
\begin{aligned}
S_4 = \int \mathrm{d}^4x \sqrt{-g}\Big( & \frac{1}{2\kappa^2}\mathcal{R} - \frac{S_2}{32\pi}(F^a_{\mu\nu}F^{\mu\nu}_a + F^{\hat{a}}_{\mu\nu}F^{\mu\nu}_{\hat{a}}) + \frac{S_1}{64\pi\sqrt{-g}}\varepsilon^{\mu\nu\rho\sigma}(F^a_{\mu\nu}F_{\rho\sigma\,a} - F^{\hat{a}}_{\mu\nu}F_{\rho\sigma\,\hat{a}}) \\
& + \frac{\kappa^4}{8}\Big(\frac{3}{8\pi^2}\log(S_2^k|\Delta_k(S)|^2)\delta_{(ab}\delta_{cd)} - \hat{F}^{(2k-2,6)}_{abcd}(\Phi)\Big)t^{\mu\nu\rho\sigma\kappa\lambda\vartheta\tau}\Big(\frac{S_2}{8\pi}\Big)^2 F^a_{\mu\nu}F^b_{\rho\sigma}F^c_{\kappa\lambda}F^d_{\vartheta\tau} \\
& - \frac{1}{(8\pi)^2}\log(S_2^k|\Delta_k(S)|^2)(\mathcal{R}_{\mu\nu\rho\sigma}\mathcal{R}^{\mu\nu\rho\sigma} - 4\mathcal{R}_{\mu\nu}\mathcal{R}^{\mu\nu} + \mathcal{R}^2) \\
& - \frac{\kappa^2}{(8\pi)^2}\mathcal{R}^{\mu\nu\rho\sigma}\Big(\mathcal{D}\log(S_2^k|\Delta_k(S)|^2)\frac{S_2}{8\pi}F^{\hat{a}-}_{\mu\nu}F^-_{\rho\sigma\hat{a}} + \overline{\mathcal{D}}\log(S_2^k|\Delta_k(S)|^2)\frac{S_2}{8\pi}F^{\hat{a}+}_{\mu\nu}F^+_{\rho\sigma\hat{a}}\Big) \\
& - \frac{\kappa^4}{(8\pi)^2}\mathcal{D}^2\log(S_2^k|\Delta_k(S)|^2)\Big(\frac{S_2}{8\pi}\Big)^2\Big(2F^{\hat{a}-}_{\mu\nu}F^-_{\rho\sigma\hat{a}}F^{\mu\nu}_{\hat{b}-}F^{\rho\sigma\hat{b}}_- + F^{\hat{a}-}_{\mu\nu}F^{\mu\nu}_{\hat{a}-}F^{\rho\sigma}_{\hat{b}-}F^{\hat{b}-}_{\rho\sigma}\Big) \\
& - \frac{\kappa^4}{(8\pi)^2}\overline{\mathcal{D}}^2\log(S_2^k|\Delta_k(S)|^2)\Big(\frac{S_2}{8\pi}\Big)^2\Big(2F^{\hat{a}+}_{\mu\nu}F^+_{\rho\sigma\hat{a}}F^{\mu\nu}_{\hat{b}+}F^{\rho\sigma\hat{b}}_+ + F^{\hat{a}+}_{\mu\nu}F^{\mu\nu}_{\hat{a}+}F^{\rho\sigma}_{\hat{b}+}F^{\hat{b}+}_{\rho\sigma}\Big) + \dots \Big),
\end{aligned}
\tag{5.39}
$$

which includes in particular the exact $\mathcal{R}^2$ coupling (2.3). The components of (5.10), (5.29) with $\mu, \nu$ indices correspond to scalar field parametrizing the circle radius $R$, the scalar field $\psi$ dual to the Kaluza–Klein vector, and the axiodilaton scalar field $S$ in four dimensions. The components involving the derivative of the function of $S$ depend on the complex (anti)selfdual field $F^{\hat{a}\pm}_{\mu\nu} \equiv \frac{1}{2}F^{\hat{a}}_{\mu\nu} \pm \frac{i}{4\sqrt{-g}}\varepsilon_{\mu\nu}{}^{\rho\sigma}F^{\hat{a}}_{\rho\sigma}$, with the covariant derivative $\mathcal{D}$ defined as in Appendix D with $\mathcal{D} \equiv \mathcal{D}_0$ and $\mathcal{D}^2 \equiv \mathcal{D}_2\mathcal{D}_0$.

Let us now discuss the decompactification limit of the 1PI effective action to ten dimensions, focussing for simplicity on the maximal rank case where the lattice decomposes as

$$
\Lambda_{22,6} = D_{16} \oplus I\!I_{6,6},
\tag{5.40}
$$

where $D_{16}$ is the weight lattice of $Spin(32)/\mathbb{Z}_2$. Identifying $S_2 = \frac{2\pi(2\pi R)^6}{g_s^2}$, with $g_s$ the heterotic string coupling constant in 10 dimensions, one obtains for $a, b, c, d$ along $D_{16}$,

$$
-\frac{3}{8\pi^2}\log(S_2^k|\Delta_k(S)|^2)\delta_{(ab}\delta_{cd)} + \hat{F}^{(2k-2,6)}_{abcd}(\Phi) = (2\pi R)^6\Big(\frac{3}{g_s^2}\delta_{(ab}\delta_{cd)} + \frac{1}{2\pi^5}\delta_{abcd}\Big) + \dots
\tag{5.41}
$$

up to a threshold contribution and exponentially suppressed terms. Here $\delta_{abcd} = 1$ if all indices are identical, and zero otherwise, and we used

$$
\begin{aligned}
\int_{SL(2,\mathbb{Z})\backslash\mathcal{H}} \frac{\mathrm{d}^2\tau}{\tau_2^2} \frac{\Gamma_{D_{16}}[P_{abcd}]}{\Delta} &= \int_{SL(2,\mathbb{Z})\backslash\mathcal{H}} \frac{\mathrm{d}^2\tau}{\tau_2^2}\Big[\Big(\frac{E_4^3 - 2\hat{E}_2 E_4 E_6 + \hat{E}_2^2 E_4^2}{48\Delta} - 24\Big)\delta_{(ab}\delta_{cd)} + 48\delta_{abcd}\Big] \\
&= 32\pi\delta_{abcd}.
\end{aligned}
\tag{5.42}
$$

This equation follows from known results about the elliptic genus of the heterotic string [60]. Using an orthogonal basis for a Cartan subalgebra of $SO(32)$, one easily computes that this coupling gives the following trace combination in the vector representation of $SO(32)$

$$
\Big(\frac{3}{g_s^2}\delta_{(ab}\delta_{cd)} + \frac{1}{2\pi^5}\delta_{abcd}\Big)t_8 F^a F^b F^c F^d = \frac{(2\pi R)^6}{4}t_8\Big(\frac{3}{g_s^2}(\mathrm{Tr}F^2)^2 + \frac{1}{\pi^5}\mathrm{Tr}F^4\Big).
\tag{5.43}
$$

Using $\kappa^2 = 4\alpha'$ and reabsorbing the $(2\pi R)^6 \alpha'^3$ into the 6-torus volume one obtains in Einstein frame

$$S_{10} = \int d^{10}x \sqrt{-g} \left( \frac{1}{8\alpha'^4} \mathcal{R} + \frac{1}{8\alpha'^3} e^{-\frac{1}{2}\phi} \left( \mathrm{Tr} F_{\mu\nu} F^{\mu\nu} + \mathcal{R}_{\mu\nu\rho\sigma} \mathcal{R}^{\mu\nu\rho\sigma} - 4\mathcal{R}_{\mu\nu} \mathcal{R}^{\mu\nu} + \mathcal{R}^2 \right) \right.$$
$$\left. - \frac{1}{2\alpha'} t_8 \left( 3e^{-\frac{3}{2}\phi} \mathrm{Tr} F^2 \mathrm{Tr} F^2 + \frac{1}{\pi^5} e^{\frac{1}{2}\phi} \mathrm{Tr} F^4 \right) + \dots \right), \quad (5.44)$$

which reproduces the tree level $\mathcal{R}^2$ and $(\mathrm{Tr} F^2)^2$ coupling computed in [53] upon identifying $\phi = \sqrt{2}\kappa D - 6\log 2$, and the 1-loop $\mathrm{Tr} F^4$ coupling computed in [61, 62].

### 5.3.2 BPS dyons

The contributions of non-zero vectors to the rank-one orbit yield exponentially suppressed corrections of order $e^{-2\pi R \mathcal{M}(Q,P)}$ (5.14), where $\mathcal{M}$ is the mass of a 1/2-BPS state of electromagnetic charge $(Q,P)$ in four dimensions. The phase $e^{2\pi i(a^1 Q + a^2 P)}$ multiplying (5.14) is the expected minimal coupling of a dyonic state with charge $(Q,P)$ to the holonomies of the electric and magnetic gauge fields along the circle. The corresponding instanton is a saddle point of the three-dimensional Euclidean supergravity theory obtained by formal reduction along a time-like Killing vector, in the duality frame where the axionic scalars $a_1, a_2$ are dualized into vector fields. Following the same steps as [63], one finds that the classical action is then $S_{\mathrm{cl}} = 2\pi R \mathcal{M}(Q,P)$.

In the maximal rank case, the summation measure (5.16) is given by

$$\bar{c}(Q,P) = \sum_{\substack{d \geq 1 \\ (Q,P)/d \in \Lambda_{em}}} c\left( -\frac{\gcd(Q^2, P^2, Q \cdot P)}{2d^2} \right), \quad (5.45)$$

where $c(m)$ are the Fourier coefficients of $1/\Delta$. For $(Q,P)$ primitive, this agrees with the helicity supertrace (2.18) of 1/2-BPS states with charges $(Q,P)$. In the case of CHL models, the summation measure is instead given by (5.31) or (5.32) with $q = 8$, $\tilde{v} = 1$, depending whether the dyon is related by $\Gamma_0(N)$, acting as $\binom{Q}{P} \to \binom{a \; b}{c \; d}\binom{Q}{P}$, to a purely electric or a purely magnetic state. It is interesting to note that these two formulas can be combined as follows. We first notice using the decomposition $(Q,P) = (j',p')\hat{Q}$ and $(Q,P) = (j',p')\hat{P}$ when $(Q,P)$ belong the electric and magnetic orbit respectively, with $(j',p') = 1$, one obtains

$$\frac{\hat{Q}}{d} \in \Lambda_m \Rightarrow \frac{(Q,P)}{d} \in \Lambda_m \oplus N\Lambda_m,$$
$$\frac{\hat{P}}{d} \in N\Lambda_e \Rightarrow \frac{(Q,P)}{d} \in N\Lambda_e \oplus N\Lambda_e, \quad (5.46)$$

such that in both cases $(Q,P)/d \in \Lambda_m \oplus N\Lambda_e$. Moreover, if $(Q,P)/d \in \Lambda_m \oplus N\Lambda_e$, then $\hat{Q}/d \in \Lambda_m$ or $\hat{P}/d \in N\Lambda_e$, depending of the orbit to which $(Q,P)$ belongs to, therefore one has the equivalence

$$\frac{(Q,P)}{d} \in \Lambda_m \oplus N\Lambda_e \iff \frac{\hat{Q}}{d} \in \Lambda_m \text{ or } \frac{\hat{P}}{d} \in N\Lambda_e, \quad (5.47)$$

for $(Q,P)$ conjugate to either an electric charge $\hat{Q}$ or a magnetic charge $\hat{P}$. Similarly,

$$\frac{\hat{Q}}{d} \in \Lambda_e \Rightarrow \frac{(Q,P)}{d} \in \Lambda_e \oplus N\Lambda_e,$$
$$\frac{\hat{P}}{d} \in \Lambda_m \Rightarrow \frac{(Q,P)}{d} \in \Lambda_m \oplus \Lambda_m, \quad (5.48)$$

such that

$$\frac{(Q,P)}{d} \in \Lambda_e \oplus \Lambda_m \quad \Leftrightarrow \quad \frac{\hat{Q}}{d} \in \Lambda_e \text{ or } \frac{\hat{P}}{d} \in \Lambda_m \,, \tag{5.49}$$

for $(Q,P)$ conjugate to either an purely electric charge $(\hat{Q},0)$ or a purely magnetic charge $(0,\hat{P})$. Moreover, we have that $\gcd(NQ^2, P^2, Q \cdot P) = N\hat{Q}^2$ for a dyon in the $\Gamma_0(N)$ orbit of a purely electric charge, because then $\gcd(Nj'^2, p'^2, j'p') = N$ since $p' = 0 \bmod N$, and $\gcd(NQ^2, P^2, Q \cdot P) = \hat{P}^2$ for a dyon in the $\Gamma_0(N)$ orbit of a purely magnetic charge, because then $\gcd(Nj'^2, p'^2, j'p') = 1$ since $p' \neq 0 \bmod N$. Putting these observations together we conclude that the summation measure for a general 1/2 BPS dyon is given by

$$\bar{c}_k(Q,P) = \sum_{\substack{d \geq 1 \\ (Q,P)/d \in \Lambda_e \oplus \Lambda_m}} c_k\Big(-\frac{\gcd(NQ^2, P^2, Q \cdot P)}{d^2}\Big) + \sum_{\substack{d \geq 1 \\ (Q,P)/d \in \Lambda_m \oplus N\Lambda_e}} c_k\Big(-\frac{\gcd(NQ^2, P^2, Q \cdot P)}{2Nd^2}\Big). \tag{5.50}$$

It is worth noting that $\gcd(NQ^2, P^2, Q \cdot P)$ is invariant under $\Gamma_0(N)$ and Fricke S-duality, so that each term in (5.50) is separately invariant under Fricke duality. Further noticing that $\Lambda_m \oplus N\Lambda_e \simeq \Lambda_e[N] \oplus \Lambda_m[N]$, (5.50) can be rewritten in a more suggestive way as

$$\bar{c}_k(Q,P) = \sum_{a|N} \sum_{\substack{d \geq 1 \\ (Q,P)/d \in \Lambda_{em}[a]}} c_k\Big(-\frac{\gcd(NQ^2, P^2, Q \cdot P)}{2a\,d^2}\Big). \tag{5.51}$$

Most importantly, (5.51) agrees with the helicity supertrace $\Omega_4(Q,P)$ of a half-BPS dyon with primitive charge $(Q,P)$ which was determined in (2.16) and (2.17).

### 5.3.3 Taub-NUT instantons

Finally, the rank-two orbit (5.25) yields contributions schematically of the form

$$\sum_{M_1 \neq 0, M_2, P} \bar{c}(M_1, M_2, P)\, e^{-2\pi\sqrt{\left(R^2 M_1 + S_2 \tilde{M}_2\right)^2 + 2R^2 S_2 \tilde{P}_R^2} + 2\pi i(P \cdot a_2 + M_1(\psi - \frac{1}{2}a_1 \cdot a_2) + \tilde{M}_2 S_1)} \,, \tag{5.52}$$

where the summation measure (5.33) is given by

$$\bar{c}_k(M_1, M_2, P) = \sum_{\substack{d|(M_1,M_2) \\ P/d \in \Lambda_m}} d\, c_k\Big(\frac{D}{d^2}\Big) + \sum_{\substack{Nd|(M_1,M_2) \\ P/d \in N\Lambda_e}} Nd\, c_k\Big(\frac{D}{Nd^2}\Big), \tag{5.53}$$

and we denoted $\tilde{M}_2 = M_2 - a_1 \cdot P + \frac{1}{2}(a_1 \cdot a_1)M_1$, $\tilde{P} = P - M_1 a_1$, and $D = -\frac{1}{2}P^2 + M_1 M_2$. These $\mathcal{O}(e^{-2\pi R^2 |M_1|})$ contributions are characteristic of an Euclidean Taub-NUT solution of the form $\text{TN}_{M_1} \times T^6$, where the Taub-NUT space asymptotes to $\mathbb{R}^3 \times S_1(R)$ at spatial infinity [64].

The detailed semi-classical interpretation of these effects is complicated by the fact that in a Taub-NUT background, similarly to the case of NS5-branes, large gauge transformations of the electric and magnetic holonomies $a_1$ and $a_2$ do not commute, thus cannot be diagonalized simultaneously. The representation (5.20) corresponds to the case where translations in $a_2$ and $\psi$ are diagonalized. Accordingly, the argument of the exponential in (5.52) should be interpreted as the classical action in the duality frame in which the fields $\psi, S_1, a_2$ associated to the conserved charges $M_1, M_2$ and $P$ are dualized into vector fields $\omega, B, A$ in three dimensions. In order to reach a positive definite action after dualization, one should first analytically continue the non-linear

sigma model on $\frac{O(2k,8)}{O(2k)\times O(8)}$ into $\frac{O(2k,8)}{O(2k-1,1)\times O(7,1)}$ by taking $\psi, S_1, a_2$ to be purely imaginary. Equivalently, this is the non-linear sigma model obtained by reduction of a Euclidean four-dimensional theory. Denoting by $U, \phi, \zeta$ the scalar fields whose asymptotic values are given by $\log R, -\frac{1}{2}\log S_2$ and $a_1$, the Lagrange density in three dimensions is

$$
\begin{aligned}
\mathcal{L} =& |\mathrm{d}U|^2 + \frac{1}{4}e^{4U}|\mathrm{d}\omega|^2 + |\mathrm{d}\phi|^2 + \frac{1}{4}e^{-4\phi}|\mathrm{d}B - (\zeta,\mathrm{d}A) + \tfrac{1}{2}(\zeta,\zeta)\mathrm{d}\omega|^2 \\
&+ \frac{1}{4}e^{2U-2\phi}g(\mathrm{d}A - \zeta\mathrm{d}\omega, \mathrm{d}A - \zeta\mathrm{d}\omega) + \frac{1}{4}e^{-2U-2\phi}g(\mathrm{d}\zeta,\mathrm{d}\zeta) + P_{a\hat{b}}\star P^{a\hat{b}} ,
\end{aligned}
\tag{5.54}
$$

where we denote $|f|^2 = f \wedge \star f$, $g(F,F) \equiv F_L^a \star F_{La} + F_R^{\hat{a}} \star F_{R\hat{a}}$. For simplicity we shall consider only instantons for which the electromagnetic fields vanish, $\mathrm{d}A = \zeta = 0$. One can then write the Lagrangian as a sum of squares

$$
\mathcal{L} = \frac{1}{4}e^{4U}\left|\star\mathrm{d}e^{-2U} \pm \mathrm{d}\omega\right|^2 \pm \frac{1}{2}\mathrm{d}(e^{2U}\mathrm{d}\omega) + \frac{1}{4}e^{-4\phi}\left|\star\mathrm{d}e^{2\phi} \pm \mathrm{d}B\right|^2 \pm \frac{1}{2}\mathrm{d}(e^{-2\phi}\mathrm{d}B) + P_{a\hat{b}}\star P^{a\hat{b}} . \tag{5.55}
$$

The corresponding 1/2-BPS solutions describe $M_2$ Euclidean NS5-branes on a self-dual Taub-NUT space of charge $M_1$, with $M_1 M_2 \geq 0$.[13] For simplicity we consider the NS5-branes at the tip of the Taub-NUT space, with

$$
e^{-2U} = \frac{1}{R^2} + \frac{|M_1|}{r} , \quad e^{2\phi} = \frac{1}{S_2} + \frac{|M_2|}{r} , \quad \omega = -M_1 \cos\theta\,\mathrm{d}\varphi , \quad B = -M_2\cos\theta\,\mathrm{d}\varphi , \tag{5.56}
$$

and the fields $\Phi$ on the Grassmannian $G_{r-6,6}$ are uniform. The action then reduces to the boundary term $S_{\mathrm{cl}} = 2\pi(R^2|M_1| + S_2|M_2|) = 2\pi|R^2 M_1 + S_2 M_2|$. Note that the measure factor (5.53) vanishes for $P = 0$ unless $M_1 M_2 \geq -1$. We shall refrain from constructing 1/2-BPS instantons with generic magnetic charge $P$ such that $D \geq 0$, although we expect that their action will reproduce $S_{\mathrm{cl}}$ in (5.22).

# 6  Discussion

In this work, we have proposed a formula (2.24) for the exact $(\nabla\Phi)^4$ coupling in a class of three-dimensional string vacua obtained as freely acting orbifolds of the heterotic string on $T^7$ under a $\mathbb{Z}_N$ action with $N$ prime. Our formula is manifestly invariant under the U-duality group $G_3(\mathbb{Z})$, which unifies the S and T-duality in $D = 4$ along with Fricke duality. We derived the supersymmetric Ward identities that the exact coupling function $F_{abcd}(\Phi)$ must satisfy, and showed that the formula (2.24) satisfies this constraint. Furthermore, we analyzed its behavior in the weak coupling regime $g_3 \to 0$ and large radius regime $R \to \infty$, and found that it correctly reproduces the known tree-level and one-loop contributions in $D = 3$, and the correct non-perturbative $F^4$ couplings in $D = 4$. In addition, we extracted the exponential corrections to these power-like terms in both regimes, corresponding to non-zero Fourier coefficients with respect to parabolic subgroups $\mathbb{R}^+ \times G_{2k-1,7} \ltimes \mathbb{R}^{2k+6}$ and $\mathbb{R}^+ \times [SL(2)/SO(2) \times G_{2k-2,6}] \ltimes \mathbb{R}^{2\times(2k+4)} \times \mathbb{R}$, and found agreement with the expected form of the contributions of NS5-brane, Kaluza–Klein monopoles and H-monopole instantons as $g_3 \to 0$, and the contributions of half-BPS dyons and Taub-NUT instantons as $R \to \infty$. In the case of half-BPS dyons, we found a precise match between the

---

[13]Solutions with $M_1 M_2 \leq 0$ exist but do not preserve eight supercharges.

summation measure $\bar{c}_k(Q, P)$ and the helicity supertrace $\Omega_4(Q, P)$, at least when the charge vector $(Q, P)$ is primitive. This vindicates the general expectation that BPS saturated couplings in dimension $D$ encode BPS indices in dimension $D + 1$. It would be interesting to determine the helicity supertrace $\Omega_4(Q, P)$ when $(Q, P)$ is not primitive (which requires a careful treatment of threshold bound states), and compare with the summation measure $\bar{c}_k(Q, P)$.

It is natural to ask whether our formula (2.27) is the unique solution to the Ward identities (2.23) which is invariant under $G_3(\mathbb{Z})$, and reproduces the correct power-like terms in the weak coupling and large radius expansions $g_3 \to 0$ and $R \to \infty$. Typically, theorems in the mathematical literature guarantee that smooth automorphic forms on $K\backslash G/G(\mathbb{Z})$ which vanish at all cusps and have sufficiently sparse Fourier coefficients (in mathematical terms, are attached to a sufficiently small nilpotent orbit) necessarily vanish; so that the only smooth automorphic functions satisfying to (3.27) are necessary Eisenstein series. However, these theorems are typically concerned with Chevalley subgroups of reductive groups in the split or quasi-split real form, which is not the case here ($G_3(\mathbb{Z})$ is a proper subgroup of the Chevalley group of $O(2k, 8)$ for $N > 1$), and smoothness away from the cusps is essential.

As far as the support of Fourier coefficients is concerned, the Ward identities (3.27), imply that the trace of the modular integral (3.29) is attached to the vectorial character of $O(p, q)$, corresponding to the next-to-minimal orbit. However, the constraints imposed by the differential equations (3.17), (3.20) are stronger than (3.27), e.g. we show in Appendix B that the tensor $F_{abcd}$ derived from the scalar Eisenstein series defined in Appendix E.2 is not a solution to (3.20). The general form of the Fourier coefficients is in fact very reminiscent of the one for automorphic forms attached to the minimal orbit of $O(p, q)$: it allows for only two power-like terms at the cusp, rather than three for the next-to-minimal orbit; they involve ordinary Bessel function of one single variable, similarly to $A_1$ Whittaker vectors, rather than more complicated functions of two variables or the typical $2A_1$ Whittaker vectors which appear in the Fourier coefficients of generic vectorial Eisenstein series [65].

However, as we emphasized repeatedly, (3.28) has singularities in the bulk of $G_{p,q}$ on codimension $q$ loci where the projection $P_R^{\hat{a}}$ of a vector $P$ in $\Lambda_{p,q}$ with norm 2 (or the projection $Q_R^{\hat{a}}$ of a vector $Q$ in $\Lambda_{p,q}^*$ with norm $2/N$) vanishes. In order to argue for uniqueness, it is crucial to ensure that the modular integral (2.24) correctly captures the behavior of the $(\nabla\Phi)^4$ coupling at all singular loci. Since (2.24) reproduces correctly the one-loop contribution to $(\nabla\Phi)^4$, it is clear that it correctly captures the singular behavior on the loci associated to vectors $P, Q$ in the 'perturbative Narain lattice' $\Lambda_{r-5,7} \subset \Lambda_{r-4,8}$, at least in the weak coupling limit. Presumably, this suffices to guarantee agreement on all singular loci, but we do not know how to prove this rigorously.

Let us note finally that, independently of our proposed identification of the $U$-duality group in three dimensions, the general solution to the Ward identities (3.17), (3.20) derived in Appendix B implies that the exact coupling must be of the form (4.35), up to the determination of the measure factor $\bar{c}_k(Q)$. The property that we recover the exact coupling in four dimensions implies that the mesure factor is correct for null vectors by $O(r - 5, 7, \mathbb{Z})$ T-duality. Indeed, for $Q^2 = 0$, the summation measure in (4.36) reproduces the summation measure for NS5-brane instantons in (2.5). The computation of the BPS index associated to an arbitrary NS5-brane, Kaluza–Klein monopole, H-monopole instanton, would therefore give a direct proof of our result.

Clearly, it would be interesting to generalize our construction to the complete class of heterotic CHL models, whose duality properties and BPS spectrum in 4-dimensions are by now well understood. It is natural to conjecture that the duality group in $D = 3$ will still be given by the automorphism group of the non-perturbative Narain lattice (2.21), which naturally incorporates the

S and T-duality symmetries in $D = 4$. More pressingly however, the present study was a warm-up towards the more challenging problem of understanding the 1/4-BPS saturated coupling $\nabla^2(\nabla\Phi)^4$ in four dimensions, which we shall address in forthcoming work.

## Acknowledgements

We are grateful to Hervé Partouche and especially Roberto Volpato for useful discussions and correspondence. CCH and GB are grateful to CERN Theory Department for hospitality during part of this project.

# A  Perturbative spectrum and one-loop $F^4$ couplings in heterotic CHL orbifolds

In this section, we construct the one-loop vacuum amplitude in CHL models obtained as a freely acting $Z_N$-orbifold of the standard heterotic string on $T^d$ with $N$ prime. From this, we deduce the helicity supertrace for perturbative BPS states, and the one-loop contribution to the $F^4$ and $(\nabla\Phi)^4$ couplings. We start with the simplest model with $N = 2$, and then generalize the construction to $N = 3, 5, 7$.

## A.1  $\mathbb{Z}_2$ orbifold

The simplest CHL model is obtained by orbifolding the $E_8 \times E_8$ heterotic string compactified on $T^d$, by an involution $\sigma$ which exchanges the two $E_8$ gauge groups and performs a translation by half a period along one circle in $T^d$ [14]. This perturbative BPS spectrum in this model was further studied in [66, 26]. The symmetry $\sigma$ exists only on a codimension $8d$ space inside the Narain moduli space $G_{d+16,d}$ and preserves only a $U(1)^{2d+8}$ subgroup of the original $U(1)^{2d+16}$ gauge symmetry, corresponding to the usual $2d$ Kaluza–Klein and Kalb-Ramond gauge fields, and the Cartan torus of the diagonal combination of the two $E_8$ gauge groups. To implement the quotient by $\sigma$, it is simplest to work at the point in $G_{d+16,d}$ where the lattice factorizes as

$$\Lambda_{d+16,d} = E_8 \oplus E_8 \oplus I\!I_{d,d} \ . \tag{A.1}$$

The integrand of the one-loop vacuum amplitude of the original heterotic string is then

$$\mathcal{A} = Z_{E_8 \times E_8} \times \Gamma_{I\!I_{d,d}} \times \frac{1}{2} \sum_{\alpha,\beta \in \{0,1\}} (-1)^{\alpha\beta+\alpha+\beta} \frac{\overline{\vartheta}^4 \begin{bmatrix} \alpha \\ \beta \end{bmatrix}}{\tau_2^4 \eta^8 \overline{\eta}^{12}} \ , \tag{A.2}$$

where

$$Z_{E_8 \times E_8} = \left[ \frac{\sum_{Q_1 \in E_8} q^{\frac{1}{2}Q_1^2}}{\eta^8} \right] \left[ \frac{\sum_{Q_2 \in E_8} q^{\frac{1}{2}Q_2^2}}{\eta^8} \right] = \frac{[E_4(\tau)]^2}{\eta^{16}} \tag{A.3}$$

is the partition function of the 16 chiral bosons on the $E_8 \times E_8$ root lattice, and the last factor in (A.2) represents the contribution of the transverse bosonic and fermionic oscillators, while the sum over $\alpha, \beta$ implements the GSO projection. As a consequence of space-time supersymmetry, the integral

(A.2) vanishes pointwise, but it will no longer be so in the presence of vertex operators. Note that the right-moving part in (A.2) will not play any role in our case, and will be later replaced by an insertion of the polynomial $P_{abcd}$ (2.26).

Following standard rules, the one-loop partition function of the orbifold by $\sigma$ is obtained by replacing $\mathcal{A}$ by a sum $\frac{1}{2}\sum_{h,g\in\{0,1\}}\mathcal{A}\big[{}^h_g\big]$, where $\mathcal{A}\big[{}^h_g\big]$ is obtained by twisting the boundary conditions of the fields by $\sigma^g$ along the spatial direction of the string, and $\sigma^h$ along the Euclidean time direction, so that $\frac{1}{2}(\mathcal{A}\big[{}^0_0\big]+\mathcal{A}\big[{}^0_1\big])$ counts $\sigma$-invariant states in the untwisted sector, while $\frac{1}{2}(\mathcal{A}\big[{}^1_0\big]+\mathcal{A}\big[{}^1_1\big])$ counts $\sigma$-invariant states in the twisted sector. Modular invariance permutes the three blocks $\big[{}^0_1\big],\big[{}^1_0\big],\big[{}^1_1\big]$ according to

$$\mathcal{A}\big[{}^h_g\big]\Big(\tfrac{a\tau+b}{c\tau+d}\Big)=\mathcal{A}\big[{}^{ah+cg}_{bh+gd}\big](\tau)\,,\tag{A.4}$$

where $h,g$ are treated modulo 2. In particular, the block $\big[{}^0_1\big]$ is invariant under the Hecke congruence subgroup $\Gamma_0(2)$, and all other blocks can be obtained by acting on it with elements of $SL(2,\mathbb{Z})/\Gamma_0(2)=\{1,S,ST\}$.

In the case at hand, the involution $\sigma$ exchanges $Q_1\leftrightarrow Q_2$ and the corresponding oscillators, so $\sigma$-invariant states must have $Q_1=Q_2$ and the same oscillator state on both factors, thus

$$Z_{E_8\times E_8}\big[{}^0_1\big](\tau)=\frac{\sum_{Q\in E_8}q^{Q^2}}{\eta^8(2\tau)}\,.\tag{A.5}$$

The two remaining orbifold blocks are then fixed by modular covariance,

$$Z_{E_8\times E_8}\big[{}^0_0\big]=\frac{E_4^2(\tau)}{\eta^{16}(\tau)}\,,\quad Z_{E_8\times E_8}\big[{}^0_1\big]=\frac{E_4(2\tau)}{\eta^8(2\tau)}\,,$$
$$Z_{E_8\times E_8}\big[{}^1_0\big]=\frac{E_4(\frac{\tau}{2})}{\eta^8(\frac{\tau}{2})}\,,\quad Z_{E_8\times E_8}\big[{}^1_1\big]=\frac{E_4(\frac{\tau+1}{2})}{e^{2i\pi/3}\eta^8(\frac{\tau+1}{2})}\,,\tag{A.6}$$

As for the action of $\sigma$ on the torus $T^d$, it can be taken into account by replacing the partition function $\Gamma_{I_{d,d}}$ by

$$\Gamma_{I_{d,d}}\big[{}^h_g\big]=\tau_2^{d/2}\sum_{Q\in I_{d,d}+\frac{h}{2}\delta}(-1)^{g\,\delta\cdot Q}q^{\frac{1}{2}Q_L^2}\bar{q}^{\frac{1}{2}Q_R^2}\,.\tag{A.7}$$

where $\delta$ must be null modulo 2, and depends on the choice of circle $S_1$ inside $T^d$. The resulting one-loop vacuum amplitude is then the modular integral of

$$\mathcal{A}_{\text{orb}}=\frac{1}{2}\sum_{h,g\in\{0,1\}}Z_{E_8\times E_8}\big[{}^h_g\big]\Gamma_{I_{d,d}}\big[{}^h_g\big]\times\frac{1}{2}\sum_{\alpha,\beta\in\{0,1\}}(-1)^{\alpha\beta+\alpha+\beta}\frac{\bar{\vartheta}^4\big[{}^\alpha_\beta\big]}{\tau_2^4\eta^8\bar{\eta}^{12}}\,,\tag{A.8}$$

where the one-half factor is explained above (A.4). Now, a key observation is that the numerator in the blocks $Z_{E_8\times E_8}\big[{}^h_g\big]$ for $(h,g)\neq(0,0)$ can be written as a partition functions for the lattice

$\Lambda = E_8[2]$ and for its dual $\Lambda^* = E_8[1/2]$,

$$Z_{E_8 \times E_8}\begin{bmatrix} 0 \\ 1 \end{bmatrix} = \frac{1}{\eta^8(2\tau)} \sum_{Q \in E_8[2]} q^{\frac{1}{2}Q^2}$$

$$Z_{E_8 \times E_8}\begin{bmatrix} 1 \\ 0 \end{bmatrix} = \frac{1}{\eta^8(\frac{\tau}{2})} \sum_{Q \in E_8[1/2]} q^{\frac{1}{2}Q^2} \tag{A.9}$$

$$Z_{E_8 \times E_8}\begin{bmatrix} 1 \\ 1 \end{bmatrix} = \frac{1}{e^{2i\pi/3}\eta^8(\frac{\tau+1}{2})} \sum_{Q \in E_8[1/2]} (-1)^{Q^2} q^{\frac{1}{2}Q^2} \, .$$

Moreover, the untwisted, unprojected partition function satisfies

$$Z_{E_8 \times E_8}\begin{bmatrix} 0 \\ 0 \end{bmatrix} = \frac{E_4(2\tau)}{\eta^8(2\tau)} + \frac{E_4(\frac{\tau}{2})}{\eta^8(\frac{\tau}{2})} + \frac{E_4(\frac{\tau+1}{2})}{e^{2i\pi/3}\eta^8(\frac{\tau+1}{2})}$$

$$= Z_{E_8 \times E_8}\begin{bmatrix} 0 \\ 1 \end{bmatrix} + Z_{E_8 \times E_8}\begin{bmatrix} 1 \\ 0 \end{bmatrix} + Z_{E_8 \times E_8}\begin{bmatrix} 1 \\ 1 \end{bmatrix} \, . \tag{A.10}$$

This relation can be checked using the explicit form of the blocks $Z_{E_8 \times E_8}\begin{bmatrix} 0 \\ 1 \end{bmatrix}$, but more conceptually, it follows by decomposing $Z_{E_8 \times E_8}\begin{bmatrix} 0 \\ 0 \end{bmatrix}$, the character of the level 1 representation of $\hat{E}_8 \oplus \hat{E}_8$, into characters of level 2 representations of the diagonal $\hat{E}_8$ [67]. It follows from (A.9), (A.10) that the one-loop amplitude (A.8) can be written as

$$\mathcal{A}_{\text{orb}} = \frac{1}{2} \sum_{h,g \in \{0,1\}}' \frac{\widetilde{\Gamma}_{d+8,d}\begin{bmatrix} h \\ g \end{bmatrix}}{\Delta_8\begin{bmatrix} h \\ g \end{bmatrix}} \times \frac{1}{2} \sum_{\alpha,\beta \in \{0,1\}} (-1)^{\alpha\beta+\alpha+\beta} \frac{\overline{\vartheta}^4\begin{bmatrix} \alpha \\ \beta \end{bmatrix}}{\tau_2^4 \overline{\eta}^{12}} \tag{A.11}$$

where the sum over $(h,g)$ no longer includes $(0,0)$. Here, we defined the eta products

$$\Delta_8\begin{bmatrix} 0 \\ 1 \end{bmatrix} = \eta^8(\tau)\eta^8(2\tau) = 2^{-4}\eta^{12}\vartheta_2^4 \equiv \Delta_8(\tau)$$

$$\Delta_8\begin{bmatrix} 1 \\ 0 \end{bmatrix} = \eta^8(\tau)\eta^8(\tfrac{\tau}{2}) = \eta^{12}\vartheta_4^4 = \Delta_8(\tfrac{\tau}{2}), \tag{A.12}$$

$$\Delta_8\begin{bmatrix} 1 \\ 1 \end{bmatrix} = e^{2i\pi/3}\eta^8(\tau)\eta^8(\tfrac{\tau+1}{2}) = -\eta^{12}\vartheta_3^4 = \Delta_8(\tfrac{\tau+1}{2}), \, ,$$

satisfying

$$\Delta_8\begin{bmatrix} 0 \\ 1 \end{bmatrix}(-1/\tau) = 2^{-4}\tau^8 \Delta_8\begin{bmatrix} 1 \\ 0 \end{bmatrix}(\tau), \quad \Delta_8\begin{bmatrix} 1 \\ 0 \end{bmatrix}(\tau+1) = \Delta_8\begin{bmatrix} 1 \\ 1 \end{bmatrix}(\tau), \tag{A.13}$$

and the partition functions $\widetilde{\Gamma}_{d+8,d}$ are defined over $\tilde{\Lambda}_{d+8,d} = E_8[2] \oplus II_{d,d}$ and its dual $\tilde{\Lambda}^*_{d+8,d} = E_8[1/2] \oplus II_{d,d}$, as:

$$\widetilde{\Gamma}_{d+8,d}\begin{bmatrix} 0 \\ 1 \end{bmatrix} = \tau_2^{d/2} \sum_{Q \in \tilde{\Lambda}_{d+8,d}} \left[ 1 + (-1)^{\delta \cdot Q} \right] q^{\frac{1}{2}Q_L^2} \bar{q}^{\frac{1}{2}Q_R^2}$$

$$\widetilde{\Gamma}_{d+8,d}\begin{bmatrix} 1 \\ 0 \end{bmatrix} = \tau_2^{d/2} \left[ \sum_{Q \in \tilde{\Lambda}^*_{d+8,d}} + \sum_{Q \in \tilde{\Lambda}^*_{d+8,d}+\frac{1}{2}\delta} \right] q^{\frac{1}{2}Q_L^2} \bar{q}^{\frac{1}{2}Q_R^2} \tag{A.14}$$

$$\widetilde{\Gamma}_{d+8,d}\begin{bmatrix} 1 \\ 1 \end{bmatrix} = \tau_2^{d/2} \left[ \sum_{Q \in \tilde{\Lambda}^*_{d+8,d}} + \sum_{Q \in \tilde{\Lambda}^*_{d+8,d}+\frac{1}{2}\delta} \right] (-1)^{Q^2} q^{\frac{1}{2}Q_L^2} \bar{q}^{\frac{1}{2}Q_R^2}$$

These relations were derived at the special point where the lattice $\tilde{\Lambda}_{d+8,d}$ is factorized, but it is now clear that they hold at arbitrary points on the moduli space $G_{d+8,d} \subset G_{d+16,d}$ where the $\mathbb{Z}_2$ symmetry exists.

Choosing $\delta = (0^d; 0^{d-1}, 1)$, so that the involution $\sigma$ acts by a translation along the $d$-th circle by a half period, this can be further written as

$$\Gamma_{\Lambda_{d+8,d}} \equiv \tau_2^{d/2} \sum_{Q \in \Lambda_{d+8,d}} q^{\frac{1}{2}Q_L^2} \bar{q}^{\frac{1}{2}Q_R^2} = \frac{1}{2}\widetilde{\Gamma}_{d+8,d}\begin{bmatrix} 0 \\ 1 \end{bmatrix}$$

$$(2^5/\tau^4)\Gamma_{\Lambda_{d+8,d}}(-1/\tau) = \Gamma_{\Lambda_{d+8,d}^*} \equiv \tau_2^{d/2} \sum_{Q \in \Lambda_{d+8,d}^*} q^{\frac{1}{2}Q_L^2} \bar{q}^{\frac{1}{2}Q_R^2} = \widetilde{\Gamma}_{d+8,d}\begin{bmatrix} 1 \\ 0 \end{bmatrix} \tag{A.15}$$

$$\Gamma_{\Lambda_{d+8,d}^*}\left[(-1)^{Q^2}\right] \equiv \tau_2^{d/2} \sum_{Q \in \Lambda_{d+8,d}^*} (-1)^{Q^2} q^{\frac{1}{2}Q_L^2} \bar{q}^{\frac{1}{2}Q_R^2} = \widetilde{\Gamma}_{d+8,d}\begin{bmatrix} 1 \\ 1 \end{bmatrix}$$

where $\Lambda_{d+8,d}$ is related to $\tilde{\Lambda}_{d+8,d}$ by rescaling a $II_{1,1}$ summand[14],

$$\Lambda_{d+8,d} = E_8[2] \oplus II_{1,1}[2] \oplus II_{d-1,d-1} . \tag{A.16}$$

Here $II_{1,1}[2]$ is the usual sum over momentum $m_d$ and winding $n_d$, with $m_d$ running only over even integers. The dual lattice is

$$\Lambda_{d+8,d}^* = E_8[1/2] \oplus II_{1,1}[1/2] \oplus II_{d-1,d-1} , \tag{A.17}$$

where $II_{1,1}[1/2]$ is the usual sum over momentum $m_d$ and winding $n_d$, with $n_d$ running over $\mathbb{Z}/2$. For $d = 6$, since $\Lambda_{14,6} \subset \Lambda_{14,6}^*$, we see that the electric charges carried by excitations of the heterotic string lie in the lattice $\Lambda_e = \Lambda_{14,6}^*$, in agreement with the result stated in Table 1. Moreover, it is apparent that the degeneracy of perturbative BPS states with charge $Q \in \Lambda_{d+8,d}^*$, $Q \notin \Lambda_{d+8,d}$ in the twisted sector is given by the coefficient of $q^{-Q^2/2}$ in $1/\Delta_8\begin{bmatrix} 1 \\ 0 \end{bmatrix} = 1/\Delta_8(\tau/2)$, or equivalently the coefficient of $q^{-Q^2}$ in $1/\Delta_8$, while the degeneracy of perturbative BPS states with charge $Q \in \Lambda_{d+8,d} \subset \Lambda_{d+8,d}^*$ has an additional contribution from the coefficient of $q^{-Q^2/2}$ in $1/\Delta_8$, in agreement with (2.14) and (2.15), and the analysis in [66, 26].

At last, we can turn to the one-loop $F^4$ amplitude in this model. As is the case in the usual heterotic string, the insertion of four vertex operators replaces the right-moving contribution in the vacuum amplitude (A.11) by an insertion of the polynomial $P_{abcd}$ in (2.26). Thus, we get

$$F_{abcd}^{(1\text{-loop})} = \text{R.N.} \int_{SL(2,\mathbb{Z})\backslash\mathcal{H}} \frac{d\tau_1 d\tau_2}{\tau_2^2} \sum_{\gamma \in \Gamma_0(2)\backslash SL(2,\mathbb{Z})} \left.\frac{\Gamma_{\Lambda_{d+8,d}}[P_{abcd}]}{\Delta_8}\right|_\gamma , \tag{A.18}$$

where $\Gamma_{\Lambda_{d+8,d}}[P_{abcd}]$ denotes the lattice partition function $\Gamma_{\Lambda_{d+8,d}}\begin{bmatrix} h \\ g \end{bmatrix}$ with an insertion of the polynomial $P$ as in (2.25). Equivalently, we can unfold the integral over a fundamental domain $\Gamma_0(2)\backslash\mathcal{H}$ for the action of $\Gamma_0(2)$ on $\mathcal{H}$, at the expense of keeping only the identity in the sum over cosets,

$$F_{abcd}^{(1\text{-loop})} = \text{R.N.} \int_{\Gamma_0(2)\backslash\mathcal{H}} \frac{d\tau_1 d\tau_2}{\tau_2^2} \frac{\Gamma_{\Lambda_{d+8,d}}[P_{abcd}]}{\Delta_8} , \tag{A.19}$$

which demonstrates (2.24) in this case.

---

[14]Note that this rescaling implies an extra volume factor upon Poisson resummation, namely $\Gamma_{\Lambda_{d+8,d}^*}(\tau) = (2^5/\tau^4)\Gamma_{\Lambda_{d+8,d}}(-1/\tau)$.

## A.2 $\mathbb{Z}_N$ orbifold with $N = 3, 5, 7$

The construction detailed in the previous section can be easily generalized to $\mathbb{Z}_N$ orbifolds, provided one can find a point in the moduli space $G_{d+16,d}$ where $\mathbb{Z}_N$ acts on the lattice $\Lambda_{d+16,d}$ by a permutation with cycle shape $1^k N^k$. It turns out that for $N = 3, 5, 7$, such a lattice can be obtained by applying a Wick rotation on the Niemeier lattices $D_6^4$, $D_4^6$ and $D_3^8$, respectively. Indeed, recall that given an even self-dual Euclidean lattice

$$\Lambda = \cup_{(\lambda, \lambda') \in \mathcal{G}} (D_k + \lambda) \oplus (\Lambda' + \lambda') \tag{A.20}$$

of dimension $n$, where the glue code $\mathcal{G}$ is a given sublattice of $D_k^* / D_k \oplus \Lambda'^* / \Lambda'$, one can obtain an even self-dual lattice of dimension $n - 8$, by replacing $D_k$ by $D_{k-8}$, while keeping the same glue code $\mathcal{G}$, using the fact that $\mathcal{G}_k = D_k^* / D_k$ is invariant under $k \mapsto k - 8$[15],

$$\hat{\Lambda} = \cup_{(\lambda, \lambda') \in \mathcal{G}} (D_{k-8} + \lambda) \oplus (\Lambda' + \lambda') . \tag{A.21}$$

If $1 \leq k < 8$, then $D_{k-8}$ should be understood as $D_{8-k}[-1]$, so that the new lattice is a Lorentzian lattice with signature $(n - k, 8 - k)$ [68, §A.4]. In this way, starting from the Niemeier lattice $\Lambda = D_k^{N+1}$ for $N = 3, 5, 7$, which is symmetric under cyclic permutations of the $N+1$ $D_k$ factors, we obtain an even self-dual lattice $\hat{\Lambda} = D_k^N \oplus D_{8-k}[-1]$ of signature $(Nk, 8 - k)$ with a $\mathbb{Z}_N$ symmetry $\sigma$ acting by cyclic permutations of the $N$ $D_k$ factors. Using the explicit description of the glue code for Niemeier lattices given in [69, Table 16.1], it is possible to check that the only elements $(\lambda_1, \ldots \lambda_{N+1})$ in the glue code $\mathcal{G} \subset \mathcal{G}_k^{N+1}$ which are invariant under $\mathbb{Z}_N$ are those of the form $(\lambda, \ldots, \lambda)$ with $\lambda$ running over $\mathcal{G}_k$. The partition function of the lattice $\hat{\Lambda}$ with an insertion of the element $\sigma^g$ with $g \neq 0 \bmod N$ is thus

$$Z_{\hat{\Lambda}}\begin{bmatrix} 0 \\ g \end{bmatrix} = \frac{\vartheta_3^k + \vartheta_4^k}{2\eta^k}(N\tau)\overline{\frac{\vartheta_3^{8-k} + \vartheta_4^{8-k}}{2\eta^{8-k}}} + \frac{\vartheta_3^k - \vartheta_4^k}{2\eta^k}(N\tau)\overline{\frac{\vartheta_3^{8-k} - \vartheta_4^{8-k}}{2\eta^{8-k}}}$$
$$+ \frac{\vartheta_2^k + \vartheta_1^k}{2\eta^k}(N\tau)\overline{\frac{\vartheta_2^{8-k} + \vartheta_1^{8-k}}{2\eta^{8-k}}} + \frac{\vartheta_2^k - \vartheta_1^k}{2\eta^k}(N\tau)\overline{\frac{\vartheta_2^{8-k} - \vartheta_1^{8-k}}{2\eta^{8-k}}} . \tag{A.22}$$

The other blocks are obtained by modular covariance, leading for $h \neq 0 \bmod N$ to

$$Z_{\hat{\Lambda}}\begin{bmatrix} h \\ 0 \end{bmatrix} = \frac{\vartheta_3^k + \vartheta_2^k}{2\eta^k}\left(\frac{\tau}{N}\right)\overline{\frac{\vartheta_3^{8-k} + \vartheta_2^{8-k}}{2\eta^{8-k}}} + \frac{\vartheta_3^k - \vartheta_2^k}{2\eta^k}\left(\frac{\tau}{N}\right)\overline{\frac{\vartheta_3^{8-k} - \vartheta_2^{8-k}}{2\eta^{8-k}}}$$
$$+ \frac{\vartheta_4^k + \vartheta_1^k}{2\eta^k}\left(\frac{\tau}{N}\right)\overline{\frac{\vartheta_4^{8-k} + \vartheta_1^{8-k}}{2\eta^{8-k}}} + \frac{\vartheta_4^k - \vartheta_1^k}{2\eta^k}\left(\frac{\tau}{N}\right)\overline{\frac{\vartheta_4^{8-k} - \vartheta_1^{8-k}}{2\eta^{8-k}}} , \tag{A.23}$$

while the remaining blocks with $g \neq 0 \bmod N$ follow by acting with $\tau \to \tau + 1$,

$$Z_{\hat{\Lambda}}\begin{bmatrix} h \\ g \end{bmatrix}(\tau) = Z_{\hat{\Lambda}}\begin{bmatrix} h \\ 0 \end{bmatrix}(\tau + gh^{-1}) \tag{A.24}$$

where $h^{-1}$ is the inverse of $h$ in the multiplicative group $\mathbb{Z}_N$. The untwisted, unprojected block is then

$$Z_{\hat{\Lambda}}\begin{bmatrix} 0 \\ 0 \end{bmatrix} = Z_{\hat{\Lambda}}\begin{bmatrix} 0 \\ 1 \end{bmatrix} + \sum_{g=0}^{N-1} Z_{\hat{\Lambda}}\begin{bmatrix} 1 \\ g \end{bmatrix} , \tag{A.25}$$

---

[15]Indeed, $\mathcal{G}_k = \mathbb{Z}_2 \oplus \mathbb{Z}_2$ is $k$ is even, or $\mathbb{Z}_4$ is $k$ is odd, with the 4 elements in one-to-one correspondence with the highest weights $0, s, v, c$ of the adjoint, spinor, vector and conjugate spinor representations.

*i.e.* a sum over images of $Z_{\hat{\Lambda}}\begin{bmatrix} 0 \\ 1 \end{bmatrix}$ under $\Gamma_0(N)\backslash SL(2,\mathbb{Z}) = \{1, S, TS, \dots, T^{N-1}S\}$. As a consistency check, one can verify that the analogous sum for the Euclidean lattice $\Lambda$ reproduces the partition function of the Niemeier lattice,

$$\frac{\Theta_{D_k^{N+1}}}{\eta^{24}} = Z_\Lambda\begin{bmatrix} 0 \\ 1 \end{bmatrix} + \sum_{g=0}^{N-1} Z_\Lambda\begin{bmatrix} 1 \\ g \end{bmatrix} = \frac{E_4^3}{\eta^{24}} + 48k - 768 \,, \tag{A.26}$$

where $Z_\Lambda\begin{bmatrix} 0 \\ 1 \end{bmatrix}$ is obtained by replacing $\overline{\vartheta_i^{8-k}/\eta^{8-k}}$ by $(\vartheta_i/\eta)^k$ in (A.22).

The integrand of the one-loop vacuum amplitude follows in the same way as in the previous subsection, by combining the orbifold blocks $Z_{\hat{\Lambda}}\begin{bmatrix} h \\ g \end{bmatrix}(\tau)$ for the lattice $\hat{\Lambda}$ with the shifted partition function for the remaining $d - 8 + k$ compact directions (where $d$ is assumed to be greater that $8 - k$)

$$\Gamma_{d-8+k,d-8+k}\begin{bmatrix} h \\ g \end{bmatrix} = \tau_2^{\frac{d-8+k}{2}} \sum_{Q \in \Lambda_{d-8+k,d-8+k}+\frac{h}{N}\delta} (-1)^{\frac{2}{N}g\,\delta\cdot Q} q^{\frac{1}{2}Q_L^2} \bar{q}^{\frac{1}{2}Q_R^2} \,. \tag{A.27}$$

After eliminating $Z_{\hat{\Lambda}}\begin{bmatrix} 0 \\ 0 \end{bmatrix}$ using (A.25), grouping terms into an orbit of $\Gamma_0(N)\backslash SL(2,\mathbb{Z})$, and rescaling a $\mathbb{I}_{1,1}$ factor in $\Lambda_{d+2k-8,d}$ as[16]

$$\Lambda_{d+2k-8,d} = D_k[N] \oplus D_{8-k}[-1] \oplus \mathbb{I}_{1,1}[N] \oplus \mathbb{I}_{d+k-9,d+k-9} \,, \tag{A.28}$$

with a glue code $\{(0,0),(s,s),(v,v),(c,c)\}$ for the first two factors, we find

$$\mathcal{A}_{\text{orb}} = \left[ \frac{\Gamma_{\Lambda_{d+2k-8,d}}}{\Delta_k\begin{bmatrix} 0 \\ 1 \end{bmatrix}} + \frac{1}{N}\sum_{g=0}^{N-1} \frac{\Gamma_{\Lambda^*_{d+2k-8,d}}[(-1)^{gQ^2}]}{\Delta_k\begin{bmatrix} 1 \\ g \end{bmatrix}} \right] \times \frac{1}{2} \sum_{\alpha,\beta\in\{0,1\}} (-1)^{\alpha\beta+\alpha+\beta} \frac{\overline{\vartheta}^4\begin{bmatrix} \alpha \\ \beta \end{bmatrix}}{\tau_2^4 \overline{\eta}^{12}} \,, \tag{A.29}$$

where we defined the eta products

$$\Delta_k\begin{bmatrix} 0 \\ 1 \end{bmatrix} = \eta(\tau)^k \eta(N\tau)^k \,, \quad \Delta_k\begin{bmatrix} 1 \\ g \end{bmatrix} = e^{\frac{i\pi g k}{12}} \eta(\tau)^k \eta\left(\frac{\tau+g}{N}\right)^k \tag{A.30}$$

and

$$\begin{aligned} \Gamma_{\Lambda_{d+2k-8,d}} &= \tau_2^{\frac{d}{2}} \sum_{Q \in \Lambda_{d+2k-8,d}} q^{\frac{1}{2}Q_L^2} \bar{q}^{\frac{1}{2}Q_R^2} \\ \Gamma_{\Lambda^*_{d+2k-8,d}}[(-1)^{gQ^2}] &= \tau_2^{\frac{d}{2}} \sum_{Q \in \Lambda^*_{d+2k-8,d}} (-1)^{gQ^2} q^{\frac{1}{2}Q_L^2} \bar{q}^{\frac{1}{2}Q_R^2} \,. \end{aligned} \tag{A.31}$$

From this description, it is apparent that the degeneracy of twisted perturbative BPS states with charge $Q \in \Lambda^*_{d+2k-8,d}$, $Q \notin \Lambda_{d+2k-8,d}$ is given by the coefficient of $q^{-Q^2/2}$ in $1/\Delta_k\begin{bmatrix} 1 \\ 0 \end{bmatrix} = 1/\Delta_k(\tau/N)$, or equivalently the coefficient of $q^{-NQ^2/2}$ in $1/\Delta_k$, while the degeneracy of perturbative BPS states with charge $Q \in \Lambda_{d+2k-8,d} \subset \Lambda^*_{d+2k-8,d}$ has an additional contribution from the coefficient of $q^{-Q^2/2}$ in $1/\Delta_k$, in agreement with (2.14) and (2.15). For four-dimensional vacua ($d = 6$), we see that the electric charges carried by perturbative BPS states lie in the lattice $\Lambda_e = \Lambda^*_m$ where

---

[16]Note that this rescaling implies $\Gamma_{\Lambda^*_{d+2k-8,d}}(\tau) = (N^{\frac{k}{2}+1}/\tau^{k-4})\Gamma_{\Lambda_{d+2k-8,d}}(-1/\tau)$.

$$N = 3: \quad \Lambda_m = D_6[3] \oplus D_2[-1] \oplus I\!I_{1,1}[3] \oplus I\!I_{3,3}$$
$$N = 5: \quad \Lambda_m = D_4[5] \oplus D_4[-1] \oplus I\!I_{1,1}[5] \oplus I\!I_{1,1} \tag{A.32}$$
$$N = 7: \quad \Lambda_m = D_3[7] \oplus D_5[-1] \oplus I\!I_{1,1}[7]$$

This is in fact in agreement with the results stated in Table 1, thanks to the isomorphisms

$$D_6[3] \oplus D_2[-1] \simeq A_2 \oplus A_2 \oplus I\!I_{2,2}[3]$$
$$D_4[5] \oplus D_4[-1] \simeq I\!I_{2,2}[5] \oplus I\!I_{2,2} \tag{A.33}$$
$$D_3[7] \oplus D_5[-1] \simeq \begin{pmatrix} -4 & -1 \\ -1 & -2 \end{pmatrix} \oplus I\!I_{1,1}[7] \oplus I\!I_{2,2}$$

Indeed, both lattices on each line have the same genus, in particular the same discriminant group $L^*/L = \mathbb{Z}_N^k$. For $N = 2$ (hence $k = 8$), Eq. (A.28) continues to hold with the understanding that $D_8[2] \oplus D_0[-1] \equiv E_8[2]$.

Finally, we can obtain the one-loop $F^4$ amplitude by replacing the last factor in (A.29) by an insertion of the polynomial $P_{abcd}$ in (2.26), and then integrating over the fundamental domain $\mathcal{H}/SL(2,\mathbb{Z})$. As before, the integral can be unfolded onto a fundamental domain $\Gamma_0(N) \backslash \mathcal{H}$ for the action of $\Gamma_0(N)$ on $\mathcal{H}$, at the expense of keeping only the block $\begin{bmatrix} 0 \\ 1 \end{bmatrix}$,

$$F_{abcd}^{(\text{1-loop})} = \text{R.N.} \int_{\Gamma_0(N) \backslash \mathcal{H}} \frac{d\tau_1 d\tau_2}{\tau_2^2} \frac{\Gamma_{\Lambda_{d+2k-8,d}}[P_{abcd}]}{\Delta_k} \,, \tag{A.34}$$

where $\Delta_k \equiv \Delta_k \begin{bmatrix} 0 \\ 1 \end{bmatrix}$, thus establishing (2.24) for this class of models.

# B Ward identity in the degeneration $O(p,q) \to O(p-1, q-1)$

In section 3.2, we proved that the differential equations (3.17) and (3.22) are satisfied by the one-loop modular integral $F_{abcd}$ defined in (3.28). Here, we verify explicitly that the differential equation in (3.22) is verified by each Fourier mode in the degeneration limit $O(p,q) \to O(p-1, q-1)$, and that the solution is uniquely determined up to a moduli-independent summation measure.

Using the decomposition (4.4) and changing variable $R = e^{-\phi}$ for the non-compact Cartan generator of $O(p,q)$, the metric on moduli space reads

$$2P_{a\hat{b}} P^{a\hat{b}} = 2d\phi^2 + 2P_{\alpha\hat{\beta}} P^{\alpha\hat{\beta}} + e^{2\phi} \left( p_{L\,\alpha I} p_L{}^\alpha{}_J + p_{R\,\hat{\alpha} I} p_R{}^{\hat{\alpha}}{}_J \right) da^I da^J \tag{B.1}$$

with

$$P_{0\hat{0}} = -d\phi \,, \qquad P_{0\hat{\alpha}} = \frac{1}{\sqrt{2}} e^\phi p_{R\,\hat{\alpha} I} da^I, \qquad P_{\alpha\hat{0}} = \frac{1}{\sqrt{2}} e^\phi p_{L\,\alpha I} da^I \,. \tag{B.2}$$

Beware that in this section we use the same notations $p_L$ and $p_R$ for both $O(p,q)$ and $O(p-1, q-1)$, so $p_{L\,\alpha I} Q^I$ is not $p_{L\,\alpha \mathcal{I}} Q^{\mathcal{I}}$ for $a = \alpha$.

One can compute the covariant derivative in tangent frame such that

$$dZ_a = 2P^{b\hat{c}} \partial_{b\hat{c}} Z_a = 2P^{b\hat{c}} (\mathcal{D}_{b\hat{c}} Z_a - B_{b\hat{c}a}{}^d Z_d), \tag{B.3}$$

and similarly for hatted indices. This way one computes that, for any tensor $F_a = (F_0, F_\alpha, F_{\hat{\alpha}}, F_{\hat{0}})$, $F_b = (F_0, F_\beta, F_{\hat{\beta}}, F_{\hat{0}})$, ...

$$\mathcal{D}_{0\hat{0}}F_a = -\frac{1}{2}\frac{\partial}{\partial \phi}F_a ,$$

$$\mathcal{D}_{\alpha\hat{0}}F_a = \frac{1}{\sqrt{2}}e^{-\phi}v^{\cdot 1I}{}_\alpha\frac{\partial}{\partial a^I}F_a + \frac{1}{2}\left(F_\alpha, -\delta_{\alpha\beta}F_0, 0, 0\right)$$

$$\mathcal{D}_{0\hat{\alpha}}F_a = \frac{1}{\sqrt{2}}e^{-\phi}v^{\cdot 1I}{}_{\hat{\alpha}}\frac{\partial}{\partial a^I}F_b + \frac{1}{2}\left(0, 0, -\delta_{\alpha\beta}F_{\hat{0}}, F_{\hat{\alpha}}\right) , \tag{B.4}$$

and finally the operator $\mathcal{D}_{\alpha\beta}$ will only be acting on the moduli fields through the projectors $p^I_{L\gamma}$, $p^I_{R\hat{\gamma}}$:

$$\mathcal{D}_{\alpha\hat{\beta}}p^I_{L\gamma} = \tfrac{1}{2}\delta_{\alpha\gamma}p^I_{R\hat{\beta}}, \qquad \mathcal{D}_{\alpha\hat{\beta}}p^I_{L\hat{\gamma}} = \tfrac{1}{2}\delta_{\hat{\beta}\hat{\gamma}}p^I_{R\alpha} . \tag{B.5}$$

Recall the differential equation (2.23)

$$\mathcal{D}^{\hat{c}}_{(e}\mathcal{D}_{f)\hat{c}}F_{abcd} = \frac{2-q}{4}\delta_{ef}F_{abcd} + (4-q)\delta_{a)(e}F_{f)(bcd} + 3\delta_{(ab}F_{cd)ef} . \tag{B.6}$$

For brevity we define the vector

$$\vec{F} = \left(F_{1111}, F_{111\alpha}, F_{11\alpha\beta}, F_{1\alpha\beta\gamma}, F_{\alpha\beta\gamma\delta}\right)^{\intercal} \tag{B.7}$$

and $\vec{F}_Q$ such that $\vec{F} = \sum_Q \vec{F}_Q e^{2\pi iQ\cdot a}$. The first component $(e, f) = (0, 0)$ gives

$$4\mathcal{D}_0{}^{\hat{c}}\mathcal{D}_{0\hat{c}}\vec{F}_Q = (\partial_\phi(\partial_\phi + q - 1) - 8\pi^2 e^{-2\phi}Q_R^2)\vec{F}_Q = -\begin{pmatrix} 5(q-6)F_{1111} \\ 4(q-5)F_{111\alpha} \\ 3(q-4)F_{11\alpha\beta} - 2\delta_{\alpha\beta}F_{1111} \\ 2(q-3)F_{1\alpha\beta\gamma} - 6\delta_{(\alpha\beta}F_{111\gamma)} \\ (q-2)F_{\alpha\beta\gamma\delta} - 12\delta_{(\alpha\beta}F_{11\gamma\delta)} \end{pmatrix} . \tag{B.8}$$

Then the action of the differential operator

$$2\mathcal{D}_0{}^{\hat{c}}\mathcal{D}_{\eta\hat{c}}\vec{F}_Q + 2\mathcal{D}_\eta{}^{\hat{c}}\mathcal{D}_{0\hat{c}}\vec{F}_Q = -2\pi i\sqrt{2}e^{-\phi}(Q_{L\eta}(\partial_\phi + q - 2) + 2Q_{R\hat{\alpha}}\mathcal{D}_\eta{}^{\hat{\alpha}})\vec{F}_Q$$

$$-(\partial_\phi + \frac{q-2}{2})\begin{pmatrix} 4F_{111\eta} \\ 3F_{11\alpha\eta} - \delta_{\eta\alpha}F_{1111} \\ 2F_{1\alpha\beta\eta} - 2\delta_{\eta(\alpha}F_{111\beta)} \\ F_{\alpha\beta\gamma\eta} - 3\delta_{\eta(\alpha}F_{11\beta\gamma)} \\ -4\delta_{\eta(\alpha}F_{1\beta\gamma\delta)} \end{pmatrix} , \tag{B.9}$$

allows to obtain the second component $(e, f) = (0, \alpha)$ of the differential equation

$$-2\pi i\sqrt{2}e^{-\phi}(Q_{L\eta}(\partial_\phi + q - 2) + 2Q_{R\hat{\alpha}}\mathcal{D}_\eta{}^{\hat{\alpha}})\vec{F}_Q$$

$$= \begin{pmatrix} 4(\partial_\phi + 4)F_{111\eta} \\ 3(\partial_\phi + 3)F_{11\alpha\eta} - \delta_{\eta\alpha}(\partial_\phi + q - 3)F_{1111} \\ 2(\partial_\phi + 2)F_{1\alpha\beta\eta} - 2\delta_{\eta(\alpha}(\partial_\phi + q - 3)F_{111\beta)} + 2\delta_{\alpha\beta}F_{111\eta} \\ (\partial_\phi + 1)F_{\alpha\beta\gamma\eta} - 3\delta_{\eta(\alpha}(\partial_\phi + q - 3)F_{11\beta\gamma)} + 6\delta_{(\alpha\beta}F_{11\gamma)\eta} \\ -4\delta_{\eta(\alpha}(\partial_\phi + q - 3)F_{1\beta\gamma\delta)} + 12\delta_{(\alpha\beta}F_{1\gamma\delta)\eta} \end{pmatrix} . \tag{B.10}$$

The final differential operator for $(e,f) = (\eta,\vartheta)$

$$
\begin{aligned}
4\mathcal{D}_{(\eta}{}^{\hat{c}}\mathcal{D}_{\vartheta)\hat{c}}\vec{F}_Q &= (4\mathcal{D}_{(\eta}{}^{\hat{\gamma}}\mathcal{D}_{\vartheta)\hat{\gamma}} + \delta_{\eta\vartheta}\partial_\phi - 8\pi^2 e^{-2\phi}Q_{L\,\eta}Q_{L\,\vartheta})\vec{F}_Q \\
&\quad + 4\pi\mathrm{i}\sqrt{2}e^{-\phi}Q_{L(\eta}\begin{pmatrix} 4F_{111\vartheta)} \\ 3F_{11\alpha|\vartheta)} - \delta_{\vartheta)\alpha}F_{1111} \\ 2F_{1\alpha\beta|\vartheta)} - 2\delta_{\vartheta)(\alpha}F_{111\beta)} \\ F_{\alpha\beta\gamma|\vartheta)} - 3\delta_{\vartheta)(\alpha}F_{11\beta\gamma)} \\ -4\delta_{\vartheta)(\alpha}F_{1\beta\gamma\delta)} \end{pmatrix} \\
&\quad + \begin{pmatrix} 12F_{11\eta\vartheta} - 4\delta_{\eta\vartheta}F_{1111} \\ 6F_{1\alpha\eta\vartheta} - 3\delta_{\eta\vartheta}F_{111\alpha} - 7\delta_{\alpha(\eta}F_{111\vartheta)} \\ 2F_{\alpha\beta\eta\vartheta} - 2\delta_{\eta\vartheta}F_{11\alpha\beta} - 10\delta_{\alpha)(\eta}F_{11\vartheta)(\beta} + 2\delta_{\alpha)(\eta}\delta_{\vartheta)(\beta}F_{1111} \\ -\delta_{\eta\vartheta}F_{1\alpha\beta\gamma} - 9\delta_{\alpha)(\eta}F_{1\vartheta)(\beta\gamma} + 6\delta_{\alpha)(\eta}\delta_{\vartheta)(\beta}F_{111\gamma} \\ -4\delta_{\alpha)(\eta}F_{\vartheta)(\beta\gamma\delta} + 12\delta_{\alpha)(\eta}\delta_{\vartheta)(\beta}F_{11\gamma\delta} \end{pmatrix},
\end{aligned}
\tag{B.11}
$$

gives a third differential equation

$$
\begin{aligned}
&(4\mathcal{D}_{(\eta}{}^{\hat{\gamma}}\mathcal{D}_{\vartheta)\hat{\gamma}} + \delta_{\eta\vartheta}\partial_\phi - 8\pi^2 e^{-2\phi}Q_{L\,\eta}Q_{L\,\vartheta})\vec{F}_Q + 4\pi\mathrm{i}\sqrt{2}e^{-\phi}Q_{L(\eta}\begin{pmatrix} 4F_{111\vartheta)} \\ 3F_{11\alpha|\vartheta)} - \delta_{\vartheta)\alpha}F_{1111} \\ 2F_{1\alpha\beta|\vartheta)} - 2\delta_{\vartheta)(\alpha}F_{111\beta)} \\ F_{\alpha\beta\gamma|\vartheta)} - 3\delta_{\vartheta)(\alpha}F_{11\beta\gamma)} \\ -4\delta_{\vartheta)(\alpha}F_{1\beta\gamma\delta)} \end{pmatrix} \\
&= -\begin{pmatrix} (q-6)\delta_{\eta\vartheta}F_{1111} \\ (q-5)\delta_{\eta\vartheta}F_{111\alpha} + (q-11)\delta_{\alpha(\eta}F_{111\vartheta)} \\ (q-4)\delta_{\eta\vartheta}F_{11\alpha\beta} - 2\delta_{\alpha\beta}F_{11\eta\vartheta} + 2(q-9)\delta_{\alpha)(\eta}F_{11\vartheta)(\beta} + 2\delta_{\alpha)(\eta}\delta_{\vartheta)(\beta}F_{1111} \\ (q-3)\delta_{\eta\vartheta}F_{1\alpha\beta\gamma} - 6\delta_{(\alpha\beta}F_{1\gamma)\eta\vartheta} + 3(q-7)\delta_{\alpha)(\eta}F_{1\vartheta)(\beta\gamma} + 6\delta_{\alpha)(\eta}\delta_{\vartheta)(\beta}F_{111\gamma} \\ (q-2)\delta_{\eta\vartheta}F_{\alpha\beta\gamma\delta} - 12\delta_{(\alpha\beta}F_{\gamma\delta)\eta\vartheta} + 4(q-5)\delta_{\alpha)(\eta}F_{\vartheta)(\beta\gamma\delta} + 12\delta_{\alpha)(\eta}\delta_{\vartheta)(\beta}F_{11\gamma\delta} \end{pmatrix}
\end{aligned}
\tag{B.12}
$$

One can then check that the only exponentially suppressed solution to the three equations (B.8), (B.10) and (B.12) is given, up to a moduli-independent prefactor, by

$$
\vec{F}_Q = \begin{pmatrix} F_1^{(4)} \\ Q_{L\,\alpha}F_1^{(3)} \\ Q_{L\,\alpha}Q_{L\,\beta}F_1^{(2)} + \delta_{\alpha\beta}F_2^{(2)} \\ Q_{L\,\alpha}Q_{L\,\beta}Q_{L\,\gamma}F_1^{(1)} + \delta_{(\alpha\beta}Q_{L\,\gamma)}(Q)F_2^{(1)} \\ Q_{L\,\alpha}Q_{L\,\beta}Q_{L\,\gamma}Q_{L\,\delta}F_1^{(0)} + \delta_{(\alpha\beta}Q_{L\,\gamma}Q_{L\,\delta)}F_2^{(0)} + \delta_{(\alpha\beta}\delta_{\gamma\delta)}F_3^{(0)} \end{pmatrix},
\tag{B.13}
$$

$$
F_1^{(k)} = \left(\frac{\mathrm{i}}{\sqrt{2}}\right)^k 2^{\frac{q-2}{2}}(2\pi)^{\frac{q-3-2k}{2}}R^{\frac{q-1}{2}}\sqrt{2|Q_R|^2}^{\frac{2k+3-q}{2}}K_{\frac{2k+3-q}{2}}(2\pi R\sqrt{2|Q_R|^2})
$$

$$
F_2^{(k)} = -\left(\frac{\mathrm{i}}{\sqrt{2}}\right)^k 2^{\frac{q-4}{2}}\frac{(4-k)(3-k)}{2}(2\pi)^{\frac{q-5-2k}{2}}R^{\frac{q-3}{2}}\sqrt{2|Q_R|^2}^{\frac{2k+5-q}{2}}K_{\frac{2k+5-q}{2}}(2\pi R\sqrt{2|Q_R|^2})
$$

$$
F_3^{(0)} = 3 \times 2^{\frac{q-6}{2}}(2\pi)^{\frac{q-7}{2}}R^{\frac{q-5}{2}}\sqrt{2|Q_R|^2}^{\frac{7-q}{2}}K_{\frac{7-q}{2}}(2\pi R\sqrt{2|Q_R|^2}),
\tag{B.14}
$$

In particular, the tensorial part of the function $\vec{F}_Q$ is polynomial in $Q_{L\alpha}, \dots$, and the rest only depends on the moduli through $Q_R^2$ and $R = e^{-\phi}$. We conclude that the Fourier coefficient $\vec{F}_Q$ for

a fixed $Q$ is uniquely determined by the differential equations (3.17) and (3.22) up to an overal constant corresponding to the measure factor.

The power-low terms satisfy to the same equations for $Q = 0$. One easily computes that the only two solutions are such that

$$
\vec{F} = \begin{pmatrix} (7-q)(9-q)c_0 R^{q-6} \\ 0 \\ (7-q)c_0 R^{q-6}\delta_{\alpha\beta} \\ 0 \\ 3c_0 R^{q-6}\delta_{(\alpha\beta}\delta_{\gamma\delta)} + R F^{p-1,q-1}_{\alpha\beta\gamma\delta} \end{pmatrix},
\tag{B.15}
$$

for an arbitrary constant $c_0$ and a solution $F^{p-1,q-1}_{\alpha\beta\gamma\delta}$ to (3.17) and (3.22) on $G_{p-1,q-1}$.

# C  Polynomials appearing in Fourier modes

In the degeneration limit $O(p,q) \rightarrow O(p-1,q-1)$ studied in §4, the monomials $\tilde{P}^{(\ell)}_{\alpha_{h+1}\dots\alpha_4}(Q)$ with $\ell \geq 0$ are of degree $4 - 2\ell - h$ in $Q$, and defined by

$$
\sum_{\ell \geq 0} \tilde{P}^{(\ell)}_{\alpha\beta\gamma\delta}(Q = Q_{L,\alpha}Q_{L,\beta}Q_{L,\gamma}Q_{L,\delta} - \frac{3}{2\pi}\delta_{(\alpha\beta}Q_{L,\gamma}Q_{L,\delta)} + \frac{3}{16\pi^2}\delta_{(\alpha\beta}\delta_{\gamma\delta)},
$$

$$
\sum_{\ell \geq 0} \tilde{P}^{(\ell)}_{\alpha\beta\gamma}(Q) = Q_{L,\alpha}Q_{L,\beta}Q_{L,\gamma} - \frac{3}{4\pi}Q_{L,(\alpha}\delta_{\beta\gamma)},
$$

$$
\sum_{\ell \geq 0} \tilde{P}^{(\ell)}_{\alpha\beta}(Q) = Q_{L,\alpha}Q_{L,\beta} - \frac{1}{4\pi}\delta_{\alpha\beta},
\tag{C.1}
$$

$$
\sum_{\ell \geq 0} \tilde{P}^{(\ell)}_{\alpha}(Q) = Q_{L,\alpha},
$$

$$
\sum_{\ell \geq 0} \tilde{P}^{(\ell)}(Q) = 1.
$$

In the degeneration limit $O(p,q) \rightarrow O(p-2,q-2)$ studied in §5, the monomials $\mathcal{P}^{(\ell)}_{\mu_1\dots\mu_h\alpha_{h+1}\dots\alpha_4}(Q'^i,S)$ with $\ell \geq 0$ are of degree $4 - 2\ell - h$ in $Q'^i$, and defined by

$$
\sum_{\ell \geq 0} \mathcal{P}^{(\ell)}_{\alpha\beta\gamma\delta}(Q'^i,S) = Q'^i_{L,(\alpha}Q'^j_{L,\beta}Q'^k_{L,\gamma}Q'^l_{L,\delta)}M_{ij}M_{kl} - \frac{3}{2\pi}\delta_{(\alpha\beta}Q'^i_{L,\gamma}Q'^j_{L,\delta)}M_{ij} + \frac{3}{16\pi^2}\delta_{(\alpha\beta}\delta_{\gamma\delta)},
$$

$$
\sum_{\ell \geq 0} \mathcal{P}^{(\ell)}_{\mu\alpha\beta\gamma}(Q'^i,S) = Q'_{L,\mu(\alpha}Q'^i_{L,\beta}Q'^j_{L,\gamma)}M_{ij} - \frac{3}{4\pi}Q'_{L,\mu(\alpha}\delta_{\beta\gamma)},
$$

$$
\sum_{\ell \geq 0} \mathcal{P}^{(\ell)}_{\mu\nu\alpha\beta}(Q'^i,S) = Q'_{L,\mu\alpha}Q'_{L,\nu\beta} - \frac{1}{4\pi}\delta_{\alpha\beta}\frac{Q'_\mu \cdot Q'_\nu}{Q'_\tau \cdot Q'^\tau},
\tag{C.2}
$$

$$
\sum_{\ell \geq 0} \mathcal{P}^{(\ell)}_{\mu\nu\rho\alpha}(Q'^i,S) = Q'_{L,\mu\alpha}\frac{Q'_\nu \cdot Q'_\rho}{Q'_\tau \cdot Q'^\tau},
$$

$$
\sum_{\ell \geq 0} \mathcal{P}^{(\ell)}_{\mu\nu\rho\sigma}(Q'^i,S) = \frac{Q'_{(\mu} \cdot Q'_\nu Q'_\mu \cdot Q'_{\sigma)}}{(Q'_\tau \cdot Q'^\tau)^2},
$$

where $M_{ij} = v_{i\mu}v^\mu_j$ is the torus metric (5.4), and $Q'_\mu \cdot Q'_\nu = \frac{1}{S_2}\begin{pmatrix} (Q+S_1P)^2 & (Q+S_1P)S_2P \\ (Q+S_1P)S_2P & S_2^2P^2 \end{pmatrix}$.

# D   Tensorial Eisenstein series

In the degeneration limit $O(p,q) \to O(p-2,q-2)$ studied in §5, the power-like terms in (5.29) involve tensorial Eisenstein series that we rewrote as tensorial derivatives of real analytic Eisenstein series, using $\mathcal{D}_{\mu\nu}$ the traceless differential operator on $SL(2,\mathbb{R})/O(2)$. Here we exhibit these relations, and show how this operator can be rewritten in terms of lowering and raising operators $\mathcal{D}_w$ and $\overline{\mathcal{D}}_w$.

The non-holomorphic Eisenstein series

$$\mathcal{E}_{s,w}(S) = \frac{1}{2\zeta(2s)} \sum_{(c,d)\in\mathbb{Z}^2\smallsetminus\{0,0\}} \frac{S_2^s}{(c+dS)^{s+\frac{w}{2}}(c+d\bar{S})^{s-\frac{w}{2}}} \tag{D.1}$$

has modular weight $(\frac{w}{2}, -\frac{w}{2})$ under $SL(2,\mathbb{Z})$. The raising and lowering operators, $\mathcal{D}_w = 2iS_2\partial_S + \frac{w}{2}$ and $\overline{\mathcal{D}}_w = -2iS_2\partial_{\bar{S}} - \frac{w}{2}$ act on $\mathcal{E}_{s,w}(S)$ according to

$$\mathcal{D}_w \mathcal{E}_{s,w} = \left(s+\frac{w}{2}\right)\mathcal{E}_{s,w+2}, \qquad \overline{\mathcal{D}}_w \mathcal{E}_{s,w} = \left(s-\frac{w}{2}\right)\mathcal{E}_{s,w-2}. \tag{D.2}$$

Non-holomorphic Eisenstein series are thus eigenmodes of the laplacian $\Delta_w = \bar{\mathcal{D}}_{w+2}\mathcal{D}_w$ with eigenvalue $\left(s+\frac{w}{2}\right)\left(s-\frac{w}{2}-1\right)$.

Alternatively, one can denote the momenta and winding along a torus as $z_\mu = m_i v_\mu^i$ with $(m_1, m_2) = (c,d)$, $v_\mu{}^i$ is the vielbein defined in (5.4), such that $z_\mu z^\mu = \frac{1}{S_2}|c+dS|^2$ is invariant under $SL(2,\mathbb{Z})$. The traceless differential operator $\mathcal{D}_{\mu\nu}$ acts as

$$\mathcal{D}_{\mu\nu}z_\rho = \frac{1}{2}\delta_{\rho(\mu}z_{\nu)} - \frac{1}{4}\delta_{\mu\nu}z_\rho. \tag{D.3}$$

One can show that they are related to the lowering and raising operator through

$$\mathcal{D}_{\mu\nu} = -\frac{1}{2}\sigma^+_{\mu\nu}\mathcal{D}_w - \frac{1}{2}\sigma^-_{\mu\nu}\bar{\mathcal{D}}_w \tag{D.4}$$

where $\sigma^\pm = \frac{1}{2}(\sigma_3 \pm i\sigma_1)$ and $\sigma_i$ are the Pauli matrices. By acting on non-holomorphic Eisenstein series of weight 0 with $\mathcal{D}_{\mu\nu}$ and $\mathcal{D}_{(\mu\nu}\mathcal{D}_{\rho\sigma)}$, one obtains the relations

$$\frac{s}{2}\sigma^+_{\mu\nu}\mathcal{E}_{s,2} + \frac{s}{2}\sigma^-_{\mu\nu}\mathcal{E}_{s,-2} = \frac{s}{2\zeta(2s)}\sum_{(j,p)}' \frac{1}{(z_\tau z^\tau)^s}\left(\frac{z_\mu z_\nu}{z_\tau z^\tau} - \frac{1}{2}\delta_{\mu\nu}\right)$$

$$\times \frac{s(s+1)}{4}\sigma^+_{(\mu\nu}\sigma^+_{\rho\sigma)}\mathcal{E}_{s,4} + \frac{s(s-1)}{4}\sigma^-_{(\mu\nu}\sigma^-_{\rho\sigma)}\mathcal{E}_{s,-4} + s(s-1)\left(\sigma^+_{(\mu\nu}\sigma^-_{\rho\sigma)} - \frac{1}{8}\delta_{(\mu\nu}\delta_{\rho\sigma)}\right)\mathcal{E}_{s,0} \tag{D.5}$$

$$= \frac{s(s+1)}{2\zeta(2s)}\sum_{(j,p)}' \frac{1}{(z_\tau z^\tau)^s}\left(\frac{z_\mu z_\nu z_\rho z_\sigma}{(z_\tau z^\tau)^2} - \frac{\delta_{(\mu\nu}z_\rho z_{\sigma)}}{z_\tau z^\tau} + \frac{1}{8}\delta_{(\mu\nu}\delta_{\rho\sigma)}\right)$$

where the second line is traceless.

Now, the components $F^{(p,q),1,0}_{\alpha\beta\mu\nu}$ and $F^{(p,q),1,0}_{\mu\nu\rho\sigma}$ in (5.10) were obtained originally as

$$F^{(p,q),1,0}_{\alpha\beta\mu\nu} = R^{q-6}\frac{c(0)}{4\pi^2}\left(\frac{8-q}{2}\right)\frac{1}{2\zeta(8-q)}\sum_{(j,p)}' \frac{1}{(z_\tau z^\tau)^{\frac{8-q}{2}}}\frac{z_\mu z_\nu}{z_\tau z^\tau},$$

$$F^{(p,q),1,0}_{\mu\nu\rho\sigma} = R^{q-6}\frac{c(0)}{2\pi^2}\left(\frac{8-q}{2}\right)\left(\frac{10-q}{2}\right)\frac{1}{2\zeta(8-q)}\sum_{(j,p)}' \frac{1}{(z_\tau z^\tau)^{\frac{8-q}{2}}}\frac{z_\mu z_\nu z_\rho z_\sigma}{(z_\tau z^\tau)^2} \tag{D.6}$$

They can be written as in (5.10) by rewritting the relations above, for $s \neq -1$

$$\frac{s}{2\zeta(2s)} \sum_{(j,p)}' \frac{1}{(z_\tau z^\tau)^s} \frac{z_\mu z_\nu}{z_\tau z^\tau} = \frac{s}{2} \left( \delta_{\mu\nu} \mathcal{E}_{s,0} + \sigma^+_{\mu\nu} \mathcal{E}_{s,2} + \sigma^-_{\mu\nu} \mathcal{E}_{s,-2} \right),$$

$$\frac{s(s+1)}{2\zeta(2s)} \sum_{(j,p)}' \frac{1}{(z_\tau z^\tau)^s} \frac{z_\mu z_\nu z_\rho z_\sigma}{(z_\tau z^\tau)^2} = \frac{s(s+1)}{4} \sigma^+_{(\mu\nu} \sigma^+_{\rho\sigma)} \mathcal{E}_{s,4} + \frac{s(s-1)}{4} \sigma^-_{(\mu\nu} \sigma^-_{\rho\sigma)} \mathcal{E}_{s,-4}$$
$$+ \frac{s(s+1)}{2} \left( \delta_{(\mu\nu} \sigma^+_{\rho\sigma)} \mathcal{E}_{s,2} + \delta_{(\mu\nu} \sigma^-_{\rho\sigma)} \mathcal{E}_{s,-2} \right)$$
$$+ \frac{s^2}{2} \left( \sigma^+_{(\mu\nu} \sigma^-_{\rho\sigma)} - \frac{1}{4} \delta_{(\mu\nu} \delta_{\rho\sigma)} \right) \mathcal{E}_{s,0} + \frac{3s(s+1)}{8} \delta_{(\mu\nu} \delta_{\rho\sigma)} \mathcal{E}_{s,0}$$
(D.7)

In other words, all the tensorial series in (5.29) appearing as low-energy propagators on the torus can be rewritten a combination of $\mathcal{E}_{s,0}$, $\mathcal{D}\mathcal{E}_{s,0}$, $\overline{\mathcal{D}}\mathcal{E}_{s,0}$, $\mathcal{D}^2 \mathcal{E}_{s,0}$ and $\overline{\mathcal{D}}^2 \mathcal{E}_{s,0}$. This is used extensively to rewrite the 1-PI effective action in four dimensions (5.39).

Similarly, they can also be rewritten using traceless differential operators $\mathcal{D}_{\mu\nu}$ and

$$\mathcal{D}^2_{\mu\nu\rho\sigma} = \mathcal{D}_{(\mu\nu} \mathcal{D}_{\rho\sigma)} - \tfrac{1}{4} \delta_{(\mu\nu} \delta_{\rho\sigma)} \mathcal{D}_{\tau\kappa} \mathcal{D}^{\tau\kappa} \qquad (D.8)$$

as

$$\frac{s}{2\zeta(2s)} \sum_{(j,p)}' \frac{1}{(z_\tau z^\tau)^s} \frac{z_\mu z_\nu}{z_\tau z^\tau} = \left( \frac{s}{2} \delta_{\mu\nu} - \mathcal{D}_{\mu\nu} \right) \mathcal{E}_{s,0}$$

$$\frac{s(s+1)}{2\zeta(2s)} \sum_{(j,p)}' \frac{1}{(z_\tau z^\tau)^s} \frac{z_\mu z_\nu z_\rho z_\sigma}{(z_\tau z^\tau)^2} = \left( \mathcal{D}^2_{\mu\nu\rho\sigma} - (s+1) \delta_{(\mu\nu} \mathcal{D}_{\rho\sigma)} + \frac{3}{8} s(s+1) \delta_{(\mu\nu} \delta_{\rho\sigma)} \right) \mathcal{E}_{s,0}$$
(D.9)

# E  Poincaré series and Eisenstein series for $O(p,q,\mathbb{Z})$

In this section, we evaluate the modular integrals (3.28) and (3.29) using the method developed in [50, 44], which keeps invariance under the automorphism group $O(p,q,\mathbb{Z})$ of the lattice $\Lambda_{p,q}$ manifest. The result is expressed as a sum over lattice vectors with fixed norm, which is a special type of Poincaré series for $O(p,q,\mathbb{Z})$. In §E.2, we use a similar method to construct Eisenstein series for $O(p,q,\mathbb{Z})$.

## E.1  Poincaré series representation of $F^{p,q}$

The method developed in [50, 44] relies on expressing the factor multiplying the lattice sum in the integrand in terms of a special type of Poincaré series for $\Gamma_0(N)$, known as the Niebur-Poincaré series of weight $w \in 2\mathbb{Z}$,

$$\mathcal{F}_N(s, \kappa, w; \tau) = \frac{1}{2} \sum_{\gamma \in \Gamma_\infty \backslash \Gamma_0(N)} \mathcal{M}_{s,w}(-\kappa \tau_2) e^{-2\pi i \kappa \tau_1} |_w \gamma , \qquad (E.1)$$

where $\mathcal{M}_{s,w}(y)$ is the Whittaker function defined in [50, Eq. (2.7)], and $|_w \gamma$ is the Petersson slash operator, $[f|_w \gamma](\tau) = (c\tau + d)^{-k} f\left(\frac{a\tau+b}{c\tau+d}\right)$ for $\gamma = \left( \begin{smallmatrix} a & b \\ c & d \end{smallmatrix} \right)$. The series converges absolutely

for $\text{Re}(s) > 1$, grows as $\frac{\Gamma(2s)}{\Gamma(s+\frac{w}{2})} q^{-\kappa}$ near the cusp $\tau \to i\infty$ and is regular at the cusp $\tau = 0$. It transforms under the Maass raising and lower operators according to

$$
\begin{aligned}
D\mathcal{F}_N(s,\kappa,w) &= 2\kappa\left(s + \tfrac{w}{2}\right)\mathcal{F}_N(s,\kappa,w+2)\,, \\
\bar{D}\mathcal{F}_N(s,\kappa,w) &= \frac{1}{8\kappa}\left(s - \tfrac{w}{2}\right)\mathcal{F}_N(s,\kappa,w-2)\,,
\end{aligned}
\tag{E.2}
$$

which implies that it is an eigenmode of the weight $w$ Laplacian on $\mathcal{H}$ with the eigenvalue $(s - \frac{w}{2})(s - 1 + \frac{w}{2})$. In particular, for $w < 0$ and $s = 1 - \frac{w}{2}$, $\mathcal{F}_N(s,\kappa,w)$ is a harmonic Maass form of weight $w$. In cases where there exists no cusp form of weight $2 - w$, it is actually a weakly holomorphic modular form of weight $w$ [49]. The Fourier expansion of $\mathcal{F}_N(s,\kappa,w) \equiv \mathcal{F}_\infty(s,\kappa,w;\tau)$ around the cusps at $\infty$ and at 0 is given in [44, Eq. (5.8-10)], in terms of the Kloosterman sums $\mathcal{Z}_{\infty\infty}(m,n;s)$ and $\mathcal{Z}_{0\infty}(m,n;s)$ defined in Eq. A.3 and A.4 of loc. cit. For $N = 1$, one has, by matching the residue of the pole at $\tau = i\infty$,

$$
\frac{1}{\Delta(\tau)} = \lim_{s\to 7} \frac{\mathcal{F}_1(s,1,-12;\tau)}{\Gamma(2s)}\,.
\tag{E.3}
$$

For $N = 2, 3, 5, 7$, using the fact that $\Delta_k$ is invariant under the Fricke involution, one has instead

$$
\frac{1}{\Delta_k(\tau)} = \lim_{s\to 1+\frac{k}{2}} \frac{\left[\mathcal{F}_N(s,1,-k;\tau) + \hat{\mathcal{F}}_N(s,1,-k;\tau)\right]}{\Gamma(2s)}\,,
\tag{E.4}
$$

where $\hat{\mathcal{F}}_N(s,\kappa,w;\tau)$ is the image of $\mathcal{F}_N(s,\kappa,w;\tau)$ under the Fricke involution.[17]

We shall compute the family of integrals

$$
\begin{aligned}
F^{(p,q)}(\Phi,s,\kappa) &= \frac{1}{\Gamma(2s)} \int_{\Gamma_0(N)\backslash\mathcal{H}} \frac{\mathrm{d}\tau_1 \mathrm{d}\tau_2}{\tau_2^2}\, \Gamma_{\Lambda_{p,q}}\, \mathcal{F}_N\!\left(s,\kappa,-\tfrac{p-q}{2};\tau\right)\,, \\
F_{abcd}^{(p,q)}(\Phi,s,\kappa) &= \frac{1}{\Gamma(2s)} \int_{\Gamma_0(N)\backslash\mathcal{H}} \frac{\mathrm{d}\tau_1 \mathrm{d}\tau_2}{\tau_2^2}\, \Gamma_{\Lambda_{p,q}}[P_{abcd}]\, \mathcal{F}_N\!\left(s,\kappa,-\tfrac{p-q}{2}-4;\tau\right)\,,
\end{aligned}
\tag{E.5}
$$

which converges absolutely for $\text{Re}(s) > \frac{p+q}{4}$. Here, $\Gamma_{\Lambda_{p,q}}[P_{abcd}]$ is the partition function of a $N$-modular lattice $\Lambda_{p,q}$ of signature $(p,q)$. It follows from the $N$-modularity property that $|\Lambda_{p,q}^*/\Lambda_{p,q}| = N^{(p+q)/2}$, and that $\Gamma_{\Lambda_{p,q}}[P_{abcd}]$ satisfies

$$
\Gamma_{\Lambda_{p,q}}[P_{abcd}](\Phi,\tau) = \left(-i\tau\sqrt{N}\right)^{-4-\frac{p-q}{2}} \Gamma_{\Lambda_{p,q}}[P_{abcd}]\left(\sigma\cdot\Phi, -\frac{1}{N\tau}\right)
\tag{E.6}
$$

where $\sigma$ is the $O(p,q,\mathbb{R})$ transformation realizing the isomorphism $\Lambda_{p,q}^* \simeq \Lambda_{p,q}[1/N]$. The desired integrals (4.1) are then obtain by taking a limit

$$
\begin{aligned}
F^{(p,q)}(\Phi) &= \frac{1}{8} \lim_{s\to 1+\frac{k}{2}} \left[F^{p,q}(\Phi,s,1) + F^{(p,q)}(\sigma\cdot\Phi,s,1)\right] \\
F_{abcd}^{(p,q)}(\Phi) &= \lim_{s\to 1+\frac{k}{2}} \left[F_{abcd}^{(p,q)}(\Phi,s,1) + F_{abcd}^{(p,q)}(\sigma\cdot\Phi,s,1)\right]\,.
\end{aligned}
\tag{E.7}
$$

---

[17] For $N = 7$, $1/\Delta_3$ is a modular form of odd weight with character $\chi = (\frac{\cdot}{7})$, so the Petersson slash operator $|_w\gamma$ in (E.1) involves an additional factor of $\chi(d)^{-1}$. This results in additional factors of $\chi(d)^{-1}$ and $\chi(c)^{-1}$ in the Kloosterman sums $\mathcal{Z}_{\infty\infty}(m,n;s)$ and $\mathcal{Z}_{0\infty}(m,n;s)$, respectively.

By unfolding the integration domain against the sum over $\gamma$, one obtains, for $\mathrm{Re}(s) > \frac{p+q}{4}$,

$$F^{(p,q)}_{abcd}(s,\kappa) = \frac{1}{\Gamma(2s)} \sum_{Q\in\Lambda_{p,q}} \int_{\mathcal{S}} d\tau_1 d\tau_2\, \tau_2^{q/2-2}\, P_{abcd}\, e^{i\pi(\tau p_L^2 - \bar{\tau} p_R^2)} \mathcal{M}_{s,w}(-\kappa\tau_2)\, e^{-2\pi i \tau_1 \kappa}, \tag{E.8}$$

where $\mathcal{S}$ denotes the strip $-\frac{1}{2} < \tau_1 < \frac{1}{2}, \tau_2 > 0$. The integral over $\tau_1$ enforces the BPS condition $Q^2 = Q_L^2 - Q_R^2 = 2\kappa$. Decomposing

$$P_{abcd}(Q,\tau_2) = \sum_{0\le\ell\le2} \tilde{P}_{abcd,\ell}(Q)\, \tau_2^{-\ell}, \tag{E.9}$$

where $\tilde{P}_{abcd,\ell}$ is a polynomial of degree $4-2\ell$ in $Q$, and integrating over $\tau_2$, we get

$$\begin{aligned}
F^{(p,q)}_{abcd}(s,\kappa) &= \frac{1}{\Gamma(2s)} \sum_{0\le\ell\le2} (4\pi\kappa)^{\ell+1-\frac{q}{2}} \sum_{\substack{Q\in\Lambda_{p,q}\\ Q^2=2\kappa}} \tilde{P}_{abcd,\ell}(Q) \left(\frac{Q_L^2}{2\kappa}\right)^{\ell+1-s-\frac{q-w}{2}} \\
&\quad \times \Gamma\left(s + \tfrac{q-w}{2} - \ell - 1\right) {}_2F_1\left(s + \tfrac{w}{2}, s + \tfrac{q-w}{2} - \ell - 1; 2s; \frac{2\kappa}{Q_L^2}\right) \\
&= \frac{1}{\Gamma(2s)} \sum_{1\le k\le3} (4\pi\kappa)^{\ell+1-\frac{q}{2}} \sum_{\substack{Q\in\Lambda_{p,q}\\ Q^2=2\kappa}} \tilde{P}_{abcd,\ell}(Q) \left(\frac{Q_R^2}{2\kappa}\right)^{\ell+1-s-\frac{q-w}{2}} \\
&\quad \times \Gamma\left(s + \tfrac{q-w}{2} - \ell - 1\right) {}_2F_1\left(s - \tfrac{w}{2}, s + \tfrac{q-w}{2} - \ell - 1; 2s; -\frac{2\kappa}{Q_R^2}\right)
\end{aligned} \tag{E.10}$$

where in the second line, we used Pfaff's equality ${}_2F_1(a,b;c;z) = (1-z)^{-b} {}_2F_1(b,c-a;c;\frac{z}{z-1})$. Similarly, for the scalar integral we get

$$F^{(p,q)}(s,\kappa) = \frac{(4\pi\kappa)^{1-\frac{q}{2}}}{\Gamma(2s)} \sum_{\substack{Q\in\Lambda_{p,q}\\ Q^2=2\kappa}} \left(\frac{Q_R^2}{2\kappa}\right)^{1-s-\frac{p+q}{4}} {}_2F_1\left(s + \tfrac{q}{4}, s + \tfrac{p+q}{4} - 1; 2s; -\frac{2\kappa}{Q_R^2}\right) \tag{E.11}$$

For $q < 6$, the series (E.10) and (E.11) are absolutely convergent at $s = 1 + \frac{k}{2}$, so the limit (E.7) can be taken term by term. For $q \ge 6$, the limit must be taken after analytically continuing the sum, and subtracting the pole when $q = 6$. In either case, the series (E.10) and (E.11) correctly encode the singular behavior of the integral at codimension-$q$ singularities in $G_{p,q}$ where $P_R^2 \to 0$ for a norm $2\kappa$ in $\Lambda_{p,q}$ or $Q_R^2 \to 0$ for a norm $2\kappa/N$ vector in $\Lambda_{p,q}^*$. Near these loci, the leading singular behavior of (E.10) is given, for $\kappa = 1$, by

$$F^{(p,q)}_{abcd} \sim \frac{\Gamma\left(\frac{q-2}{2}\right)}{(2\pi)^{\frac{q-2}{2}}} \left[\frac{Q_{L,a}Q_{L,b}Q_{L,c}Q_{L,d}}{(Q_R^2)^{\frac{q-2}{2}}} - \frac{6}{q-4}\frac{\delta_{(ab}Q_{L,c}Q_{L,d)}}{(Q_R^2)^{\frac{q-4}{2}}} - \frac{3}{(q-6)(q-4)}\frac{\delta_{(ab}\delta_{cd)}}{(Q_R^2)^{\frac{q-6}{2}}}\right] \tag{E.12}$$

and similarly for $F^{(p,q)}$.

Using the same argument as in (3.51) and making use of (E.2), it is easy to show that the integrals (E.5) satisfy the differential equation

$$\begin{aligned}
\mathcal{D}^2_{ef} F^{(p,q)}_{abcd}(s) &= (2-q)\delta_{ef} F^{(p,q)}_{abcd}(s) + (16-4q)\delta_{e)(a} F^{(p,q)}_{bcd)(f}(s) + 12\delta_{(ab} F_{cd)ef}(s) \\
&\quad + \int_{\Gamma_0(N)\backslash\mathcal{H}} \frac{d\tau_1 d\tau_2}{\tau_2^2} \frac{2(2s+k)}{2\kappa\Gamma(2s)} \mathcal{F}_N(s,\kappa,-k-2)\Gamma_{\Lambda_{p,q}}[P_{abcdef}]
\end{aligned} \tag{E.13}$$

The modular integral on the second line can again by evaluating by the unfolding trick, as a sum over vectors $Q \in \Lambda_{p,q}$ with $Q^2 = 2\kappa$ . For the relevant value $s = 1 + \frac{k}{2}$ with small $|k|$, such that $\mathcal{F}_N(s, \kappa, -k)$ is weakly holomorphic, $\mathcal{F}_N(s, \kappa, -k-2)$ vanishes so the sum over $Q$ must vanish. We have checked that this is indeed the case in the Euclidean case $q = 0, N = 1$, such that only a finite number of vectors $Q$ contribute.

## E.2 Eisenstein series for $O(p, q, \mathbb{Z})$

While the modular integrals (E.5) result into automorphic forms with singularities on $G_{p,q}$, due to the pole of order $\kappa$ in the Niebur-Poincaré series $\mathcal{F}_N(s, \kappa, w; \tau)$, it is useful to consider the analogue

$$E^{(p,q)}(\Phi, s) = \int_{\Gamma_0(N) \backslash \mathcal{H}} \frac{d\tau_1 d\tau_2}{\tau_2^2} \Gamma_{\Lambda_{p,q}} E_N(s, -\tfrac{p-q}{2}; \tau) , \tag{E.14}$$

where $\mathcal{F}_N(s, \kappa, w; \tau)$ is replaced by the non-holomorphic Eisenstein series for $\Gamma_0(N)$,

$$E_N(s, w; \tau) = \frac{1}{2} \sum_{\gamma \in \Gamma_\infty \backslash \Gamma_0(N)} \tau_2^{s - \frac{w}{2}} |_w \gamma , \tag{E.15}$$

which can be obtained formally by taking the limit $\kappa \to 0$ in (E.1). The integral converges for $\mathrm{Re}(s) > \frac{p+q-2}{2}$, and can be computed using the unfolding trick, leading to a standard vectorial Eisenstein series for $O(p, q, \mathbb{Z})$, the automorphism group of $\Lambda_{p,q}$,

$$E^{(p,q)}(\Phi, s) = \pi^{-s'} \Gamma(s') \sum_{\substack{P \in \Lambda_{p,q} \backslash \{0\} \\ P^2 = 0}} \frac{1}{(P_L^2 + P_R^2)^{s'}} , \tag{E.16}$$

with $s' = s + \frac{p+q}{4} - 1$. Another Eisenstein series for the same group is obtained by replacing $E_N(s, w; \tau)$ by its image under the Fricke involution, which amounts to changing $\Phi \mapsto \sigma \cdot \Phi$ in (E.16). Unlike (E.5), both Eisenstein series are smooth automorphic forms on $G_{p,q}$. Their behavior in the degeneration limits $O(p, q) \to O(p-1, q-1)$ and $O(p, q) \to O(p-2, q-2)$ is easily obtained by applying the same methods as in §4 and §5. In particular, the constant terms proportional to $\tau_2^{s - \frac{w}{2}}$ and to $\tau_2^{1-s-\frac{w}{2}}$ in the Fourier expansion of $E_N(s, w; \tau)$ lead to power-like terms proportional to $R^{2s'}$ and $R^{p+q-2-2s'}$ in the degeneration limit $O(p, q) \to O(p-1, q-1)$.

By direct computation, or using the fact that $E_N(s, w; \tau)$ is an eigenmode of the weight $w$ Laplacian on $\mathcal{H}$ with eigenvalue $(s - \frac{w}{2})(s - 1 + \frac{w}{2})$, one sees that

$$\Delta_{G_{p,q}} E^{(p,q)}(\Phi, s) = s'(2s' - p - q + 2) E^{(p,q)}(\Phi, s) . \tag{E.17}$$

For $s' = \frac{p+4}{2}$, corresponding to $s = 3 + \frac{p-q}{2}$, the eigenvalue coincides with the eigenvalue of $F^{(p,q)}$ in (3.27) (the other value $s' = \frac{q-6}{2}$, $s = -2 - \frac{p-q}{2}$ lies outside the fundamental domain, and is related to the former by the functional equation $s \mapsto 1 - s$). Moreover, using the same methods as in §3.2 it is easy to check that $E^{(p,q)}(\Phi, s)$ satisfies the second constraint in (3.27). It is thus natural to ask if the exact $(\nabla \Phi)^4$ coupling could involve an extra term proportional to $E^{(p,q)}(\Phi, 3 + \frac{p-q}{2})$ in addition to the proposed formula (2.27). However, it turns out that the latter contains terms of order $R^{p+4}$ and $R^{q-6}$ in the degeneration limit $O(p, q) \to O(p-1, q-1)$ with a non-zero coefficient, respectively, and the first term $R^{p+4}$ is ruled out by the differential equation (3.22).

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
