# Peer review of "Four-derivative couplings and BPS dyons in heterotic CHL orbifolds"

_SciPost Physics, doi:SciPost Phys. 3, 008 (2017)_

## Round 2 · Referee Report · Anonymous (Referee 1) · 2017-4-11

Strengths

1- clear starting point (the perturbative couplings they study) and motivation of the form of the exact couplings. 2- clear an non-trivial consistency checks : Ward identities, weak coupling limit, decompactification limit.

Weaknesses

1- It is not clear (to me) how much we learn, given the previous works in the same spirit in the subject.

Report

Dear Editor,

the authors consider heterotic string backgrounds with 16 supercharges in 3 dimensions. In one of their previous works [7], they have analyzed the case of a compactification on $T^7$, while the present manuscript considers models where an additional $Z_N$ free action is implemented. The effect of the latter is to reduce the rank of the gauge group, while preserving the supercharges. The authors focus on specific scalar couplings of the form $F(\Phi)(\nabla \Phi)^4$ and propose an exact expression for $F_{abcd}(\Phi)$.

They first determine the exact duality group the models in 3 dimensions are conjectured to satisfy. With the knowledge of the perturbative expression of $F_{abcd}$, Eq. (2.24), which involves tree-level and 1-loop contributions and is invariant under a perturbative T-duality group, they propose an exact expression invariant under the full, exact, U-duality group, Eq. (2.27). Then, they propose various consistency checks :

  • They determine the supersymmetric Ward identities associated to $F_{abcd}$ and show they are satisfied.

  • They consider the weak string coupling limit and show that their expression leads to the correct perturbative result in $D=3$, and a series of instanton corrections (NS5-branes, KK- and H-monopoles).

  • They consider the decompactification limit in 4 dimensions and show that the results reproduce the expected non-perturbative couplings in $D=4$, with additional exponentially suppressed corrections ($\frac{1}{2}$-BPS dyons winding the large circle and TAUB-NUT instantons).

The above strategy is not new. It is based on perturbative couplings covariantized under an exact duality group, and has been used for 20 years by now. It still remains instructive on the structure of the exact spectrum of string theory, but I must admit it is not clear (to me) how much we learn, as compared to the previous works (and announced future ones by the authors) on the subject. The manuscript contains few minor typos that are not disturbing to the reader. I just mention the references to Eq. (2.24) in the conclusion that should be replaced by (2.27).

The analysis of the authors is rigorous, with (more than) enough convincing arguments on the validity of the conjectured duality group and expression of the couplings they study. Altogether, I think this work deserves to be published.

Requested changes

1- minor typos.

  • validity: top
  • significance: good
  • originality: ok
  • clarity: top
  • formatting: perfect
  • grammar: excellent

Author:  Boris Pioline  on 2017-05-08  [id 127]

(in reply to Report 1 on 2017-04-11)
Category:
remark

We are grateful to the referee for his/her comments. In reply we would like to make the following remarks:

1- The strategy of covariantizing perturbative couplings under U-duality is admittedly not new, although up to now it has been mostly applied to the case of 1/2-BPS saturated couplings where the perturbative series stops at one-loop. The $(\nabla\Phi)^4$ coupling studied in the present work falls in this category, but its study is a necessary first step before tackling the more challenging case of the $\nabla^2(\nabla\Phi)^4$ coupling (which receives perturbative corrections at two-loop, and 1/4-BPS instanton corrections). Preliminary results for both $(\nabla\Phi)^4$ and $\nabla^2(\nabla\Phi)^4$ couplings were announced in the letter [7], and the present paper provides detailed proofs of these claims for $(\nabla\Phi)^4$ couplings, in a class of heterotic orbifolds which generalizes the model considered in [7].

2- Prior to our work, the spectrum of 1/2-BPS dyons in CHL orbifolds were only known for certain duality orbits. Our results provide the exact (indexed) degeneracies for all 1/2-BPS dyonic states in the case of the $N=2,3,5,7$ CHL orbifolds. In particular, we find that the degeneracies take two different forms, (2.16) and (2.17), depending whether the electromagnetic charge $(Q,P)$ is twisted or untwisted (a notion which we introduce in this work). In the revised version we have added a sentence in the abstract highlighting this result, which will play an important role in our future study of $\nabla^2(\nabla\Phi)^4$ couplings, since degeneracies of 1/4-BPS dyons exhibit wall-crossing behavior on loci where they can split into two 1/2-BPS dyons. In that work we shall also obtain the exact degeneracy of 1/4 BPS black holes in CHL models for prime $N$ in all duality orbits, complementing existing partial results in the literature.

---

## Round 2 · Referee Report · Anonymous (Referee 2) · 2017-4-23

Strengths

1- Clear explanation 2- Very detailed and careful calculations

Weaknesses

1- Minor corrections to be made

Report

The article under review proposes an ansatz for a four-derivative scalar coupling in the effective action of certain three dimensional models with 16 supersymmetries. The models are obtained from compactifications of heterotic strings on $T^7$ or some orbifold of $T^7$ by symmetries $\mathbb{Z}_N$ of order $2$, $3$, $5$, or $7$. The ansatz is U-duality invariant, satisfies the expected supersymmetric Ward identities and correctly reproduces the correct perturbative results in the weak coupling limit. In the limit of where one of the circles of $T^7$ decompactifies, the coupling reproduces the known four dimensional couplings, corrected by various kind of instantonic contributions, in particular the ones corresponding to 1/2 BPS states in four dimensions formally reduced along the time-like direction. The ansatz reproduces the expected degeneracy of four dimensional 1/2 BPS particles and makes precise predictions about the contributions of the NS5, KK-monopole and H-monopole instantons in the weak coupling limit and of the Taub-NUT instantons in the decompactification limit. This result is also an important step toward the study of 1/4 BPS saturated couplings.

The article is carefully written and very detailed. I have just a few minor comments and questions for the authors. Once these minor issues are fixed, I definitely recommend the paper for publication.

Requested changes

1- page 5, three lines above eq.(2.2): "The U-duality group $G_4(\mathbb{Z})$ now includes $\Gamma_0(N)\times O(r-6,6,\mathbb{Z})$" While I have no rigorous argument against this statement, I think it is far from obvious that the U-duality group contains the whole group $O(r-6,6,\mathbb{Z})$ of automorphisms of the lattice $\Lambda_{r-6,6}$. In fact, only a subgroup of $O(r-6,6,\mathbb{Z})$ is induced by dualities of the original model (before the CHL orbifold). The group $O(r-6,6,\mathbb{Z})$ includes the heterotic T-dualities described in section 3.3 of ref.[18]; while the authors of [18] claim that the latter are true dualities of the CHL model, their derivation is definitely not trivial. Do the authors have a reference for this statement or a good argument in favour of it? Otherwise I would suggest to make a weaker statement here (i.e. "We conjecture that") 2- page 8, five lines below (2.21): "The automorphism group of $II_{1,1}\oplus II_{1,1}[N]$ is then $\sigma_{T\leftrightarrow S} \ltimes[\Gamma_0(N)\times \Gamma_0(N)]$" This is not completely correct: the automorphism group of this lattice contains also a pair of Fricke involutions acting both on $T$ and $S$ (see for example ref.[18] or Theorem 1 in arXiv 1601.05412. (I guess that $\sigma_{T\leftrightarrow S}$ is the automorphism that exchanges $T$ and $S$, right?). 3- page 10, five lines below eq.(2.29): "are singular on on codimension-8 loci" $\rightarrow$ delete one "on" 4- page 15, five lines below eq.(3.32): "is finite for at generic points" $\rightarrow$ delete either "for" or "at" 5- page 27, three lines below eq.(4.41): "independantly yet" $\rightarrow$ "independently yet" 6- page 40, two lines below eq.(5.56): "mesure factor" $\rightarrow$ "measure factor" 7- page 40, four line from the bottom: "unifies S and T-duality and $D=4$"$\rightarrow$ I guess the authors mean "in $D=4$" here 8- page 41 beginning of the second paragraph: "It is natural to ask whether the fact that" $\rightarrow$ If I understand the sentence correctly, "the fact that" should be deleted.

  • validity: top
  • significance: high
  • originality: good
  • clarity: high
  • formatting: excellent
  • grammar: excellent

Author:  Boris Pioline  on 2017-05-08  [id 128]

(in reply to Report 2 on 2017-04-23)
Category:
answer to question

We are grateful to the referee for his/her comments. In reply we would like to make the following remarks:

  1. As the referee points out, the precise identification of the U-duality group for CHL models, both in $D=4$ and $D=3$, is subtle. In the first version we claimed above (2.2) that the U-duality group in $D=4$ includes $\Gamma_0(N)\times O(r-6,6,Z)$, where $O(r-6,6,Z)$ was defined as the group of automorphisms of the lattice $\Lambda_{r-6,6}$ of magnetic charges. This group preserves the four derivative coupling considered in the paper, but is almost certainly not a symmetry of the complete theory. However, a more precise discussion of the true U-duality group would break the flow of this introductory section, and we decided to focus on a proper subgroup of the U-duality group. As we explain on page 5 of the revised version, the U-duality group must contain $\Gamma_1(N)\times \widetilde{O}(r-6,6,Z)$, where $\widetilde{O}(r-6,6,Z)$ is the group of restricted automorphisms of $\Lambda_{r-6,6}$ which act trivially on the discriminant group. There are more dualities that descend from elements of $O(22,6,Z)$ which commute with the orbifold action. Moreover there are strong reasons to believe that Fricke involutions on two out of the three S,T,U moduli acting in pair are also dualities of the theory.

  2. Similarly, as we explain on page 8 of the revised version, the U-duality group in $D=3$ must contain $\widetilde{O}(r-4,8,Z)$ and is included in ${O}(r-4,8,Z)$. It is possible however that certain BPS observables are accidentally invariant under the larger group ${O}(r-4,8,Z)$. Indeed our proposal for the exact $(\nabla\Phi)^4$ couplings is invariant under $O(r-4,8,Z)$ extended by a Fricke involution, which implies that the $F^4$ and $R^2$ couplings in $D=4$ are invariant under $\Gamma_0(N)\times O(r-6,6,Z)$ and Fricke S-duality.

  3. We agree that the automorphism group of the lattice $II_{1,1}\oplus II_{1,1}[N]$ was not accurately stated below (2.21), as it includes additional Atkin-Lehner involutions. We replaced "is then" by "includes", referring to [18,29] for the precise determination of this group.

  4. We thank the referee for catching the various misprints, which we have corrected in the revised version.

---

## Round 3 · Author Response

We thank the referees for their valuable remarks, to which we have replied separately. In this revised version, we have made various changes in section 2.1 and 2.2, in order to clarify the relation between the automorphism group of the Narain lattice and the U-duality group. We have also corrected several misprints pointed out by the referee, made cosmetic changes to various equations, and added references [24,25]. In addition, we removed footnote 5 and the former reference [29] to Witten (1994). The reason is that the vector $Q$ in the extended Narain lattice cannot be interpreted as a charge vector for BPS particles in $D=3$, since the latter are instead classified by conjugacy classes of the U-duality group (as explained e.g. in [arXiv:1209.6056]). We hope that this revised version will be found suitable for publication.

---

## Round 3 · List of Changes

• In abstract, replaced "NS5-brane, Kaluza-Klein monopole and H-monopole instantons" by "non-perturbative contributions".
  • At end of abstract, added "in particular we obtain the exact helicity supertraces for 1/2-BPS dyonic states in all duality orbits."
  • Clarified the discussion of the U-duality group $G_4(Z)$, starting 4 lines before eq (2.2) up until 2 lines above eq (2.6).
  • Replaced $q$ by $q_S=e^{2\pi i S}$ in (2.5)
  • Above (2.14), added references [24,25] to Dabholkar et al (2005) and Sen (2005)
  • Defined $q=e^{2\pi i \tau}$ below (2.14)
  • Clarified the discussion of the U-duality group $G_3(Z)$, starting below eq (2.19) up until eq (2.22).
  • removed footnote 5, along with the sentence "(which can be viewed as a one-loop corrected mass formula)" and the former reference [29]
  • Restored missing index $J$ in (3.11)
  • Replaced $Q$ by $\widetilde{Q}$ in (4.8)
  • Explicited the integration domain In Eq (4.24), (4.29), (5.18), (5.28), (5.42)
  • Rewrote sentences around (4.36)
  • Removed the unwanted sum of $\widetilde{Q}$ in (5.9), and replaced $\Lambda_{p-1,q-1}$ by $\Lambda_{p-2,q-2}$ in that same equation
  • Third bullet on page 33 (starting with "The remaining contributions A with $(n_2,m_2)=(0,0)$...) moved into main text
  • Rewrote the sentence below (5.33)
  • Added "heterotic" in the title of Appendix A
  • Corrected various misprints pointed out by the referees

---

## Editorial Decision

published